# View Space: Learning Representation across Arbitrary Graphs

Dooho Lee [1]   Myeong Kong [1]   Minho Jeong [2]   Jaemin Yoo [2]

## Abstract

Generalizing pretrained models to unseen datasets without retraining is a central challenge toward foundation models. Achieving fully inductive inference on numerical data is particularly difficult due to large variations in feature dimensionality and semantics across datasets. We observe that, in the presence of graph structure, numerical data admits a distinct structure-induced representational axis beyond the feature space, which we formalize as the *view space*. This view space enables a unified representation of graphs with heterogeneous features and motivates *Graph View Transformation* (GVT), a class of parametric mappings that can be shared across arbitrary graphs. We instantiate this framework with Recurrent GVT, an architecture for fully inductive node representation learning in node classification. Pretrained on OGBN-Arxiv and evaluated on 27 benchmarks, Recurrent GVT outperforms GraphAny, the prior fully inductive graph model, by +8.93%, and surpasses 12 individually tuned GNNs by at least +3.30%. These results establish the view space as a principled and practical foundation for learning across graphs with heterogeneous feature spaces. Code and checkpoints are available in this link.

## 1. Introduction

Foundation models in natural language processing (NLP) and computer vision (CV) have transformed how models are built and shared across datasets (Brown et al., 2020; Kolesnikov et al., 2020; Radford et al., 2021; Touvron et al., 2023). The common approach is to pretrain a backbone encoder on large-scale corpora and then adapt it to smaller datasets with lightweight, per-dataset predictors (He et al., 2016; Devlin et al., 2019). This strategy enables cross-dataset inference without retraining the backbone, reducing costly per-dataset training and optimization while leveraging knowledge transferred from pretraining.

This paradigm is possible because models in these domains are trained and inferred on standardized input formats. In NLP, a tokenization maps arbitrary text into embeddings from a shared vocabulary (Sennrich et al., 2015; Vaswani et al., 2017). In CV, images can be resized or patchified to a fixed resolution without semantic loss (Krizhevsky et al., 2012; Dosovitskiy et al., 2021). These conventions create a common representation space across datasets—tokens or pixels—enabling trained models to perform inference on unseen datasets directly, without any training.

Extending this paradigm to graph-structured data is substantially more challenging. Graphs resist standardization: they vary in size, topology, and sparsity, while node attributes differ widely in dimensionality and semantics across datasets (Ribeiro et al., 2017; Pei et al., 2020; Platonov et al., 2023). Graph neural networks (GNNs) partially address variation in graph size by operating on node sets with connectivity and performing message passing (Kipf & Welling, 2017; Hamilton et al., 2017). Building on this, a large body of work further improves structural generalization across connectivity regimes, such as differences in connectivity patterns, sparsity levels, and homophily (Zeng et al., 2019; Qu et al., 2022; Zhao et al., 2023; Cantürk et al., 2024).

Despite these advances, many methods remain limited to per-dataset settings or families of graph datasets that share the same feature semantics (Liu et al., 2024a; Chen et al., 2024a;b). This limitation stems from feature transformations intrinsic to most GNNs, which are tightly coupled to the feature spaces observed during training, with restricted reuse across different feature spaces. Several recent works address this *feature heterogeneity* problem through SVD-based dimension matching and projection (Yu et al., 2025; Zhao et al., 2024) or learnable patching (Sun et al., 2025), but they still require additional per-dataset fine-tuning to operate on unseen datasets. As a result, performing inference on truly *arbitrary* graphs without any training remains largely under-explored due to feature heterogeneity.

Recent work, GraphAny (Zhao et al., 2025), has taken an important step toward this goal by formalizing learning across arbitrary graphs as *fully inductive* learning and proposing

[1]School of Electrical Engineering, KAIST, Daejeon, Republic of Korea [2]Department of Computer Science and Engineering, Seoul National University, Seoul, Republic of Korea. Correspondence to: Jaemin Yoo <jaeminyoo@snu.ac.kr>.

*Proceedings of the 43rd International Conference on Machine Learning*, Seoul, South Korea. PMLR 306, 2026. Copyright 2026 by the author(s).

the first model capable of handling arbitrary graph datasets. GraphAny addresses feature heterogeneity by learning from relative distances between linear predictions rather than directly from node features. However, it operates by attending over linear predictions rather than learning representations through transformations of the node-feature matrix, which limits its effectiveness as a representation learner.

While feature heterogeneity is a key obstacle, it is a general characteristic of numerical datasets. Tabular foundation models address this challenge by training on synthetic data to generalize across arbitrary feature spaces and by modeling Bayesian inference from labeled samples (Müller et al., 2021; Hollmann et al., 2022; 2025). However, these models do not incorporate graph structure and remain limited to high-dimensional raw features, which are common in graph datasets (Shchur et al., 2018; Yang et al., 2023).

In this work, we take an orthogonal approach unique to graph-structured data. While feature spaces vary, all graphs share a universal property: connectivity. We show that graph structure induces a distinct representational axis beyond features, which we formalize as the *view space*. By encoding graphs as fixed-size *view vectors*, the view space provides a standardized interface for learning across graphs with heterogeneous features. Our contributions are as follows:

1. We formalize the *fully inductive node representation learning* (FI-NRL) problem, where a model maps arbitrary graphs to node representations (Section 3).

2. We introduce the *view space*, a novel representational axis for graphs that provides a unified space for learning over arbitrary graphs (Section 4).

3. We propose a parametric mapping *Graph View Transformation* (GVT) and analyze its representational power relative to aggregations in GNNs (Section 5).

4. We present *Recurrent GVT* (RGVT), a practical realization of FI-NRL for node classification (Section 6).

For evaluation, we pretrain RGVT on OGBN-Arxiv (Hu et al., 2020a) and transfer it to 27 node classification benchmarks spanning diverse feature specifications. Only with a lightweight predictor, RGVT outperforms GraphAny (Zhao et al., 2025) on 26 out of 27 benchmarks, achieving an average gain of +8.93%, and surpasses 12 GNNs by at least +3.30%, despite each being individually tuned and trained for its target dataset. These results establish the view space as a principled and practical foundation for learning across graphs with heterogeneous feature spaces.

## 2. Preliminaries

**Tensor Notations.** For a vector $a$, let $a_i$ be its $i$-th element. For a matrix $A$, $A_{i,j}$ denotes the $(i,j)$-th entry, while $A_{i,:}$ and $A_{:,j}$ denote its $i$-th row and $j$-th column, respectively.

For a 3-D tensor $\mathbf{A}$, $\mathbf{A}_{i,j,k}$ denotes the element at position $(i, j, k)$, and $\mathbf{A}_{i,j,:}$ denotes the vector along the third mode.

**Graph Notations.** Let $\mathcal{G} = (V, E)$ be an undirected graph with $N = |V|$ nodes, where $V$ and $E$ denote the node and edge sets, respectively. The adjacency matrix is denoted as $A \in \{0, 1\}^{N \times N}$, where $A_{ij} = 1$ if and only if $(i, j) \in E$. The node-feature matrix is $X \in \mathbb{R}^{N \times F}$, where each row $X_{n,:}$ is a feature vector of node $n$, and each column $X_{:,f}$ is the $f$-th feature element across nodes.

**Graph Neural Networks.** Given a graph $\mathcal{G}$, a graph neural network (GNN) updates node representations layer by layer from the input node-feature matrix $Z^{(0)} = X$. At layer $l$, the node-representation matrix is denoted by $Z^{(l)} \in \mathbb{R}^{N \times F_l}$, where $F_l$ is the size of its representation. Each layer applies two key operations, *aggregation* and *combination* (Gilmer et al., 2017; Xu et al., 2018b; Hu et al., 2020b):

$$Z^{(l)} = \text{Combine}^{(l)}\big(\text{Aggregate}^{(l)}(Z^{(l-1)}, A), Z^{(l-1)}\big).$$

At each layer, the *aggregation* conveys structural information by propagation, multiplying the node-representation matrix $Z^{(l)}$ with a matrix $\hat{A} \in \mathbb{R}^{N \times N}$ derived from the adjacency matrix $A$ either statically (Kipf & Welling, 2017; Hamilton et al., 2017) or dynamically (Veličković et al., 2018; Chien et al., 2021). The *combination* then integrates aggregated results with the original node representations, typically through a feature transformation parametrized by a learnable weight matrix $W^{(l)} \in \mathbb{R}^{F_{l-1} \times F_l}$.

## 3. Representation Learning Across Graphs

We define *fully inductive node representation learning* (FI-NRL) as learning a representation function $\Psi(X, A) = Z$ that can map any unseen graph to node representations that jointly encode node features and graph structure in a latent space. A fully inductive function $\Psi$ must satisfy

$$\Psi : \mathbb{R}^{N \times F} \times \{0, 1\}^{N \times N} \to \mathbb{R}^{N \times H} \quad \text{for all } N, F \geq 1,$$

ensuring that $\Psi$ operates on graphs with arbitrary numbers of nodes $N$ and arbitrary feature dimensionalities $F$, where the adjacency matrix encodes graph connectivity.

This formulation allows dataset-specific, lightweight predictors to be trained directly on the learned representations $Z$, while the representation function $\Psi$ is shared across arbitrary graphs. We next introduce two requirements that guarantee a function $\Psi$ is fully inductive.

**R1.** The function $\Psi$ must be **equivariant to node permutations** to ensure it respects graph isomorphisms. For any permutation matrix $P \in \{0, 1\}^{N \times N}$:

$$\Psi(PX, PAP^\top) = P\,\Psi(X, A).$$

**R2.** To support heterogenous feature sets across graphs, $\Psi$ must also be **equivariant to feature permutations**.

For any permutation matrix $\boldsymbol{Q} \in \{0,1\}^{F \times F}$:

$$\Psi(\boldsymbol{XQ}, \boldsymbol{A}) = \Psi(\boldsymbol{X}, \boldsymbol{A})\boldsymbol{Q}.$$

**Lemma 3.1.** *A function $\Psi$ that is equivariant to both node permutations (**R1**) and feature permutations (**R2**) is well-defined for graphs of arbitrary size $(N, F)$.*

The formal proof is provided in Appendix B. Intuitively, permutation equivariance enforces order-agnostic, set-based operations that naturally handle variable-sized inputs (e.g., a node's neighborhood), making the function independent of the number of elements, namely $N$ and $F$ in our case.

**R1** is a standard property satisfied by most graph models, including GNNs (Kipf & Welling, 2017; Hamilton et al., 2017; Veličković et al., 2018; Xu et al., 2018a; Maron et al., 2019), as they are explicitly designed to operate on graphs of arbitrary size. In contrast, **R2** poses a key challenge that most existing GNNs fail to meet. This is because they encode knowledge through feature transformations that rely on learnable weight matrices, e.g., $\boldsymbol{W} \in \mathbb{R}^{F \times F'}$. which are tied to the fixed feature dimensionality and its ordering. As a result, such learned parameters do not generalize to graphs with different feature specification. Together, R1 and R2 characterize the core challenge of FI-NRL.

## 4. The View Space

We introduce the *view space*, a novel representation space in which graphs with arbitrary numbers of nodes, edges, and features are mapped into consistent, fixed-dimensional *view vectors*. We then propose *Graph View Transformation* (GVT), which is the parametric transformation satisfying dual permutation equivariance stated in Section 3 via the view space. Formal proofs of Lemma 4.2 and Theorem 4.7 in this section are provided in Appendix D.

### 4.1. Uncovering the Hidden Space of Graphs

The node-feature matrix $\boldsymbol{X} \in \mathbb{R}^{N \times F}$ of a graph naturally exhibits two orthogonal spaces. Each row $\boldsymbol{X}_{n,:}$ lies in the *feature space* $\mathbb{R}^F$, representing the features of node $n$, while each column $\boldsymbol{X}_{:,f}$ lies in the *node space* $\mathbb{R}^N$, representing the feature $f$ across nodes. However, this two-dimensional representation cannot simultaneously achieve permutation equivariance along both axes: feature-axis transformations violate R2, while node-axis transformations violate R1.

Our key idea to overcome this limitation is to lift the representation into a higher-dimensional space by adding an extra axis, yielding a tensor $\mathbf{X} \in \mathbb{R}^{N \times F \times C}$. Then, the new axis $C$ permits transformations that remain equivariant under both node and feature permutations.

We introduce the *view space*, the third axis of the graph representation. This axis is formed through a *view stacking*

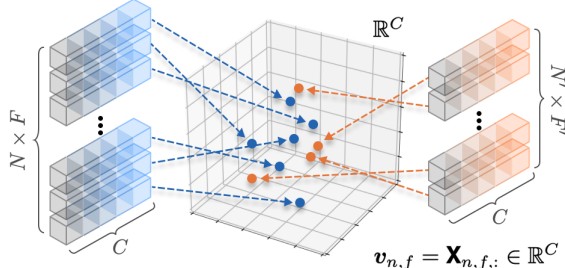

*Figure 1.* Graphs of varying sizes and features can be mapped into the view space as $N \times F$ view vectors $\boldsymbol{v}_{n,f} = \mathbf{X}_{n,f,:} \in \mathbb{R}^C$.

operation, which collects multiple versions of the node-feature matrix and arranges them along this new dimension. Each version is produced by a *view finder*, with each view finder generating a distinct node-feature matrix that captures a different structural perspective of the graph.

**Definition 4.1** (View Finder). Given an adjacency matrix $\boldsymbol{A} \in \mathbb{R}^{N \times N}$, a view finder is a matrix-valued function $\nu : \mathbb{R}^{N \times N} \to \mathbb{R}^{N \times N}$ that is node-permutation equivariant:

$$\nu(\boldsymbol{PAP}^\top) = \boldsymbol{P}\,\nu(\boldsymbol{A})\,\boldsymbol{P}^\top, \quad \forall \boldsymbol{P} \in \{0,1\}^{N \times N}.$$

Applying $\nu(\boldsymbol{A})$ to $\boldsymbol{X} \in \mathbb{R}^{N \times F}$ produces a propagated node-feature matrix $\nu(\boldsymbol{A})\boldsymbol{X} \in \mathbb{R}^{N \times F}$.

**Lemma 4.2.** *Common adjacency preprocessing methods used in GNNs—including self-augmented adjacency, degree-normalized variants (row-stochastic or symmetric) (Kipf & Welling, 2017; Hamilton et al., 2017), spectral filters (Defferrard et al., 2016), diffusion kernels (Gasteiger et al., 2019), and their polynomials—are all valid view finders.*

As noted in Lemma 4.2, view finders are not new; they correspond to the adjacency preprocessing methods already used in the aggregation of GNNs. Prior studies show that different preprocessing choices emphasize different structural aspects of a graph, leading to distinct propagation outcomes and variations in performance across graphs (Mao et al., 2023; Subramonian et al., 2024).

From a set of distinct view finders, we obtain multiple node-feature matrices propagated differently, each capturing a distinct "view" of the graph. The *view stacking* operation then combines these matrices along an additional axis, producing a node-feature-view tensor $\mathbf{X} \in \mathbb{R}^{N \times F \times C}$.

**Definition 4.3** (View Stacking). Given an adjacency matrix $\boldsymbol{A}$, a node feature matrix $\boldsymbol{X}$, and view finders $\{\nu_c\}_{c=1}^C$, the view stacking operator $\mathcal{V}$ is defined as

$$\mathbf{X} := \mathcal{V}(\boldsymbol{X}, \boldsymbol{A}) = [\,\nu_c(\boldsymbol{A})\boldsymbol{X}\,]_{c=1}^C \in \mathbb{R}^{N \times F \times C},$$

where $[\cdot]_{c=1}^C$ denotes the operation that stacks the $C$ propagated node-feature matrices along a new third dimension, forming the node-feature-view tensor $\mathbf{X}$.

Introducing the new axis re-organizes the representation such that each node-feature entry $(n, f)$ is associated with a $C$-dimensional vector capturing its responses across different views. The graph is therefore represented by $N \times F$ such vectors, all embedded in a shared $C$-dimensional space. We refer to each such vector as a *view vector*, and to the space they inhabit as the *view space*, defined below:

**Definition 4.4** (View Vector). For each node-feature pair $(n, f)$ in a graph, the view vector $\boldsymbol{v}_{n,f} \in \mathbb{R}^C$ is the vector obtained by slicing the node-feature-view tensor $\mathbf{X}$ along its third dimension, i.e., $\boldsymbol{v}_{n,f} = \mathbf{X}_{n,f,:}$.

**Definition 4.5** (View Space). The view space is the vector space $\mathbb{R}^C$ in which the set of all view vectors $\{\boldsymbol{v}_{n,f}\}$ resides. Each coordinate axis corresponds to a specific view finder.

Since the number $C$ and the ordering of elements in view vectors are determined solely by the predefined set of view finders, any graph—regardless of its number of nodes or features—can be represented in the view space by $N \times F$ view vectors of dimension $C$, as illustrated in Figure 1. This provides graphs a standardized input format with shareable space, analogous to those in NLP and CV.

### 4.2. Learning Representations via View Space

To leverage this common space for representation learning, we introduce *Graph View Transformation* (GVT), a parametric function $\Psi$ that transforms the node-feature matrix $\boldsymbol{X}$ via the view space. GVT operates in two steps: (i) lifting the graph into the view space through view stacking, and (ii) applying a learnable mapping $\phi$ to collapse each view vector into a scalar, yielding the node-representation matrix $\boldsymbol{Z}$, as illustrated in Figure 2. We formalize this as follows.

**Definition 4.6** (Graph View Transformation). Given an adjacency matrix $\boldsymbol{A}$ and node features $\boldsymbol{X}$, the *Graph View Transformation* $\Psi$ maps to node representations $\boldsymbol{Z}$ as

$$\Psi(\boldsymbol{X}, \boldsymbol{A}) \;=\; \big[\, \phi(\mathbf{X}_{n,f,:} \mid \theta) \,\big]_{n \in [N],\, f \in [F]} \in \mathbb{R}^{N \times F},$$

where $\mathbf{X} = \mathcal{V}(\boldsymbol{X}, \boldsymbol{A})$ is the node-feature-view tensor obtained via view stacking (Definition 4.3), and $\phi : \mathbb{R}^C \to \mathbb{R}$ is a learnable dimension-collapsing function with parameters $\theta$, shared across all node-feature entries $(n, f)$.

The mapping $\phi$ takes each view vector $\mathbf{X}_{n,f,:}$ as input, learning patterns directly in the view space. Because $\phi$ is applied independently to each node-feature pair $(n, f)$, it is invariant to both node and feature order. Consequently, GVT satisfies dual permutation equivariance:

**Theorem 4.7.** *GVT is equivariant to both node permutations (**R1**) and feature permutations (**R2**) required for fully inductive node representation learning (FI-NRL).*

The dimension-collapsing mapping $\phi$ can take any form—such as linear projections, attention, or multi-layer percep-

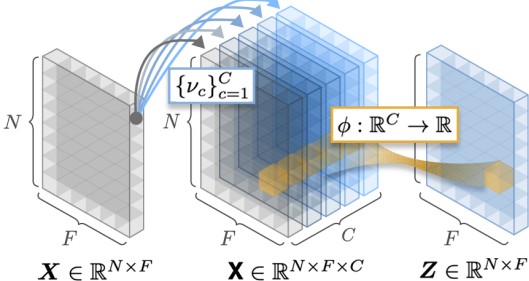

**Figure 2.** Graph View Transformation (GVT) transforms the node-feature matrix $\boldsymbol{X}$ into a node-feature-view tensor $\mathbf{X}$ through view stacking, and then applies $\phi$ to each view vector to produce a scalar, yielding the node-representation matrix $\boldsymbol{Z}$.

trons (MLPs)—while preserving the guarantees of Theorem 4.7. In this work, we use an MLP for its simplicity.

Thus, GVT is a parametric transformation through the view space that satisfies the requirements of FI-NRL. It thereby serves as a representation function across arbitrary graphs.

## 5. Representational Power of GVT

As GVT is provably fully inductive, it may seem unclear how it injects structural information into node representations in unseen graphs. In this section, we show that nonlinear GVT enables dynamic structural aggregation specific to each node-feature pair, which goes beyond the aggregation abilities of existing GNNs. Formal proofs of Lemma 5.1 and Lemma 5.2 are provided in Appendix D.

### 5.1. As Learnable Aggregation

We first consider the linear case of GVT which uses an affine function as the dimension-collapsing mapping: $\phi(\boldsymbol{v} \mid \boldsymbol{g}, b) = \boldsymbol{g}^\top \boldsymbol{v} + b$, with learnable parameters $\boldsymbol{g} \in \mathbb{R}^C$ and $b \in \mathbb{R}$. This simplifies GVT to the following formulation:

$$\Psi(\boldsymbol{X}, \boldsymbol{A}) = \big(\Sigma_{c=1}^C g_c\, \nu_c(\boldsymbol{A})\big)\boldsymbol{X} + b\boldsymbol{1}_{N \times F}.$$

The resulting equation shows that the linear GVT combines adjacency matrices from different view finders using learned weights $g_c$ to form a new adjacency, which is then applied to the node-feature matrix. This formulation generalizes the static aggregation operations used in many GNNs.

**Lemma 5.1.** *With a set of view finders including the identity and powers of the symmetric and row-normalized adjacency matrices, a linear GVT suffices to produce the aggregation operations of many GNNs (Kipf & Welling, 2017; Hamilton et al., 2017; Gasteiger et al., 2019; Wu et al., 2019; Chen et al., 2020; Chien et al., 2021).*

That is, static aggregation can be viewed as a special case of linear GVT. This explains why static aggregations (e.g., neighborhood averaging) apply consistently across arbitrary

graphs—they are instances of GVT, which is fully inductive as shown in Section 4. Learning a linear GVT can be seen as a data-driven search for an effective aggregation, and its inference on another graph amounts to adopting the learned knowledge of what constitutes an effective aggregation.

## 5.2. As Node-Feature Dynamic Aggregation

Introducing nonlinearity into GVT, such as using MLPs in the mapping $\phi$, enhances its expressivity by fundamentally altering the picture, as it can no longer be reduced to a weighted sum of adjacencies. To understand its behavior, we analyze it via local linear approximations.

**Lemma 5.2.** *Suppose that $\phi : \mathbb{R}^C \to \mathbb{R}$ is twice differentiable almost everywhere. For any $\boldsymbol{v}_0 \in \mathbb{R}^C$, Taylor's theorem gives the local affine approximation*

$$\phi(\boldsymbol{v}) \;=\; g(\boldsymbol{v}_0)^\top \boldsymbol{v} + b(\boldsymbol{v}_0) + R_2(\boldsymbol{v}; \boldsymbol{v}_0),$$

*where $g(\boldsymbol{v}_0) = \nabla_{\boldsymbol{v}} \phi(\boldsymbol{v}_0)$ and $b(\boldsymbol{v}_0) = \phi(\boldsymbol{v}_0) - g(\boldsymbol{v}_0)^\top \boldsymbol{v}_0$. If $|\nabla^2 \phi(\boldsymbol{v})| \le M$ near $\boldsymbol{v}_0$, then $|R_2(\boldsymbol{v}; \boldsymbol{v}_0)| \le \frac{M}{2} |\boldsymbol{v} - \boldsymbol{v}_0|^2$, so $\phi$ is locally affine with quadratic error decay.*

Thus, a nonlinear GVT behaves locally like a linear GVT around each view vector $\boldsymbol{v}_0$. In particular, the weights given by the gradient $g(\boldsymbol{v}_0) = \nabla_{\boldsymbol{v}} \phi(\boldsymbol{v}_0)$ depend on $\boldsymbol{v}_0$ unlike the static weights of the linear GVT globally applied to all node-feature pairs. This input dependence means that a nonlinear GVT implements an input-adaptive, or *dynamic*, aggregation in the sense of GNNs.

Each view vector corresponds to a specific node-feature pair $(n, f)$, allowing the nonlinear GVT to assign distinct aggregation behaviors at this fine-grained level. We refer to this property as *node-feature dynamic aggregation*. To the best of our knowledge, no existing GNNs achieve this granularity; prior work has been limited to node-wise (Zhang et al., 2021; 2022) or edge-wise (Veličković et al., 2018; Brody et al., 2021) dynamic aggregation. Thus, the nonlinear GVT is not only fully inductive but also unlocks new representational power through fine-grained aggregation.

Consequently, the nonlinear GVT learns from each view vector and determines its own aggregation method. Inference on a different graph is equivalent to adopting the learned mapping from view vectors to aggregation rules.

## 6. RGVT: A Fully Inductive Graph Encoder

Building on the fully inductive capability (Section 4) and expressivity (Section 5) of GVT, we instantiate FI-NRL for node classification by introducing a concrete architecture, RGVT, together with its training strategy.

**Decoupling Depth and Parameterization.** Fully inductive capability does not guarantee coverage of high-order graph structures, as many graphs demand broader receptive fields to capture long-range dependencies (Chen et al., 2020; Alon & Yahav, 2021). However, simply stacking GVT layers exhibits a key limitation: different graphs require different depths. Some tasks rely on local neighborhoods, others on multi-hop information, and the impact of over-smoothing varies across datasets. As a result, fixed-depth models often fail to generalize, even across graphs that share the same feature space (Xiao et al., 2023; Liu et al., 2024b).

To address this limitation, we introduce *Recurrent Graph View Transformation* (RGVT), drawing inspiration from recurrent neural networks (RNNs) (Hochreiter & Schmidhuber, 1997), which decouple parameterization from sequence length. Instead of stacking multiple GVT layers with distinct parameters, RGVT applies a single nonlinear GVT $\Psi$ recurrently for $L$ steps using shared parameters $\theta$, yielding

$$\boldsymbol{Z} = \Psi(\cdot, \boldsymbol{A} \mid \theta)^L(\boldsymbol{X}) \in \mathbb{R}^{N \times F}.$$

By decoupling parameterization from depth, the model can be applied at any $L$, allowing the depth to be selected per dataset without retraining. Recurrence expands the receptive field while adapting to dataset-specific requirements, ultimately improving generalization across diverse graphs.

**Training of RGVT.** We consider node classification as the downstream task, which is commonly used to evaluate the quality of node representations. We adopt an encoder–predictor paradigm for RGVT training. Given a graph dataset $\mathcal{G}_k$ with node feature matrix $\boldsymbol{X}_k$ and adjacency matrix $\boldsymbol{A}_k$, RGVT produces node representations by applying the encoder recurrently for $L_k$ steps:

$$\boldsymbol{Z}_k = \Psi(\boldsymbol{X}_k, \boldsymbol{A}_k \mid \theta)^{L_k}.$$

To adapt these representations to the dataset-specific label space $\boldsymbol{Y}_k \in \mathbb{R}^{N_k \times D_k}$, with $D_k$ as the number of classes, we attach a lightweight predictor $f_k$ and optimize

$$\arg\min_{\hat{\theta}} \mathcal{L}\big(f_k(\boldsymbol{Z}_k), \boldsymbol{Y}_k\big), \quad f_k : \mathbb{R}^{N_k \times F_k} \to \mathbb{R}^{N_k \times D_k}.$$

During pretraining, the optimization target $\hat{\theta}$ includes both the encoder parameters $\theta$ and the predictor $f_k$, whereas during adaptation to a new dataset, only $f_k$ is optimized given representations by frozen $\Psi$. This enables a pretrained RGVT to produce representations for arbitrary graphs without retraining, while lightweight predictors specialize the representations to each dataset. With an appropriate choice of recurrent depth $L_k$, such predictors can adapt effectively even in label-scarce or previously unseen graphs.

## 7. Related Works

**Graph Neural Networks.** GNNs have achieved notable success in graph learning by capturing structural information, but also face limitations such as over-smoothing, over-squashing, and inconsistent performance across nodes, features, and graphs (Alon & Yahav, 2021; Rusch et al., 2023;

*Table 1.* Performance (%) comparison against predictor architectures and GraphAny across datasets grouped by feature type. The numbers in parentheses represent the number of datasets in each category; for example, we include 5 graphs with signed dense features. **1st** and 2nd best results are highlighted. RGVT consistently and significantly outperforms all baselines.

| Method | OGBN-Arxiv (1) | Signed Dense (5) | Unsigned Dense (4) | Sparse (4) | Binary Dense (3) | Binary Sparse (8) | One-hot (4) | Total Avg. (28) |
|---|---|---|---|---|---|---|---|---|
| Linear | $52.44_{\pm0.04}$ | $53.29_{\pm0.08}$ | $75.67_{\pm0.64}$ | $66.41_{\pm0.57}$ | $72.18_{\pm0.69}$ | $57.11_{\pm0.77}$ | $38.86_{\pm2.52}$ | $59.41_{\pm0.84}$ |
| MLP | $53.80_{\pm0.14}$ | $55.08_{\pm0.31}$ | $75.86_{\pm0.68}$ | $69.02_{\pm0.77}$ | $72.88_{\pm0.64}$ | $57.65_{\pm1.84}$ | $39.34_{\pm2.43}$ | $60.43_{\pm1.20}$ |
| GraphAny (Wisconsin) | $57.77_{\pm0.45}$ | $59.12_{\pm0.46}$ | $71.78_{\pm1.22}$ | $81.61_{\pm0.20}$ | $83.44_{\pm0.24}$ | $55.25_{\pm2.62}$ | $52.68_{\pm0.40}$ | $64.72_{\pm0.96}$ |
| GraphAny (Cora) | $58.58_{\pm0.10}$ | $59.38_{\pm0.42}$ | $71.76_{\pm1.60}$ | $81.49_{\pm0.10}$ | $83.35_{\pm0.09}$ | $53.40_{\pm2.44}$ | $53.30_{\pm0.93}$ | $64.30_{\pm0.94}$ |
| GraphAny (Arxiv) | $58.63_{\pm0.14}$ | $59.70_{\pm0.36}$ | $72.62_{\pm1.32}$ | $81.68_{\pm0.23}$ | $83.56_{\pm0.08}$ | $54.18_{\pm2.49}$ | $53.02_{\pm0.35}$ | $64.71_{\pm0.89}$ |
| GraphAny (Best) | $58.63_{\pm0.14}$ | $59.74_{\pm0.32}$ | $72.64_{\pm1.02}$ | $81.89_{\pm0.12}$ | $83.92_{\pm0.13}$ | $55.58_{\pm2.47}$ | $53.81_{\pm0.77}$ | $65.30_{\pm0.93}$ |
| **RGVT+ Linear (Arxiv)** | $70.14_{\pm0.28}$ | $64.95_{\pm0.44}$ | $76.44_{\pm0.39}$ | $84.33_{\pm0.45}$ | $85.11_{\pm0.54}$ | $62.77_{\pm1.93}$ | $58.85_{\pm3.14}$ | $70.03_{\pm1.26}$ |
| **RGVT+ MLP (Arxiv)** | $71.11_{\pm0.28}$ | $66.37_{\pm0.90}$ | $77.12_{\pm0.45}$ | $83.98_{\pm0.81}$ | $84.86_{\pm0.51}$ | $63.87_{\pm1.58}$ | $62.48_{\pm3.95}$ | $71.13_{\pm1.41}$ |

Mao et al., 2023). To mitigate these, prior works explored structure editing (Jin et al., 2020; Ju et al., 2023), feature editing (Liu et al., 2021; Lee et al., 2025), and dynamic aggregation at the edge (Veličković et al., 2018; Brody et al., 2021) or node level (Zhang et al., 2021; 2022). Yet most approaches remain learning from the feature space, despite the aggregation is a view-space operation (Section 5). In contrast, GVT operates directly in the view space, leveraging its underexplored representational capacity.

**Graph Foundation Models.** Some existing works named graph foundation models (GFMs) seek to improve generalization across different datasets. One line of GFMs (Liu et al., 2024a; Chen et al., 2024a) focuses on text-attributed graphs, inducing a shared feature space via natural language; however, many graphs lack textual attributes, and converting numerical features into text often leads to information loss and degraded performance (Chen et al., 2024b). Another direction unifies non-textual feature spaces using singular value decomposition with alignment (Yu et al., 2025; Zhao et al., 2024) or learnable patching (Sun et al., 2025), but these approaches typically assume per-dataset fine-tuning to handle unseen feature spaces. In contrast, our work differs in two key aspects: (1) we target full *feature heterogeneity* to operate on arbitrary graphs with any feature sets, texts or non-texts; and (2) we adopt a fully inductive setting that requires no per-dataset fine-tuning during adaptation.

**GraphAny.** GraphAny (Zhao et al., 2025) encodes knowledge in the relative-distance space of predictions, which is consistently defined across arbitrary graphs. By attending over a set of pseudo-inverse-based linear predictors, it becomes the first model to support fully inductive inference. While our work also targets fully inductive learning, we focus on representation learning (FI-NRL) that maps graphs to node embeddings rather than directly producing predictions. This offers two key advantages: (1) representations can be reused across different label sets; and (2) unlike GraphAny's reliance on linear predictors, representation learning supports a flexible choice of downstream predictors, enabling more expressive and stable label mappings.

# 8. Experiments

Our experiments evaluate how well view-space knowledge learned by RGVT transfers across arbitrary graphs. After pretraining, frozen RGVT is applied to unseen downstream graphs, training only a lightweight predictor, with recurrent depth $L$ selected by validation accuracy. We use two predictors: a linear classifier and a one-hidden-layer MLP.

**Datasets** We largely follow prior work (Zhao et al., 2025) for dataset selection and their splitting. Pretraining is conducted on OGBN-Arxiv (Hu et al., 2020a) using its public split, and downstream evaluation is conducted on 27 node-classification benchmarks with diverse graph structures and features. We use public splits when available; otherwise, we sample 20 nodes per class for training, split the remainder evenly for validation and testing. Due to the large number of benchmarks, results are grouped by feature type in the main paper; the full dataset list, grouping details and complete results are provided in Appendix Q and Appendix R.

**Selection of View Finders.** For RGVT, we use $\{\boldsymbol{I}\} \cup \{(\boldsymbol{D}^{-1}\boldsymbol{A})^k, (\boldsymbol{D}^{-\frac{1}{2}}\boldsymbol{A}\boldsymbol{D}^{-\frac{1}{2}})^k\}_{k=1}^K$ as the set of view finders, where $\boldsymbol{D}$ is the degree matrix, and consider $K \in \{1, 2, 3\}$ as a hyperparameter. This choice is motivated by the fact that many aggregation operations used in popular GNNs can be expressed as linear combinations of elements in this set, and are therefore realizable by linear GVT, as shown in Lemma 5.1. We further present empirical analyses of the influence of the view-finder set on performance, along with guidelines for selecting candidate sets, in Appendix I.

**Implementation Details.** The search space includes the number of MLP layers in the mapping $\phi$, the recurrent depth $L$ during pretraining, the maximum view-finder order $K$ and the learning rate. Hyperparameters are chosen using validation accuracy on the pretraining dataset (OGBN-Arxiv) and then fixed to ensure downstream graphs remain unseen. All experiments are repeated five times with independent seeds, and we report the mean and standard deviation. Detailed information is given in Appendix J.

*Table 2.* Performance (%) comparison against 12 dataset-specialized GNNs across datasets grouped by feature type. 1st, 2nd, 3rd best results are highlighted. RGVT achieves the best average performance with the MLP predictor and ranks 2nd with the linear predictor.

| Method | OGBN-Arxiv (1) | Signed Dense (5) | Unsigned Dense (4) | Sparse (4) | Binary Dense (3) | Binary Sparse (8) | One-hot (4) | Total Avg. (28) |
|---|---|---|---|---|---|---|---|---|
| GIN | $65.77_{\pm 1.08}$ | $55.57_{\pm 3.94}$ | $67.95_{\pm 3.90}$ | $71.93_{\pm 4.66}$ | $44.83_{\pm 4.72}$ | $43.30_{\pm 2.49}$ | $55.31_{\pm 3.88}$ | $54.98_{\pm 3.70}$ |
| GAT | $71.89_{\pm 0.24}$ | $60.89_{\pm 0.31}$ | $73.24_{\pm 0.62}$ | $82.50_{\pm 1.26}$ | $83.36_{\pm 0.90}$ | $49.20_{\pm 2.58}$ | $54.39_{\pm 4.98}$ | $63.88_{\pm 1.87}$ |
| GATv2 | $72.13_{\pm 0.11}$ | $64.45_{\pm 0.45}$ | $72.00_{\pm 1.22}$ | $81.68_{\pm 1.44}$ | $83.20_{\pm 0.80}$ | $49.61_{\pm 2.80}$ | $56.12_{\pm 3.41}$ | $64.57_{\pm 1.83}$ |
| S$^2$GC | $69.17_{\pm 0.02}$ | $59.94_{\pm 0.08}$ | $75.28_{\pm 0.35}$ | $82.04_{\pm 0.29}$ | $84.48_{\pm 0.14}$ | $52.28_{\pm 1.15}$ | $56.97_{\pm 1.35}$ | $65.31_{\pm 0.64}$ |
| SGC | $68.69_{\pm 0.03}$ | $58.95_{\pm 0.05}$ | $75.18_{\pm 0.26}$ | $82.36_{\pm 0.37}$ | $84.73_{\pm 0.10}$ | $51.15_{\pm 0.85}$ | $62.54_{\pm 3.24}$ | $65.66_{\pm 0.82}$ |
| JKNet | $71.73_{\pm 0.24}$ | $63.18_{\pm 1.19}$ | $76.12_{\pm 0.42}$ | $83.30_{\pm 0.71}$ | $84.02_{\pm 0.49}$ | $51.72_{\pm 1.60}$ | $58.46_{\pm 4.93}$ | $66.19_{\pm 1.59}$ |
| APPNP | $70.85_{\pm 0.15}$ | $64.86_{\pm 0.21}$ | $76.27_{\pm 0.36}$ | $83.66_{\pm 0.38}$ | $84.11_{\pm 0.21}$ | $55.21_{\pm 1.47}$ | $49.29_{\pm 1.26}$ | $66.26_{\pm 0.77}$ |
| GCN | $71.19_{\pm 0.30}$ | $61.49_{\pm 0.27}$ | $76.00_{\pm 0.38}$ | $83.48_{\pm 0.43}$ | $84.43_{\pm 0.26}$ | $53.03_{\pm 1.49}$ | $59.51_{\pm 1.43}$ | $66.46_{\pm 0.82}$ |
| GCNII | $72.05_{\pm 0.08}$ | $67.11_{\pm 0.25}$ | $77.54_{\pm 0.45}$ | $81.00_{\pm 0.97}$ | $83.65_{\pm 1.15}$ | $54.90_{\pm 1.73}$ | $56.11_{\pm 4.25}$ | $67.30_{\pm 1.47}$ |
| SAGE | $71.22_{\pm 0.24}$ | $67.98_{\pm 0.34}$ | $76.63_{\pm 0.65}$ | $82.14_{\pm 0.90}$ | $83.69_{\pm 0.51}$ | $60.07_{\pm 2.57}$ | $51.84_{\pm 3.39}$ | $68.36_{\pm 1.55}$ |
| GPRGNN | $69.45_{\pm 0.41}$ | $67.24_{\pm 0.28}$ | $76.78_{\pm 0.18}$ | $84.35_{\pm 0.40}$ | $85.26_{\pm 0.31}$ | $60.61_{\pm 1.89}$ | $50.40_{\pm 1.65}$ | $68.68_{\pm 0.94}$ |
| UniMP | $71.78_{\pm 0.12}$ | $69.09_{\pm 0.34}$ | $76.97_{\pm 0.60}$ | $82.28_{\pm 0.62}$ | $84.22_{\pm 0.54}$ | $59.15_{\pm 3.46}$ | $54.94_{\pm 3.61}$ | $68.86_{\pm 1.80}$ |
| **RGVT** + Linear | $70.14_{\pm 0.28}$ | $64.95_{\pm 0.44}$ | $76.44_{\pm 0.39}$ | $84.33_{\pm 0.45}$ | $85.11_{\pm 0.54}$ | $62.77_{\pm 1.93}$ | $58.85_{\pm 3.14}$ | $70.03_{\pm 1.26}$ |
| **RGVT** + MLP | $71.11_{\pm 0.28}$ | $66.37_{\pm 0.90}$ | $77.12_{\pm 0.45}$ | $83.98_{\pm 0.81}$ | $84.86_{\pm 0.51}$ | $63.87_{\pm 1.58}$ | $62.48_{\pm 3.95}$ | $71.13_{\pm 1.41}$ |

**Baselines.** We evaluate three categories of baselines. (i) A linear classifier and a one-hidden-layer MLP, which serve as RGVT's predictors. (ii) GraphAny (Zhao et al., 2025), the prior fully inductive graph model, with three variants pretrained on the Wisconsin, Cora, and OGBN-Arxiv datasets, respectively. (iii) Twelve supervised GNNs spanning diverse aggregation operations. Unlike RGVT and GraphAny, these models are trained and extensively tuned separately on each dataset, making them highly specialized. Further details of all baselines are provided in Section J.2.

### 8.1. Overall Performance

**Validation of FI-NRL.** In Table 1, we first validate whether fully inductive node representations produced by RGVT remain effective when adapted to graphs with unseen graph structures and features. Across all datasets, RGVT achieves substantial gains over its predictor models, outperforming the linear classifier by +17.71% and MLP by +17.88% on average. This demonstrates that the pretrained RGVT effectively integrates structural information into node representations, while being useful even when transferring from dense word-embeddings (OGBN-Arxiv) to categorical, binary, or one-hot vectors. Overall, these results indicate that knowledge captured in the view space is not only theoretically but also empirically fully inductive.

We then compare RGVT with GraphAny, another fully inductive graph model. As shown in Table 1, RGVT surpasses GraphAny even in its best-performing variant by +7.24% with the linear predictor and +8.93% with the MLP predictor on average. These results demonstrate that (1) GVT's representation-predictor framework is substantially more expressive and robust than GraphAny's predictor-attention approach, and (2) GVT's node-feature dynamic aggregation exploits structural information across diverse graphs more

effectively than GraphAny, whose linear predictors operate with fixed, static feature propagation.

**Power of FI-NRL.** We compare RGVT with 12 different GNNs. This comparison favors individual GNNs, as they are extensively tuned and trained on each downstream graph, whereas RGVT operates fully inductively, adapting only the recurrent depth $L$ and a lightweight predictor for each dataset. This comparison assesses whether view-space knowledge, transferred from graphs with completely different feature spaces, can stand against dataset-specific graph knowledge obtained through conventional GNN training.

Remarkably, RGVT with both predictors outperforms all 12 GNNs on average, as shown in Table 2. Moreover, RGVT with the MLP predictor surpasses the best-performing GNN, UniMP (Shi et al., 2021), by an average margin of +3.30%. This demonstrates that fully inductive view-space knowledge can match or even exceed the dataset-specific knowledge of supervised GNNs. It suggests that feature transformations are not essential for understanding graphs—*graph view transformation* (GVT) provides a powerful alternative.

Finally, a Wilcoxon signed-rank test (Appendix A) further confirms the statistical significance of RGVT over each GNN across 28 datasets ($p < 0.05$), with GPRGNN (Chien et al., 2021) as the only exception ($p = 0.07$). Complete per-dataset results are provided in Appendix R.

### 8.2. Ablation Studies

In Table 3, we examine the role of two key components of RGVT: nonlinearity and recurrence. Removing nonlinearity reduces accuracy to 68.12%, comparable to GNN baselines. This is expected because without nonlinearity, GVT collapses into a learnable aggregation (Section 5), giving it expressive power fundamentally similar to that of standard

*Table 3.* Ablation results (%) highlighting the contribution of nonlinearity and recurrence in RGVT.

| Method | OGBN-Arxiv (1) | Signed Dense (5) | Unsigned Dense (4) | Sparse (4) | Binary Dense (3) | Binary Sparse (8) | One-hot (4) | Total Avg. (28) |
|---|---|---|---|---|---|---|---|---|
| **RGVT + MLP** | $\mathbf{71.11_{\pm 0.28}}$ | $\mathbf{66.37_{\pm 0.90}}$ | $\mathbf{77.12_{\pm 0.45}}$ | $\mathbf{83.98_{\pm 0.81}}$ | $\mathbf{84.86_{\pm 0.51}}$ | $\mathbf{63.87_{\pm 1.58}}$ | $\mathbf{62.48_{\pm 3.95}}$ | $\mathbf{71.13_{\pm 1.41}}$ |
| w/o non-linearity | $70.22_{\pm 2.55}$ | $64.53_{\pm 2.67}$ | $75.89_{\pm 1.86}$ | $78.82_{\pm 5.99}$ | $84.16_{\pm 2.46}$ | $61.12_{\pm 5.66}$ | $56.13_{\pm 6.38}$ | $68.12_{\pm 4.39}$ |
| w/o recurrence | $70.91_{\pm 0.17}$ | $63.73_{\pm 0.43}$ | $73.79_{\pm 0.66}$ | $82.61_{\pm 0.79}$ | $83.90_{\pm 0.63}$ | $53.29_{\pm 1.92}$ | $54.53_{\pm 2.25}$ | $65.73_{\pm 1.22}$ |
| w/o both | $70.53_{\pm 0.25}$ | $61.69_{\pm 3.57}$ | $75.10_{\pm 0.43}$ | $77.52_{\pm 7.46}$ | $84.57_{\pm 0.53}$ | $53.41_{\pm 5.59}$ | $54.73_{\pm 1.86}$ | $64.96_{\pm 3.68}$ |

GNNs. Eliminating recurrence leads to an even larger drop in accuracy, underscoring its role in receptive-field expansion and depth adaptation, both of which are critical for generalization across diverse graphs (Section 6). Further ablations on recurrent depth during pretraining and on the choice of pretraining dataset are reported in Appendix K.

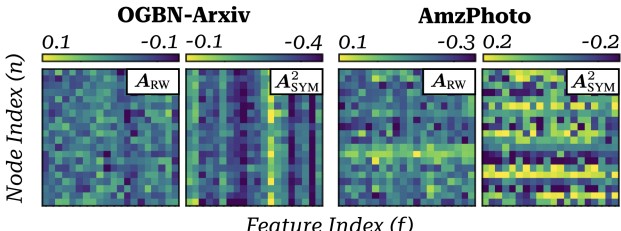

*Figure 3.* Node-feature heatmaps of linearly approximated aggregation weights for $\boldsymbol{A}_{\mathrm{RW}}$ and $\boldsymbol{A}_{\mathrm{SYM}}^2$ at the first layer of RGVT on the pretraining (OGBN-Arxiv) and transferred (AmzPhoto) datasets.

### 8.3. Visualization of Node-feature Dynamic Aggregation

To empirically verify the node-feature dynamic aggregation of nonlinear GVT (Section 5) in practice, we compute the contribution of the $c$-th view to each node-feature pair using local linearization done in Lemma 5.2. We visualize these contributions as heatmaps over sampled nodes and features (Figure 3), where the $x$-axis denotes feature indices, the $y$-axis denotes node indices, and the color intensity represents the corresponding linearized weights.

The distinct weight patterns across views ($\boldsymbol{A}_{\mathrm{RW}}$ vs. $\boldsymbol{A}_{\mathrm{SYM}}^2$) show that GVT performs adaptive view aggregation. Different node-feature pairs assign different weights to each view, confirming dynamic aggregation at the node-feature level. This behavior persists from the pretraining graph (OGBN-Arxiv) to a downstream graph (AmzPhoto), suggesting that node-feature dynamic aggregation transfers across datasets. The row- and column-wise patterns in the heatmaps further suggest that view vectors encode shared semantics across nodes and features, which GVT implicitly captures.

### 8.4. Additional Analysis

Due to space limitations, we summarize analyses of computational cost and text-attributed graphs here, with full details in Appendix H and Appendix G, respectively.

**Computational Cost.** Theoretically, RGVT scales linearly with the number of nodes $N$ and feature dimensionality $F$. Empirically, although RGVT is slower than typical GNNs, pretraining on OGBN-Arxiv takes about 10 minutes and inference across all 28 downstream datasets requires 5 ms on NVIDIA RTX A6000, remaining well within a practical range. Moreover, by eliminating extensive per-dataset hyperparameter tuning, RGVT offers substantially greater efficiency when adapting to new datasets; each GNN required *at least two days* of tuning in our experiments.

**Text-attributed Graphs.** We further evaluate RGVT on seven text-attributed graph datasets using LLM-embedded features from recent work (Wang et al., 2025). While GraphAny exhibits performance degradation relative to the GNNs, RGVT achieves competitive performance and ranks first on two datasets. These results suggest that GraphAny's linear predictors are limited in exploiting the rich information in LLM-embedded features, whereas RGVT effectively preserves such information in its representations.

## 9. Discussion

In this section, we discuss GVT's design tradeoffs, current limitations, and future research directions.

### 9.1. Design Tradeoffs

To achieve feature equivariance, GVT learns solely along the view axis and avoids explicit feature mixing. As a result, GVT cannot directly model cross-feature interactions. However, this does not imply that GVT eliminates such interactions, since it preserves the original feature dimensions without compression, allowing downstream predictors to capture them. The t-SNE visualizations of raw features and output representations in Appendix N further suggest that GVT preserves the underlying feature structure effectively.

This feature-wise design is also closely tied to scalability. Because GVT operates independently on each feature, its complexity scales linearly with the feature dimension. In contrast, explicitly modeling feature interactions while preserving feature equivariance would generally require feature-wise attention or transformer-style operations, resulting in quadratic complexity with respect to the number of features. Such approaches quickly become impractical for many real-

world graph datasets containing thousands of features. We therefore view the absence of explicit feature interaction modeling as an important tradeoff that enables scalable fully inductive learning across heterogeneous feature spaces.

### 9.2. Limitations and Future Work

Despite its strong transferability, GVT still has several limitations and opportunities for future improvement.

**Predictor training.** Although GVT removes the need for per-dataset representation learning, it still requires training a lightweight predictor on top of the representations. One possible alternative is to use recent tabular foundation models (TFMs) (Hollmann et al., 2022; 2025), which can perform prediction through in-context learning without additional training by directly using GVT representations as input contexts. As shown in Appendix L, TFMs can further improve performance over standard MLP predictors. However, current TFMs typically have quadratic complexity with respect to feature dimensionality, making them impractical for many graph datasets with thousands of features. We therefore consider lightweight MLP predictors a more practical default choice at present, while more scalable TFMs remain a promising direction for future work.

**Recurrent-depth selection.** GVT currently requires selecting the recurrent depth by training $L$ separate MLP predictors. Although this cost is substantially smaller than conventional GNN optimization with extensive hyperparameter tuning, it still introduces additional overhead. Nevertheless, our analysis in Appendix M shows that recurrent depths of 4 or 6 generally perform well and remain competitive with the strongest baseline, UniMP, on average. Future work may further eliminate this overhead through depth-adaptive architectures pretrained across diverse graph datasets.

**Beyond node classification.** This work primarily focuses on node classification. Extending the view-space framework toward more general graph representation learning that supports edge- and graph-level tasks remains an important direction for future work. Incorporating edge features and broader graph families, such as hypergraphs, may further expand the applicability of view-space learning.

### 10. Conclusion

In this work, we introduce a graph-centric approach to address feature heterogeneity across datasets. We propose the *view space*, an adjacency-induced representational axis that enables arbitrary graphs to be represented and processed in a unified manner. We further introduce a parametric mapping in this space, *Graph View Transformation* (GVT), which enables node-feature dynamic aggregation beyond the standard GNN aggregations. As a practical instantiation for node classification, we propose Recurrent GVT (RGVT)

and empirically demonstrate that view-space knowledge transfers effectively across graphs with diverse feature types, outperforming 12 dataset-specialized GNNs. Overall, the view space provides a unified, graph-centric framework for learning over heterogeneous feature spaces, bridging theoretical generality with strong empirical performance.

## Acknowledgements

This work was partly supported by the National Research Foundation of Korea (NRF) grant funded by the Korea government (MSIT) (RS-2024-00341425 and RS-2024-00406985), "Advanced GPU Utilization Support Program" funded by the Government of the Republic of Korea (Ministry of Science and ICT), and the New Faculty Startup Fund from Seoul National University.

## Impact Statement

In this paper, we aim to advance graph representation learning toward fully inductive applicability across graphs, with the primary goal of scientific research. Our approach is designed to broaden knowledge transfer across graphs, potentially benefiting a wide range of applications. We do not anticipate direct negative societal impacts, but we encourage responsible and ethical use of the work.

## Use of LLMs

LLMs were used to polish the writing of the paper.

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

## A. Wilcoxon Signed Rank Test

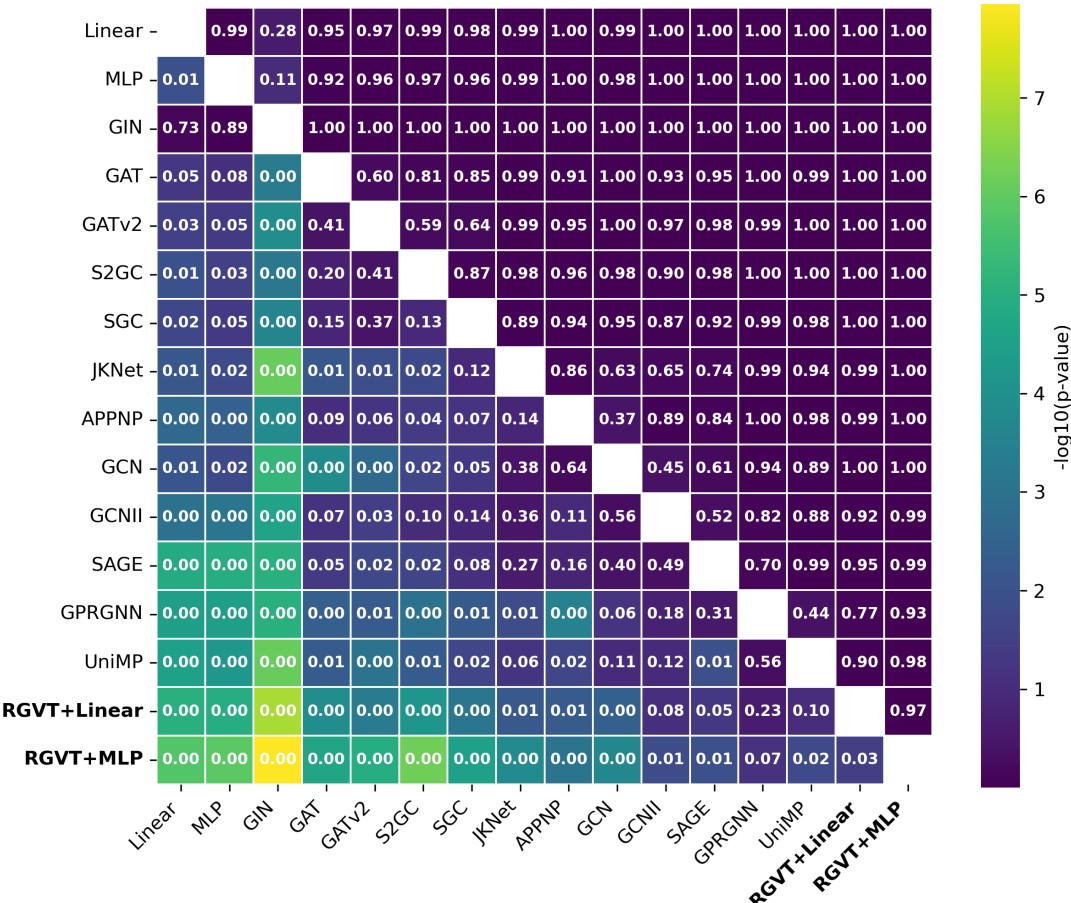

*Figure 4.* Pairwise Wilcoxon signed-rank tests comparing RGVT using either a Linear or MLP predictor, against 12 fully tuned GNNs across 28 datasets. Brighter colors correspond to stronger statistical significance. Remarkably, RGVT with a MLP predictor—despite being pretrained on a single dataset—achieves statistically significant gains over all GNNs at the $p < 0.05$ level, with GPRGNN (Chien et al., 2021) as the sole exception ($p = 0.07$).

## B. Proof of Lemma 3.1

**Lemma B.1** (Permutation equivariance ⇒ node-size independence (root-preserving))**.** *For each $N \in \mathbb{N}$ let $\Psi_N : \mathbb{R}^{N \times F} \times \mathbb{R}^{N \times N} \to \mathbb{R}^{N \times G}$ satisfy*

$$\Psi_N(P\boldsymbol{X}, \ P\boldsymbol{A}P^\top) = P\,\Psi_N(\boldsymbol{X}, \boldsymbol{A}) \quad \text{for all permutation matrices } P \in \{0, 1\}^{N \times N}.$$

*Fix a node $i$. Suppose that whenever two inputs $(\boldsymbol{X}, \boldsymbol{A})$ and $(\boldsymbol{X}', \boldsymbol{A}')$ have identical $i$-rooted neighborhoods as multisets,*

$$\left\{(\boldsymbol{X}_j, \boldsymbol{A}_{ij})\right\}_{j=1}^N \ = \ \left\{(\boldsymbol{X}'_j, \boldsymbol{A}'_{ij})\right\}_{j=1}^N,$$

*there exists a permutation $P$ that bijects the two $i$-centered neighborhoods and fixes the root, i.e., $P(i) = i$. Then there exists a function $f$ on finite multisets, independent of $N$, such that*

$$\left(\Psi_N(\boldsymbol{X}, \boldsymbol{A})\right)_i = f\Big(\boldsymbol{X}_i, \ \left\{(\boldsymbol{X}_j, \boldsymbol{A}_{ij})\right\}_{j=1}^N\Big).$$

*In particular, the rule and its parameters do not depend on the node size $N$.*

*Proof.* Let $(\boldsymbol{X}, \boldsymbol{A})$ and $(\boldsymbol{X}', \boldsymbol{A}')$ be as in the statement. By the assumption, there is a permutation $P$ with $P(i) = i$ that maps the two $i$-rooted neighborhoods bijectively. By equivariance,

$$
\begin{aligned}
\left(\Psi_N(\boldsymbol{X}', \boldsymbol{A}')\right)_i &= \left(\Psi_N(P\boldsymbol{X},\ P\boldsymbol{A}P^\top)\right)_i \\
&= \left(P\,\Psi_N(\boldsymbol{X}, \boldsymbol{A})\right)_i \\
&= \left(\Psi_N(\boldsymbol{X}, \boldsymbol{A})\right)_{P^{-1}(i)} \\
&= \left(\Psi_N(\boldsymbol{X}, \boldsymbol{A})\right)_i,
\end{aligned}
$$

where the last equality uses $P(i) = i$. Hence the $i$-th output depends only on the multiset $\{(\boldsymbol{X}_j, \boldsymbol{A}_{ij})\}_{j=1}^N$, not on labels or $N$. Define $f$ by evaluating $\Psi_N$ on any canonical ordering of that multiset and taking the $i$-th row; $f$ is well-defined and does not involve $N$. $\square$

**Lemma B.2** (Feature permutation equivariance $\Rightarrow$ feature-size independence (root-preserving)). *For each $F \in \mathbb{N}$ let* $\Psi_F : \mathbb{R}^{N \times F} \times \mathbb{R}^{N \times N} \to \mathbb{R}^{N \times F}$ *satisfy*

$$
\Psi_F(\boldsymbol{X}Q,\ \boldsymbol{A}) = \Psi_F(\boldsymbol{X}, \boldsymbol{A})\,Q \quad \text{for all feature permutations } Q \in \{0,1\}^{F \times F}.
$$

*Fix a feature index $f \in [F]$. Suppose that whenever two inputs $(\boldsymbol{X}, \boldsymbol{A})$ and $(\boldsymbol{X}', \boldsymbol{A})$ have identical column multisets $\{\boldsymbol{X}_{:,h}\}_{h=1}^F = \{\boldsymbol{X}'_{:,h}\}_{h=1}^F$, there exists a feature permutation $Q$ that bijects the two multisets and fixes $f$, i.e., $Q(f) = f$.*

*Then there exists a function $g$ on finite multisets, independent of $F$, such that*

$$
\left(\Psi_F(\boldsymbol{X}, \boldsymbol{A})\right)_{:,f} = g\Big(\boldsymbol{X}_{:,f},\ \{\boldsymbol{X}_{:,h}\}_{h=1}^F,\ \boldsymbol{A}\Big).
$$

*In particular, the rule and its parameters do not depend on the feature dimension $F$.*

*Proof.* Let $(\boldsymbol{X}, \boldsymbol{A})$ and $(\boldsymbol{X}', \boldsymbol{A})$ be as in the statement. By assumption, there is $Q$ with $Q(f) = f$ that bijects the two column multisets.

By feature equivariance,

$$
\begin{aligned}
\left(\Psi_F(\boldsymbol{X}', \boldsymbol{A})\right)_{:,f} &= \left(\Psi_F(\boldsymbol{X}Q, \boldsymbol{A})\right)_{:,f} \\
&= \left(\Psi_F(\boldsymbol{X}, \boldsymbol{A})\,Q\right)_{:,f} \\
&= \left(\Psi_F(\boldsymbol{X}, \boldsymbol{A})\right)_{:,Q^{-1}(f)} \\
&= \left(\Psi_F(\boldsymbol{X}, \boldsymbol{A})\right)_{:,f}.
\end{aligned}
$$

where the last step uses $P(i) = i$ and $Q(f) = f$. Thus the $(i, f)$ entry depends only on the unordered $i$-rooted node multiset and on the unordered column multiset, not on labels, $N$, or $F$. Define $h$ by applying $\Psi_{N,F}$ after canonically ordering both multisets and reading the $(i, f)$ entry. $\square$

**Lemma B.3** (Fully Inductive Graph Transformation). *A function $\Psi$ that is equivariant to both node permutations (R1) and feature permutations (R2) is well-defined for graphs of arbitrary size $(N, F)$.*

*Proof.* Let $\Psi_{N,F} : \mathbb{R}^{N \times F} \times \mathbb{R}^{N \times N} \to \mathbb{R}^{N \times G}$ satisfy R1 and R2 for every $(N, F)$.

**Step 1 (node-size independence).** Fix any node index $i \in [N]$. By Lemma B.1, there exists a function $f$ on finite multisets, independent of $N$, such that

$$
\left(\Psi_{N,F}(\boldsymbol{X}, \boldsymbol{A})\right)_i = f\Big(\boldsymbol{X}_i,\ \{(\boldsymbol{X}_j, \boldsymbol{A}_{ij})\}_{j=1}^N\Big).
$$

Hence the $i$-th row of $\Psi_{N,F}(\boldsymbol{X}, \boldsymbol{A})$ depends only on the $i$-rooted neighborhood multiset and not on $N$.

**Step 2 (feature-size independence).**   Fix any feature index $f \in [F]$. By Lemma B.2, there exists a function $g$ on finite multisets, independent of $F$, such that

$$\big(\Psi_{N,F}(\boldsymbol{X}, \boldsymbol{A})\big)_{:,f} = g\Big(\boldsymbol{X}_{:,f}, \ \{\boldsymbol{X}_{:,h}\}_{h=1}^{F}, \ \boldsymbol{A}\Big).$$

Hence the $f$-th output channel depends only on the unordered multiset of feature columns (together with $\boldsymbol{A}$) and not on $F$.

**Step 3 (simultaneous independence of $N$ and $F$).**   Let $(\boldsymbol{X}, \boldsymbol{A})$ and $(\boldsymbol{X}', \boldsymbol{A}')$ be two inputs (possibly with different sizes) and fix any pair $(i, f)$. Assume they have the same $i$-rooted neighborhood multiset and the same column multiset. By the root-preserving hypotheses in Lemmas B.1 and B.2, there exist permutations $P, Q$ with $P(i) = i$ and $Q(f) = f$ that biject these multisets. Using R1 and R2 (dual equivariance),

$$
\begin{aligned}
\big(\Psi_{N',F'}(\boldsymbol{X}', \boldsymbol{A}')\big)_{i,f} &= \big(\Psi_{N,F}(P\boldsymbol{X}Q, \ P\boldsymbol{A}P^\top)\big)_{i,f} \\
&= \big(P\,\Psi_{N,F}(\boldsymbol{X}, \boldsymbol{A})\,Q\big)_{i,f} \\
&= \big(\Psi_{N,F}(\boldsymbol{X}, \boldsymbol{A})\big)_{P^{-1}(i),\,Q^{-1}(f)} \\
&= \big(\Psi_{N,F}(\boldsymbol{X}, \boldsymbol{A})\big)_{i,f}.
\end{aligned}
$$

Therefore, each entry $(i, f)$ depends only on (i) the *unordered* multiset of the $i$-rooted neighborhood and (ii) the *unordered* multiset of feature columns; it is independent of node/feature labels and of the sizes $(N, F)$.

From the three steps above, it follows that the entire output of $\Psi$ is well defined independently of $(N, F)$. In other words, $\Psi$ can be interpreted as a single rule for graphs of arbitrary size—one that assigns identical outputs to identical multisets—thereby the function is fully inductive.

$\square$

*Remark* B.4 (On the root-preserving hypothesis). The root-preserving assumption used in Lemmas B.1 and B.2 (i.e., choosing permutations $P, Q$ with $P(i) = i$ and $Q(f) = f$) is convenient but not necessary. One can remove it by *canonicalization*.

**Node side.**   Fix $i$. Define a deterministic permutation $\kappa_i(\boldsymbol{X}, \boldsymbol{A}) \in \mathfrak{S}_N$ that (a) sends $i$ to index 1 and (b) sorts the $i$-rooted neighborhood multiset $\{(\boldsymbol{X}_j, \boldsymbol{A}_{ij})\}_{j=1}^{N}$ by a fixed total order (e.g., lexicographic in $(\boldsymbol{A}_{ij}, \boldsymbol{X}_j)$ with $j$ as a tie-breaker). Set

$$f\big(\boldsymbol{X}_i, \{(\boldsymbol{X}_j, \boldsymbol{A}_{ij})\}_j\big) := \Big(\Psi_N\big(\kappa_i(\boldsymbol{X}, \boldsymbol{A})\,\boldsymbol{X}, \ \kappa_i(\boldsymbol{X}, \boldsymbol{A})\,\boldsymbol{A}\,\kappa_i(\boldsymbol{X}, \boldsymbol{A})^\top\big)\Big)_1.$$

If two inputs share the same $i$-rooted multiset, then their canonicalized pairs coincide, and by node equivariance,

$$\big(\Psi_N(\boldsymbol{X}, \boldsymbol{A})\big)_i = \Big(\Psi_N\big(\kappa_i\boldsymbol{X}, \ \kappa_i\boldsymbol{A}\kappa_i^\top\big)\Big)_1.$$

Hence $f$ depends only on the multiset and is independent of $N$.

**Feature side.**   Fix $f$. Define a deterministic permutation $\tau_f(\boldsymbol{X}) \in \mathfrak{S}_F$ that (a) sends $f$ to column 1 and (b) sorts the column multiset $\{\boldsymbol{X}_{:,h}\}_{h=1}^{F}$ by a fixed total order (lexicographic in the column vector with $h$ as a tie-breaker). Set

$$g\big(\boldsymbol{X}_{:,f}, \{\boldsymbol{X}_{:,h}\}_h, \boldsymbol{A}\big) := \Big(\Psi_F\big(\boldsymbol{X}\,\tau_f(\boldsymbol{X}), \ \boldsymbol{A}\big)\Big)_{:,1}.$$

If two inputs share the same column multiset, their canonicalized pairs coincide; by feature equivariance (with the chosen output action, e.g. $\rho_F(Q) = Q$ in the strong form),

$$\big(\Psi_F(\boldsymbol{X}, \boldsymbol{A})\big)_{:,f} = \Big(\Psi_F\big(\boldsymbol{X}\,\tau_f(\boldsymbol{X}), \ \boldsymbol{A}\big)\Big)_{:,1}.$$

Thus $g$ depends only on the multiset and is independent of $F$.

Combining the two sides gives the simultaneous independence of both $N$ and $F$ used in Theorem 4.7, without assuming root-preserving permutations.

## C. About Feature Permutation Equivariance

### C.1. Permutation Invariance and Equivariance

In this section, we formalize a fact used in the main text: any feature-permutation–invariant predictor can be realized as an invariant readout of some feature-permutation–equivariant representation.

**Proposition C.1.** *Fix $N \geq 1$ and let $\mathfrak{S}_F$ be the permutation group of $F$ feature indices, acting on node-feature matrices $X \in \mathbb{R}^{N \times F}$ by right-multiplication $X \mapsto XQ$ with permutation matrices $Q \in \{0,1\}^{F \times F}$.*

*Let*

$$f : \mathbb{R}^{N \times F} \times \{0,1\}^{N \times N} \to \mathcal{Y}$$

*be* feature-permutation invariant, *i.e. it satisfies*

$$f(XQ, A) = f(X, A) \qquad \forall Q \in \mathfrak{S}_F, \ \forall (X, A).$$

*Then there exist*

- *a representation space $\mathcal{Z}$,*

- *a feature-permutation–equivariant map $\Phi : \mathbb{R}^{N \times F} \times \{0,1\}^{N \times N} \to \mathcal{Z}$,*

- *and a (not necessarily equivariant) post-map $\psi : \mathcal{Z} \to \mathcal{Y}$,*

*such that*

$$f = \psi \circ \Phi.$$

*In particular, every feature-permutation–invariant mapping can be written as an invariant readout of a feature-permutation–equivariant representation.*

*Proof.* We first introduce a simple equivalence relation on pairs $(X, A)$:

$$(X, A) \sim (X', A') \quad \Longleftrightarrow \quad \exists Q \in \mathfrak{S}_F \text{ such that } X' = XQ \text{ and } A' = A.$$

Thus two graphs are equivalent if they only differ by a permutation of their feature indices. Let $(\mathbb{R}^{N \times F} \times \{0,1\}^{N \times N})/\sim$ denote the set of equivalence classes, and write $[(X, A)]$ for the class containing $(X, A)$.

Define the *quotient map*

$$\pi : \mathbb{R}^{N \times F} \times \{0,1\}^{N \times N} \to (\mathbb{R}^{N \times F} \times \{0,1\}^{N \times N})/\sim, \qquad \pi(X, A) = [(X, A)].$$

By construction we have

$$\pi(XQ, A) = [(XQ, A)] = [(X, A)] = \pi(X, A),$$

so $\pi$ is invariant under feature permutations.

We now equip the quotient space with the trivial action of feature permutations: for every $Q \in \mathfrak{S}_F$ and every class $[X, A]$, we set

$$Q \cdot [(X, A)] := [(X, A)].$$

With this action, $\pi$ becomes feature-permutation *equivariant* in the usual sense:

$$\pi(XQ, A) = [(XQ, A)] = [(X, A)] = Q \cdot [(X, A)] = Q \cdot \pi(X, A).$$

Next, since $f$ is invariant, it is constant on each equivalence class: if $(X', A') \sim (X, A)$, then $X' = XQ$ and $A' = A$ for some $Q$, hence

$$f(X', A') = f(XQ, A) = f(X, A).$$

Therefore there exists a unique function

$$\psi : (\mathbb{R}^{N \times F} \times \{0,1\}^{N \times N})/\sim \to \mathcal{Y}$$

such that

$$\psi([(X,A)]) = f(X,A),$$

and by definition we have $f = \psi \circ \pi$.

Finally, set $\mathcal{Z} := (\mathbb{R}^{N \times F} \times \{0,1\}^{N \times N})/\sim$ and $\Phi := \pi$. Then $\Phi$ is feature-permutation equivariant and $f = \psi \circ \Phi$, which proves the claim. $\qquad\square$

### C.2. Loss of Information

**Lemma C.2** (Permutation Invariance Removes Feature-Wise Structure). *Let*

$$g : \mathbb{R}^{N \times F} \times \{0,1\}^{N \times N} \to \mathcal{Y}$$

*be feature-permutation invariant, i.e.,*

$$g(XQ, A) = g(X, A) \qquad \forall Q \in \mathfrak{S}_F, \ \forall (X, A), \tag{1}$$

*where $\mathfrak{S}_F$ is the permutation group on $F$ feature indices acting by right-multiplication $X \mapsto XQ$.*

*Then $g$ cannot represent any dependency or interaction tied to specific feature coordinates. In particular, for any indices $i \neq j$, let $X^{(i \leftrightarrow j)}$ denote the node-feature matrix obtained from $X$ by swapping its $i$-th and $j$-th feature columns. Then*

$$g(X, A) = g(X^{(i \leftrightarrow j)}, A) \qquad \forall (X, A). \tag{2}$$

*Proof.* Fix indices $i \neq j$ and let $Q_{(ij)} \in \mathfrak{S}_F$ be the permutation matrix that swaps feature indices $i$ and $j$ while leaving all other indices unchanged. For any $(X, A)$, define

$$X^{(i \leftrightarrow j)} := XQ_{(ij)}.$$

By construction, $X^{(i \leftrightarrow j)}$ is exactly $X$ with its $i$-th and $j$-th feature columns exchanged.

Since $g$ is feature-permutation invariant in the sense of equation 1, we have

$$g(X^{(i \leftrightarrow j)}, A) = g(XQ_{(ij)}, A) = g(X, A) \qquad \forall (X, A),$$

which proves equation 2. Thus $g$ assigns identical outputs to inputs that only differ by swapping the same pair of feature channels. $\qquad\square$

## D. Proofs of Section 4

### D.1. Proof of Lemma 4.2

*Proof.* Let $A \in \mathbb{R}^{N \times N}$ be an adjacency matrix with degree matrix $D = \mathrm{diag}(A\mathbf{1})$, and let $P \in \{0,1\}^{N \times N}$ be any permutation matrix. Each adjacency preprocessing operator listed in the lemma is a matrix-valued mapping $\nu : \mathbb{R}^{N \times N} \to \mathbb{R}^{N \times N}$ defined as follows:

1. **Self-augmented:** $\nu(A) = A + I$ (and trivially $\nu(A) = A$).

2. **Degree-normalized:** $\nu(A) := D^{-p} A D^{-q}$ for exponents $p, q \in \mathbb{R}$ (row/symmetric/column cases correspond to $p + q = 1$).

3. **Spectral/Laplacian filters:** $\nu(A) = f(L)$ with $L = D - A$ or $L_{\mathrm{sym}} = I - D^{-1/2} A D^{-1/2}$, for any matrix function $f$ given by a power series convergent on $\mathrm{spec}(L)$ (e.g., $e^{-tL}$).

4. **Diffusion kernels:** $\nu(\boldsymbol{A}) = \alpha \left(\boldsymbol{I} - (1-\alpha)\boldsymbol{D}^{-1}\boldsymbol{A}\right)^{-1}$, with $\alpha \in (0, 1]$.

5. **Polynomial variants:** $\nu(\boldsymbol{A}) = \sum_{k=0}^{K} c_k \, \nu_0(\boldsymbol{A})^k$ for some base operator $\nu_0$ from items 1–4.

We now show that each operator is permutation equivariant. Let $\boldsymbol{A} \in \mathbb{R}^{N \times N}$ with degree matrix $\boldsymbol{D} = \operatorname{diag}(\boldsymbol{A}\mathbf{1})$, and let $\boldsymbol{P}$ be a permutation matrix. For $\boldsymbol{A}' = \boldsymbol{P}\boldsymbol{A}\boldsymbol{P}^\top$ we have $\boldsymbol{D}' = \boldsymbol{P}\boldsymbol{D}\boldsymbol{P}^\top$. We use $\boldsymbol{P}^\top = \boldsymbol{P}^{-1}$, conjugation rules for sums/products, and $(\boldsymbol{P}\boldsymbol{X}\boldsymbol{P}^\top)^{-1} = \boldsymbol{P}\boldsymbol{X}^{-1}\boldsymbol{P}^\top$.

**1. Self-augmented.**
$$\nu(\boldsymbol{P}\boldsymbol{A}\boldsymbol{P}^\top) = \boldsymbol{P}\boldsymbol{A}\boldsymbol{P}^\top + \boldsymbol{I} = \boldsymbol{P}(\boldsymbol{A}+\boldsymbol{I})\boldsymbol{P}^\top = \boldsymbol{P}\nu(\boldsymbol{A})\boldsymbol{P}^\top.$$

**2. Degree-normalized.**

$$\nu(\boldsymbol{P}\boldsymbol{A}\boldsymbol{P}^\top) = (\boldsymbol{P}\boldsymbol{D}\boldsymbol{P}^\top)^{-p}\,(\boldsymbol{P}\boldsymbol{A}\boldsymbol{P}^\top)\,(\boldsymbol{P}\boldsymbol{D}\boldsymbol{P}^\top)^{-q} = \boldsymbol{P}\,\boldsymbol{D}^{-p}\,\boldsymbol{A}\,\boldsymbol{D}^{-q}\,\boldsymbol{P}^\top = \boldsymbol{P}\,\nu(\boldsymbol{A})\,\boldsymbol{P}^\top.$$

**3. Spectral/Laplacian filters.**
$$\boldsymbol{P}\boldsymbol{L}\boldsymbol{P}^\top = \boldsymbol{P}\boldsymbol{D}\boldsymbol{P}^\top - \boldsymbol{P}\boldsymbol{A}\boldsymbol{P}^\top = \boldsymbol{D}' - \boldsymbol{A}' = \boldsymbol{L}',$$

and for $f(z) = \sum_{k \geq 0} a_k z^k$,

$$f(\boldsymbol{P}\boldsymbol{L}\boldsymbol{P}^\top) = \sum_{k \geq 0} a_k (\boldsymbol{P}\boldsymbol{L}\boldsymbol{P}^\top)^k = \boldsymbol{P}\Big(\sum_{k \geq 0} a_k \boldsymbol{L}^k\Big)\boldsymbol{P}^\top = \boldsymbol{P}f(\boldsymbol{L})\boldsymbol{P}^\top.$$

Same for $\boldsymbol{L}_{\text{sym}}$.

**4. Diffusion kernels.** Let $\boldsymbol{B} = \boldsymbol{D}^{-1}\boldsymbol{A}$, so $\boldsymbol{B}' = (\boldsymbol{D}')^{-1}\boldsymbol{A}' = \boldsymbol{P}\boldsymbol{B}\boldsymbol{P}^\top$. Then
$$\nu(\boldsymbol{P}\boldsymbol{A}\boldsymbol{P}^\top) = \alpha(\boldsymbol{I} - (1-\alpha)\boldsymbol{B}')^{-1} = \alpha\,\boldsymbol{P}(\boldsymbol{I} - (1-\alpha)\boldsymbol{B})^{-1}\boldsymbol{P}^\top = \boldsymbol{P}\nu(\boldsymbol{A})\boldsymbol{P}^\top.$$

**5. Polynomial variants.** If $\nu_0(\boldsymbol{P}\boldsymbol{A}\boldsymbol{P}^\top) = \boldsymbol{P}\nu_0(\boldsymbol{A})\boldsymbol{P}^\top$, then

$$p(\nu_0(\boldsymbol{P}\boldsymbol{A}\boldsymbol{P}^\top)) = \sum_{k=0}^{K} c_k (\nu_0(\boldsymbol{P}\boldsymbol{A}\boldsymbol{P}^\top))^k = \sum_{k=0}^{K} c_k \, \boldsymbol{P}\nu_0(\boldsymbol{A})^k \boldsymbol{P}^\top = \boldsymbol{P}p(\nu_0(\boldsymbol{A}))\boldsymbol{P}^\top.$$

Thus, all listed preprocessing operators satisfy $\nu(\boldsymbol{P}\boldsymbol{A}\boldsymbol{P}^\top) = \boldsymbol{P}\nu(\boldsymbol{A})\boldsymbol{P}^\top$ and are permutation equivariant. $\qquad\square$

### D.2. Proof of Theorem 4.7.

**Theorem D.1** (Fully Inductive GVT). *GVT is equivariant to both node permutations (R1) and feature permutations (R2) required for fully inductive node representation learning (FI-NRL).*

*Proof.* **Setup.** Let $\boldsymbol{X} \in \mathbb{R}^{N \times F}$, $\boldsymbol{A} \in \mathbb{R}^{N \times N}$, and view finders $\{\nu_c\}_{c=1}^{C}$. Define view stacking by

$$\mathbf{X} \;=\; \mathcal{V}(\boldsymbol{X}, \boldsymbol{A} \mid \{\nu_c\}) \quad \text{with} \quad \mathbf{X}_{:,:,c} = \nu_c(\boldsymbol{A})\,\boldsymbol{X} \in \mathbb{R}^{N \times F}.$$

Let the dimension-collapsing map be $\phi : \mathbb{R}^C \to \mathbb{R}$ with shared parameters $\theta$, applied independently at each $(n, f)$:

$$\boldsymbol{Z}_{n,f} \;=\; \phi\big(\mathbf{X}_{n,f,:} \mid \theta\big), \qquad \Psi(\boldsymbol{X}, \boldsymbol{A}) = \boldsymbol{Z}.$$

Each $\nu_c$ is node-permutation equivariant by the definition of view finder: for any permutation matrix $\boldsymbol{P} \in \{0, 1\}^{N \times N}$, it holds that

$$\nu_c(\boldsymbol{P}\boldsymbol{A}\boldsymbol{P}^\top) \;=\; \boldsymbol{P}\,\nu_c(\boldsymbol{A})\,\boldsymbol{P}^\top. \qquad\qquad (\star)$$

**R1: Node-permutation equivariance.** Let $\boldsymbol{P}$ be any node permutation. Consider inputs $(\boldsymbol{P}\boldsymbol{X}, \boldsymbol{P}\boldsymbol{A}\boldsymbol{P}^\top)$. For each $c$,

$$\mathbf{X}'_{:,:,c} = \nu_c(\boldsymbol{P}\boldsymbol{A}\boldsymbol{P}^\top)(\boldsymbol{P}\boldsymbol{X}) \overset{(\star)}{=} \boldsymbol{P}\,\nu_c(\boldsymbol{A})\,\boldsymbol{P}^\top\boldsymbol{P}\boldsymbol{X} = \boldsymbol{P}\big(\nu_c(\boldsymbol{A})\boldsymbol{X}\big) = \boldsymbol{P}\,\mathbf{X}_{:,:,c}.$$

Hence the stacked tensor transforms as $\mathbf{X}' = \boldsymbol{P}\,\mathbf{X}$ (mode-1 action). Therefore, for every $(n, f)$,

$$\mathbf{X}'_{n,f,:} = \mathbf{X}_{\pi(n),f,:} \quad \text{where } \pi \text{ is the permutation represented by } \boldsymbol{P}.$$

Since $\phi$ is applied identically and independently to each $(n, f)$ (shared $\theta$, no cross-index coupling),

$$\boldsymbol{Z}'_{n,f} = \phi(\mathbf{X}'_{n,f,:} \mid \theta) = \phi(\mathbf{X}_{\pi(n),f,:} \mid \theta) = \boldsymbol{Z}_{\pi(n),f},$$

i.e., $\boldsymbol{Z}' = \boldsymbol{P}\,\boldsymbol{Z}$. Thus

$$\Psi(\boldsymbol{PX}, \boldsymbol{PAP}^\top) = \boldsymbol{P}\,\Psi(\boldsymbol{X}, \boldsymbol{A}),$$

which proves **R1**.

**R2: Feature-permutation equivariance.** Let $\boldsymbol{Q} \in \{0,1\}^{F \times F}$ be a feature permutation. With inputs $(\boldsymbol{XQ}, \boldsymbol{A})$, for each $c$,

$$\mathbf{X}''_{:,:,c} = \nu_c(\boldsymbol{A})\,(\boldsymbol{XQ}) = \big(\nu_c(\boldsymbol{A})\boldsymbol{X}\big)\boldsymbol{Q} = \mathbf{X}_{:,:,c}\,\boldsymbol{Q}.$$

Hence $\mathbf{X}'' = \mathbf{X}\,\boldsymbol{Q}$ (mode-2 action), so for every $(n, f)$,

$$\mathbf{X}''_{n,f,:} = \mathbf{X}_{n,\sigma(f),:} \quad \text{where } \sigma \text{ is the permutation represented by } \boldsymbol{Q}.$$

Applying the same pointwise/shared $\phi$,

$$\boldsymbol{Z}''_{n,f} = \phi(\mathbf{X}''_{n,f,:} \mid \theta) = \phi(\mathbf{X}_{n,\sigma(f),:} \mid \theta) = \boldsymbol{Z}_{n,\sigma(f)},$$

i.e., $\boldsymbol{Z}'' = \boldsymbol{Z}\,\boldsymbol{Q}$. Therefore

$$\Psi(\boldsymbol{XQ}, \boldsymbol{A}) = \Psi(\boldsymbol{X}, \boldsymbol{A})\,\boldsymbol{Q},$$

which proves **R2**.

**Conclusion.** Both **R1** and **R2** hold; hence the GVT layer is equivariant to node and feature permutations. $\qquad\square$

# E. Proofs of Section 5

### E.1. Proof of Lemma 5.1.

*Proof.* With view finders including $\boldsymbol{I}$ and powers of the normalized adjacencies $\{\hat{A}^k_{\mathrm{SYM}}, \hat{A}^k_{\mathrm{RW}}\}$ where $\hat{A}_{\mathrm{RW}} = \boldsymbol{D}^{-1}\boldsymbol{A}$ and $\hat{A}_{\mathrm{SYM}} = \boldsymbol{D}^{-\frac{1}{2}}\boldsymbol{A}\boldsymbol{D}^{-\frac{1}{2}}$ with $\boldsymbol{D}$ denoting the degree matrix, a linear GVT layer outputs

$$\boldsymbol{Z} = \sum_c g_c\,\nu_c(\boldsymbol{A})\,\boldsymbol{X} = p(\boldsymbol{A})\,\boldsymbol{X},$$

which matches many static aggregation filters $p(\boldsymbol{A})$ through suitable coefficients $\{g_c\}$. Representative choices of $\{g_c\}$ for well-known GNNs are summarized in Table 4.

Thus, each listed aggregation operator $p(\boldsymbol{A})$ is realized by a linear GVT. $\qquad\square$

### E.2. Proof of Theorem 5.2.

*Proof.* Assume $\nabla\phi$ is $M$-Lipschitz on a convex neighborhood of $v_0$ (equivalently, $\|\nabla^2\phi(w)\| \le M$ for all $w$ on the segment $[v_0, v]$).

Fix $\boldsymbol{v}_0 \in \mathbb{R}^C$ and let $\boldsymbol{h} = \boldsymbol{v} - \boldsymbol{v}_0$. Consider the scalar function

$$\psi(t) = \phi(\boldsymbol{v}_0 + t\boldsymbol{h}), \qquad t \in [0, 1].$$

By the chain rule, $\psi'(t) = \nabla\phi(\boldsymbol{v}_0 + t\boldsymbol{h})^\top\boldsymbol{h}$ and $\psi''(t) = \boldsymbol{h}^\top\nabla^2\phi(\boldsymbol{v}_0 + t\boldsymbol{h})\,\boldsymbol{h}$ whenever the derivatives exist. By the fundamental theorem of calculus,

$$\phi(\boldsymbol{v}) - \phi(\boldsymbol{v}_0) = \psi(1) - \psi(0) = \int_0^1 \psi'(t)\,dt = \int_0^1 \nabla\phi(\boldsymbol{v}_0 + t\boldsymbol{h})^\top\boldsymbol{h}\,dt.$$

*Table 4.* Static GNN aggregations $p(\boldsymbol{A})$ reproduced by linear GVT through suitable coefficients $g_c$.

| Model | Aggregation $p(\boldsymbol{A})$ | Linear GVT coefficients |
|---|---|---|
| GCN (Kipf & Welling, 2017) | $\hat{\boldsymbol{A}}_{\mathrm{SYM}}$ | $g = 1$ on $\hat{\boldsymbol{A}}_{\mathrm{SYM}}$ |
| SAGE-mean (Hamilton et al., 2017) | $\boldsymbol{I}, \hat{\boldsymbol{A}}_{\mathrm{RW}}$ | $g = 1$ on $\{\boldsymbol{I}, \hat{\boldsymbol{A}}_{\mathrm{RW}}\}$ |
| SGC (Wu et al., 2019) | $\hat{\boldsymbol{A}}_{\mathrm{SYM}}^{K}$ | $g = 1$ on $\hat{\boldsymbol{A}}_{\mathrm{SYM}}^{K}$ |
| APPNP (Gasteiger et al., 2019) | $(1-\alpha)\sum_{k=0}^{K}\alpha^{k}\,\hat{\boldsymbol{A}}_{\mathrm{SYM}}^{k}$ | $g_k = (1-\alpha)\alpha^k$ for $\hat{\boldsymbol{A}}_{\mathrm{SYM}}^{k}$ |
| S$^2$GC (Zhu & Koniusz, 2021) | $\frac{1}{K}\sum_{k=1}^{K}\big((1-\alpha)\hat{\boldsymbol{A}}_{\mathrm{SYM}}^{k}+\alpha\boldsymbol{I}\big)$ | $g_k = \frac{1-\alpha}{K}$ for $\hat{\boldsymbol{A}}_{\mathrm{SYM}}^{k}$, $g = 1$ on $\boldsymbol{I}$ |
| GCNII (Chen et al., 2020) | $(1-\alpha)\hat{\boldsymbol{A}}_{\mathrm{SYM}}, \alpha\boldsymbol{I}$ | $g = 1-\alpha$ on $\hat{\boldsymbol{A}}_{\mathrm{SYM}}$, $g = \alpha$ on $\boldsymbol{I}$ |
| GPRGNN (Chien et al., 2021) | $\sum_{k=0}^{K}\gamma_k\,\hat{\boldsymbol{A}}_{\mathrm{SYM}}^{k}$ | $g_k = \gamma_k$ for $\hat{\boldsymbol{A}}_{\mathrm{SYM}}^{k}$ |

Add and subtract $\nabla\phi(\boldsymbol{v}_0)^{\top}\boldsymbol{h}$ inside the integral to obtain

$$\phi(\boldsymbol{v}) = \phi(\boldsymbol{v}_0) + \nabla\phi(\boldsymbol{v}_0)^{\top}\boldsymbol{h} + \int_{0}^{1}\big(\nabla\phi(\boldsymbol{v}_0+t\boldsymbol{h}) - \nabla\phi(\boldsymbol{v}_0)\big)^{\top}\boldsymbol{h}\,dt.$$

Define the remainder

$$R_2(\boldsymbol{v};\boldsymbol{v}_0) \;=\; \int_{0}^{1}\big(\nabla\phi(\boldsymbol{v}_0+t\boldsymbol{h}) - \nabla\phi(\boldsymbol{v}_0)\big)^{\top}\boldsymbol{h}\,dt.$$

Using the mean value (integral) form with the Hessian and the bound on its operator norm,

$$\big\|\nabla\phi(\boldsymbol{v}_0+t\boldsymbol{h}) - \nabla\phi(\boldsymbol{v}_0)\big\| \;\leq\; \int_{0}^{t}\big\|\nabla^2\phi(\boldsymbol{v}_0+s\boldsymbol{h})\big\|\,\|\boldsymbol{h}\|\,ds \;\leq\; Mt\,\|\boldsymbol{h}\|,$$

for all $t \in [0,1]$ (here $\|\cdot\|$ is the Euclidean norm and $\|\nabla^2\phi\|$ its induced operator norm). Therefore

$$|R_2(\boldsymbol{v};\boldsymbol{v}_0)| \;\leq\; \int_{0}^{1} Mt\,\|\boldsymbol{h}\|^2\,dt \;=\; \frac{M}{2}\,\|\boldsymbol{h}\|^2 \;=\; \frac{M}{2}\,\|\boldsymbol{v}-\boldsymbol{v}_0\|^2.$$

Finally, set $g(\boldsymbol{v}_0) = \nabla\phi(\boldsymbol{v}_0)$ and $b(\boldsymbol{v}_0) = \phi(\boldsymbol{v}_0) - g(\boldsymbol{v}_0)^{\top}\boldsymbol{v}_0$ to write

$$\phi(\boldsymbol{v}) = g(\boldsymbol{v}_0)^{\top}\boldsymbol{v} + b(\boldsymbol{v}_0) + R_2(\boldsymbol{v};\boldsymbol{v}_0),$$

with the stated quadratic bound on $R_2$. This shows $\phi$ is locally affine with quadratic error decay near $\boldsymbol{v}_0$. $\quad\square$

# F. Non-attributed Graphs

In this section, we present additional experiments on non-attributed graphs. While three non-attributed datasets used in the main experiment (AirBrazil, AirEU, AirUSA) adopt one-hot vectors as node features, since the dimensionality of one-hot features scales with graph size, we further investigate alternative feature constructions to identify scalable options for RGVT.

**Datasets** We evaluate performance on six graphs: three airport traffic networks (AirBrazil, AirEU, AirUSA), which originally have no node features, and three citation networks (Cora, Citeseer, Pubmed), for which we explicitly remove the provided features. Inspired by prior work (Cui et al., 2022), we construct three types of artificial node features: (1) one-hot vectors based on node indices, (2) random Gaussian vectors, and (3) DeepWalk embeddings (Grover & Leskovec, 2016). For DeepWalk, we adopt the hyperparameter settings from (Cui et al., 2022) consistently across datasets. For both the random and DeepWalk features, the feature dimension is fixed at 128.

**Experiment Setting** The experimental setup follows the main experiment described in Appendix J. GCN is tuned and trained separately for each dataset and feature type, allowing it to specialize its optimization to each configuration. In

*Table 5.* Performance (%) comparison between GCN, GraphAny, and RGVT on six non-attributed graphs. 1st and 2nd best results for each dataset are highlighted. RGVT with DeepWalk embeddings achieves the best performance on 5 out of 6 datasets.

| | AirBrazil | | | AirEU | | | AirUSA | | |
|---|---|---|---|---|---|---|---|---|---|
| **Feature** | GCN | GraphAny | **RGVT** | GCN | GraphAny | **RGVT** | GCN | GraphAny | **RGVT** |
| One-hot | $63.08_{\pm3.44}$ | $42.31_{\pm0.00}$ | $66.15_{\pm10.67}$ | $41.50_{\pm0.84}$ | $43.12_{\pm0.76}$ | $47.00_{\pm2.70}$ | $53.30_{\pm1.18}$ | $46.31_{\pm0.49}$ | $56.79_{\pm2.26}$ |
| Random | $39.23_{\pm5.70}$ | $37.69_{\pm11.67}$ | $62.31_{\pm7.05}$ | $41.38_{\pm5.42}$ | $36.56_{\pm2.37}$ | $47.25_{\pm3.98}$ | $49.12_{\pm3.20}$ | $38.16_{\pm1.71}$ | $54.09_{\pm1.66}$ |
| DeepWalk | $36.15_{\pm2.11}$ | $27.14_{\pm0.48}$ | $72.31_{\pm4.49}$ | $43.50_{\pm2.88}$ | $25.03_{\pm0.07}$ | $48.00_{\pm2.22}$ | $57.12_{\pm1.29}$ | $38.11_{\pm1.98}$ | $58.27_{\pm1.10}$ |

| | Cora | | | Citeseer | | | Pubmed | | |
|---|---|---|---|---|---|---|---|---|---|
| **Feature** | GCN | GraphAny | **RGVT** | GCN | GraphAny | **RGVT** | GCN | GraphAny | **RGVT** |
| One-hot | $74.24_{\pm2.29}$ | $65.40_{\pm0.00}$ | $71.80_{\pm1.07}$ | $52.04_{\pm0.85}$ | $39.80_{\pm0.00}$ | $46.04_{\pm3.80}$ | $70.46_{\pm0.65}$ | $61.30_{\pm0.00}$ | $49.14_{\pm6.15}$ |
| Random | $56.16_{\pm1.51}$ | $22.18_{\pm1.11}$ | $51.76_{\pm2.98}$ | $36.48_{\pm1.64}$ | $19.34_{\pm1.47}$ | $33.90_{\pm1.94}$ | $47.52_{\pm2.10}$ | $35.88_{\pm1.72}$ | $41.42_{\pm2.84}$ |
| DeepWalk | $76.36_{\pm0.81}$ | $53.44_{\pm1.94}$ | $74.96_{\pm0.94}$ | $52.36_{\pm2.53}$ | $27.98_{\pm1.79}$ | $53.20_{\pm0.67}$ | $74.20_{\pm1.73}$ | $64.12_{\pm1.90}$ | $74.80_{\pm0.83}$ |

*Table 6.* Performance (%) comparison with the MLP predictor and GraphAny across graph datasets with LLM-embedded features. **1st** and 2nd best results are highlighted. $\Delta$ reports relative improvements (%) over the baselines. RGVT shows consistent and significant gains across all datasets.

| | Cora | Citeseer | Pubmed | History | Children | Sportsfit | WikiCS |
|---|---|---|---|---|---|---|---|
| MLP | $78.54_{\pm1.24}$ | $72.73_{\pm0.95}$ | $84.75_{\pm1.01}$ | $71.64_{\pm3.76}$ | $31.73_{\pm1.24}$ | $75.14_{\pm2.10}$ | $74.69_{\pm0.85}$ |
| GraphAny | $79.27_{\pm0.63}$ | $74.15_{\pm0.35}$ | $78.69_{\pm0.04}$ | $45.87_{\pm0.13}$ | $25.77_{\pm0.16}$ | $79.27_{\pm0.02}$ | $71.01_{\pm0.24}$ |
| **RGVT + MLP** | $84.40_{\pm1.27}$ | $75.22_{\pm1.66}$ | $85.05_{\pm0.86}$ | $75.86_{\pm1.44}$ | $33.74_{\pm0.48}$ | $82.82_{\pm2.23}$ | $78.21_{\pm0.39}$ |
| $\Delta$ MLP | ↑ **7.46**% | ↑ **3.42**% | ↑ **0.35**% | ↑ **5.89**% | ↑ **6.34**% | ↑ **10.23**% | ↑ **4.71**% |
| $\Delta$ GraphAny | ↑ **6.48**% | ↑ **1.44**% | ↑ **8.08**% | ↑ **65.38**% | ↑ **30.93**% | ↑ **4.49**% | ↑ **10.15**% |

contrast, GraphAny and RGVT use the pretrained weights from the main experiment (trained on OGBN-Arxiv) and are evaluated in a fully inductive manner. For RGVT, we use an MLP predictor in all experiments. All experiments are repeated five times with different random seeds, and we report the mean and standard deviation.

**Experiment Results** RGVT consistently outperforms GraphAny in 17 of the 18 dataset–feature configurations, demonstrating strong generalization to unseen, artificially constructed feature spaces. When equipped with DeepWalk embeddings, RGVT shows the highest accuracy on 5 of the 6 datasets and ranks second on the remaining one. These results highlight the importance of feature design for non-attributed graphs and show that DeepWalk embeddings, when combined with RGVT, yield strong and reliable performance.

## G. Graphs with LLM Embedded Features

In this section, we conduct additional experiments on text-attributed graphs using LLM-embedded node features. The goal is to evaluate whether fully inductive node representation learning via RGVT remains effective in text-attributed graphs, where node features are highly informative due to text embeddings produced by LLMs.

**Datasets** We adopt seven text-attributed graphs introduced in recent work (Wang et al., 2025), which investigates generalization on text-attributed graphs. Each node feature is generated using the `text-embedding-3-large` model from OpenAI, resulting in a 3704 dimensional embedding. The embedded text corresponds to the opening passages of academic papers (Cora, Citeseer, Pubmed), book descriptions or titles (History, Children), product titles in sports and fitness (SportsFit), and Wikipedia entries or article content (WikiCS). Dataset statistics are summarized in Table 8. For each dataset, we sample 20 nodes per class for training, 500 nodes for validation, and use the remaining nodes for testing. All experiments are repeated five times with different data splits.

**Experiment Setting** The experimental setup follows the main experiment described in Appendix J. We select GCN, GPRGNN, and UniMP—three models that perform strongly in our main experiments—along with MLP and GraphAny as baselines. While the GNNs baselines are tuned and trained separately for each dataset, GraphAny and RGVT use the pretrained weights from the main experiment (trained on OGBN-Arxiv) and are applied directly to these datasets in a fully inductive manner. For RGVT, we use the MLP as the predictor in all experiments.

*Table 7.* Performance (%) comparison with dataset-specialized GCN, GPRGNN, and UniMP on graph datasets with LLM-embedded features. Best results are highlighted in **bold**. RGVT achieves the highest accuracy on two datasets and remains competitive on the others.

| | Cora | Citeseer | Pubmed | History | Children | Sportsfit | WikiCS |
|---|---|---|---|---|---|---|---|
| GCN | $\mathbf{84.44_{\pm 0.52}}$ | $75.65_{\pm 0.99}$ | $83.46_{\pm 0.33}$ | $72.58_{\pm 4.88}$ | $33.85_{\pm 1.84}$ | $82.23_{\pm 2.47}$ | $78.70_{\pm 0.91}$ |
| GPRGNN | $84.24_{\pm 1.15}$ | $\mathbf{76.03_{\pm 1.24}}$ | $84.59_{\pm 1.58}$ | $73.38_{\pm 4.70}$ | $35.70_{\pm 0.82}$ | $81.92_{\pm 3.00}$ | $\mathbf{79.83_{\pm 1.00}}$ |
| UniMP | $83.97_{\pm 0.81}$ | $75.71_{\pm 1.15}$ | $84.71_{\pm 1.27}$ | $75.05_{\pm 2.11}$ | $\mathbf{36.23_{\pm 3.17}}$ | $\mathbf{83.12_{\pm 1.71}}$ | $78.51_{\pm 0.66}$ |
| **RGVT** + MLP | $84.40_{\pm 1.27}$ | $75.22_{\pm 1.66}$ | $\mathbf{85.05_{\pm 0.86}}$ | $\mathbf{75.86_{\pm 1.44}}$ | $33.74_{\pm 0.48}$ | $82.82_{\pm 2.23}$ | $78.21_{\pm 0.39}$ |

*Table 8.* Statistics of the seven graphs with features embedded by the LLM.

| | Cora | Citeseer | Pubmed | History | Children | Sportsfit | WikiCS |
|---|---|---|---|---|---|---|---|
| #Nodes | 2708 | 3186 | 19717 | 41551 | 76875 | 173055 | 11701 |
| #Edges | 10556 | 8450 | 88648 | 503180 | 2325044 | 3020134 | 431726 |
| #Classes | 7 | 6 | 3 | 12 | 24 | 13 | 10 |
| Homophily Ratio | 0.8100 | 0.7841 | 0.8024 | 0.6398 | 0.4043 | 0.8980 | 0.6543 |
| Avg. Degree | 3.90 | 2.65 | 4.50 | 12.11 | 30.24 | 17.45 | 36.85 |

**Experiment Results** Unlike in the main experiment, GraphAny fails to outperform the MLP. This indicates that the linear predictor in GraphAny cannot effectively exploit the rich semantic information provided by LLM-embedded features. The degradation is more severe on the History, Children, and WikiCS datasets, which exhibit heterophilous graph structure. These results further indicate weak generalization to unseen graph structures, caused by the static aggregation of GraphAny.

In contrast, Table 6 shows that RGVT consistently outperforms both baselines, yielding average gains of +5.48% over MLP and +18.14% over GraphAny. As shown in Table 7, RGVT also performs competitively against dataset-specialized GNNs, achieving the best accuracy on two datasets and remaining close to the top-performing model on the others. These results indicate that fully inductive view-space knowledge remains effective even on graphs with rich semantic feature spaces. This effectiveness stems from RGVT's feature-permutation-equivariant design, which allows that feature-wise structure is preserved throughout transformation. As a result, the per-dataset MLP predictor can effectively leverage this information through the representations produced by RGVT. Together, these results highlight GVT as an effective and robust solution for fully inductive node representation learning, without being constrained by the richness of the feature space.

# H. Complexity Analysis

In this section, we analyze both the theoretical and empirical complexity of RGVT, and discuss the practical efficiency benefits enabled by the fully inductive representation learning (FI-NRL) framework.

**Theoretical Analysis** RGVT consists of a sequence of view-stacking operations followed by MLPs that map each view tensor to scalar. Since view stacking computes and concatenates $C$ propagated node-feature matrices, its cost is $C$ feature-propagation steps, i.e., $\mathcal{O}(C|E|F)$. The subsequent MLP transforms every node–feature pair, treating each view vector of dimension $C$ as a single input. Because the transformation operates on $NF$ node–feature pairs with input/hidden dimension $C$, this step incurs $\mathcal{O}(NFC^2)$ complexity. Thus, the total complexity of RGVT with recurrent depth $L$ is

$$\mathcal{O}\big(LC|E|F \ + \ LNFC^2\big),$$

which scales linearly with both the graph size and the feature dimension. We provides the summarizes result at Table 9 with complexity of GNN and GraphAny.

**Empirical Measurements** We report the computational costs of RGVT with MLP predictor across three stages: (1) Pre-training: training on OGBN-Arxiv, (2) Adaptation: training an MLP predictor on the $L$ representations produced at each depth across 27 datasets, and (3) Inference: inference RGVT to produce representations and performing MLP predictors' inference for all 28 datasets. For comparison, we also measure the corresponding computational costs for GCN, GAT, and GraphAny. All experiments are conducted on an NVIDIA RTX A6000 GPU.

As shown in Table 10, RGVT exhibits higher training and inference time than the GNN baselines due to three main reasons. First, RGVT uses a deeper architecture: with a recurrent depth of 8 in our configuration, its effective depth is roughly 4×

*Table 9.* Theoretical complexity of MLP, Feature Propagation, GCN, GraphAny, and RGVT, with scaling behavior for graph size and feature dimensionality. All methods scale linearly with graph size ($N, |E|$) and feature dimensionality ($F$). $L$ is the number of layers, $H$ is the hidden size, and $C$ is the number of linear GNNs (GraphAny) or view finders (RGVT).

| Method | End-to-end Complexity | Graph-Scale | Feature-Scale |
|---|---|---|---|
| MLP | $\mathcal{O}(NFH + (L-1)NH^2)$ | $\mathcal{O}(1)$ | $\mathcal{O}(F)$ |
| Feature Propagation | $\mathcal{O}(|E|F)$ | $\mathcal{O}(|E|)$ | $\mathcal{O}(F)$ |
| GCN | $\mathcal{O}(L|E|H) + \mathcal{O}(NFH + (L-1)NH^2)$ | $\mathcal{O}(|E|)$ | $\mathcal{O}(F)$ |
| GraphAny($^*$) | $\mathcal{O}(LC|E|F) + \mathcal{O}(NFH) + \mathcal{O}(NC^3)$ | $\mathcal{O}(|E|)$ | $\mathcal{O}(F)$ |
| RGVT | $\mathcal{O}(LC|E|F) + \mathcal{O}(LNFC^2)$ | $\mathcal{O}(|E|)$ | $\mathcal{O}(F)$ |

($^*$) GraphAny's pseudo-inverse step can scale quadratically with the feature dimension, up to $\mathcal{O}(F^2)$.
However, we omitted here, as it can be effectively reduced through approximate solvers.

*Table 10.* Time (s) for training on OGBN-Arxiv, adaptation across 27 datasets, and inference over all 28 datasets for RGVT, GraphAny, GCN, and GAT. Adaptation times for GNNs include full hyperparameter search under the search space used in the main experiment (see Appendix J), while values in parentheses denote training-only times for a 2-layer architecture.

| Method | Training (1) | Adaptation (27) | Inference (28) | Adaptation Tasks |
|---|---|---|---|---|
| GCN | 96.83 | 2day 10hr (55.28) | 0.002 | Hyperparameter search and training |
| GAT | 158.59 | 3day 23hr (91.30) | 0.003 | Hyperparameter search and training |
| GraphAny | 598.10 | 55.44 | 0.571 | Computes $C$ linear GNNs |
| **RGVT** + MLP | 607.31 | 151.24 | 0.052 | Trains $L$ MLP predictors |

that of 2-layer GNNs. Second, view stacking requires $C$ aggregations per layer, whereas standard GNNs perform only a single aggregation. Third, the dense transformation in GVT incurs $\mathcal{O}(NFC^2)$ complexity, compared to $\mathcal{O}(NH^2)$ in typical GNNs. Under our settings, when the feature dimension satisfies $F > 128^2/5^2 \approx 655$, the dense multiplication becomes slower by approximately a factor of $F/655$. Despite these factors, RGVT remains practical: its pre-training time on OGBN-Arxiv is approximately 10 minutes, and inference across all 28 graphs requires only 0.052 second.

**Practical Advantages** RGVT offers two significant complexity benefits. First, its fully inductive nature enables direct operation on unseen datasets without any retraining or fine-tuning. Only a lightweight predictor needs to be trained $L$ times to select an appropriate recurrent depth. In contrast, GNNs require full retraining and extensive per-dataset hyperparameter tuning to achieve competitive performance, and often demanding substantial domain expertise. Second, once RGVT produces representations, they can be cached and reused. Second, once RGVT produces node representations, they can be cached and reused across tasks. By comparison, standard GNNs must be retrained, and GraphAny must recompute its linear GNN components whenever the label space changes. Together, these properties yield substantial efficiency benefits, mirroring the advantages of pretrained encoders in NLP and CV.

# I. Guidelines for Selecting View Finder Sets

In this section, we analyze how different view finder choices affect RGVT's performance. We evaluate several graph convolution filters, varying hop sizes, and their combinations, and we provide practical guidance on how to construct effective candidate view finder sets.

**Experiment Configurations** We evaluate three families of view finders, each defined as a matrix-valued function of the adjacency matrix. The random-walk filter (RW) (Hamilton et al., 2017) and the symmetric normalized filter (SYM) (Kipf & Welling, 2017) are expressed as

$$\nu_{\text{rw}}^{(k)}(\boldsymbol{A}) = (\boldsymbol{D}^{-1}\boldsymbol{A})^k, \qquad \nu_{\text{sym}}^{(k)}(\boldsymbol{A}) = (\boldsymbol{D}^{-1/2}\boldsymbol{A}\boldsymbol{D}^{-1/2})^k,$$

where k denotes the number of propagation hops. We also include the Chebyshev polynomial basis (Cheb) (Defferrard et al., 2016), defined as

$$\nu_{\text{cheb}}^{(k)}(\boldsymbol{A}) = T_k(\tilde{\boldsymbol{L}}), \qquad \tilde{\boldsymbol{L}} = \frac{2\boldsymbol{L}}{\lambda_{\max}} - \boldsymbol{I}, \qquad \boldsymbol{L} = \boldsymbol{I} - \boldsymbol{D}^{-1/2}\boldsymbol{A}\boldsymbol{D}^{-1/2},$$

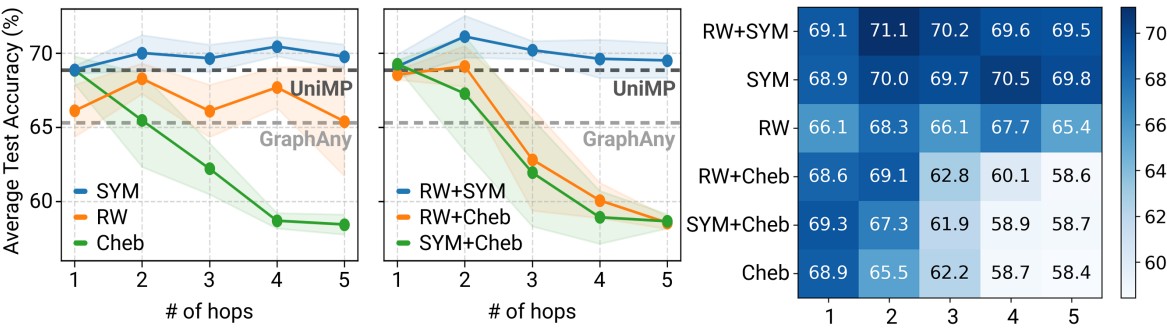

*Figure 5.* Average test accuracy (%) across 28 datasets using different view-finder sets for RGVT, shown in the plot (left) and heatmap (right). Symmetric normalization (SYM) and random-walk (RW) filters exhibit robust performance compared to the Chebyshev polynomial basis (Cheb), and their combination (RW+SYM) with moderate multi-hop yields the strongest results.

with the standard recurrence

$$T_0 = \boldsymbol{I}, \qquad T_1 = \tilde{\boldsymbol{L}}, \qquad T_k = 2\tilde{\boldsymbol{L}}T_{k-1} - T_{k-2} \quad (k \geq 2).$$

**Comparison of Filter Types**  Our first set of experiments evaluates three view-finder sets, each consisting of one filter family applied over multiple hops:

$$\{\boldsymbol{I}\} \ \cup \ \{\nu_{\text{type}}^{(k)}(\boldsymbol{A})\}_{k=1}^{K},$$

where "type" denotes RW, SYM, or Cheb. As shown in Figure 5 (left), both RW and SYM maintain strong performance across hop sizes and consistently outperform GraphAny. While RW alone does not exceed the strongest GNN baseline (UniMP), SYM alone does, especially when multi-hop views are included. This suggests that symmetric normalization provides a more stable and informative view space, and multi-hop views offer a richer set of signals for GVT to operate on.

In contrast, the Chebyshev filter performs well at $K = 1$ but degrades as more hops are added. We believe this behavior arises because higher-order Chebyshev polynomials increasingly amplify high-frequency components, introducing more noise into the view space. Without any mechanism to suppress this noise, GVT may overfit these noisy signals during training, leading to reduced performance. These observations suggest that, to leverage multi-hop information in the view space, smoother diffusion-style filters (RW, SYM) are more stable than Chebyshev filters.

**Comparison of Filter-Type Combinations**  We then study whether mixing different filter families provides benefits. Each view-finder set includes two filter types (RW-SYM, RW-Cheb, SYM-Cheb), again evaluated across multiple hops:

$$\{\boldsymbol{I}\} \ \cup \ \{\nu_{\text{type1}}^{(k)}(\boldsymbol{A})\}_{k=1}^{K} \ \cup \ \{\nu_{\text{type2}}^{(k)}(\boldsymbol{A})\}_{k=1}^{K}.$$

Mixing RW and SYM filters improves performance for hop sizes 1 through 3 but not for 4 and 5. This suggests that multiple filter types can introduce complementary information into the view space, yielding better performance at moderate hop sizes. However, at larger hops the benefit diminishes, likely due to redundant or overly diffuse information.

Consistent with the earlier results, mixing Cheb with either RW or SYM does not improve performance beyond $K = 1$. This suggests that the degradation caused by high-frequency noise cannot be mitigated simply by adding more stable views.

**Guidelines**  Based on these observations, we recommend constructing view-finder sets by combining diffusion-style filters (SYM, RW) with moderate hop sizes (typically 2–3). Spectral filters such as Chebyshev should be used cautiously, as higher-order variants can amplify high-frequency noise and lead to overfitting in GVT. Finally, because the computational cost of RGVT grows quadratically with the number of view finders (see Appendix H), it is important to keep the set compact.

## J. Experiment Configurations

This section provides additional implementation details for RGVT and the baseline methods, including the hyperparameter search spaces, model selection procedure, and training setup used in our experiments.

*Table 11.* Performance (%) of RGVT + MLP under different recurrent depths $L$ during pretraining. The **best** and second-best results are highlighted.

| Depths ($L$) | Ogbn-arxiv (1) | Signed Dense (5) | Unsigned Dense (4) | Sparse (4) | Binary Dense (3) | Binary Sparse (8) | One-hot (4) | Total Avg. (28) |
|---|---|---|---|---|---|---|---|---|
| 24 | $71.11_{\pm 0.03}$ | $66.52_{\pm 0.87}$ | $77.26_{\pm 0.43}$ | $83.80_{\pm 1.62}$ | $84.88_{\pm 0.41}$ | $63.74_{\pm 2.78}$ | $66.15_{\pm 5.31}$ | $71.64_{\pm 2.05}$ |
| 20 | $71.32_{\pm 0.10}$ | **$66.60_{\pm 0.64}$** | **$77.64_{\pm 0.24}$** | $83.80_{\pm 0.54}$ | **$85.24_{\pm 0.47}$** | $64.07_{\pm 2.62}$ | **$68.25_{\pm 4.74}$** | **$72.14_{\pm 1.70}$** |
| 16 | **$71.38_{\pm 0.10}$** | $66.20_{\pm 0.69}$ | $77.35_{\pm 0.37}$ | $83.13_{\pm 1.62}$ | $84.89_{\pm 0.64}$ | **$64.88_{\pm 2.23}$** | $67.64_{\pm 4.49}$ | $72.04_{\pm 1.75}$ |
| 12 | $71.28_{\pm 0.28}$ | $66.13_{\pm 1.03}$ | $77.50_{\pm 0.57}$ | $83.39_{\pm 1.18}$ | $84.94_{\pm 0.57}$ | $64.07_{\pm 1.56}$ | $65.55_{\pm 3.82}$ | $71.57_{\pm 1.49}$ |
| 8 | $71.11_{\pm 0.28}$ | $66.37_{\pm 0.90}$ | $77.12_{\pm 0.45}$ | **$83.98_{\pm 0.81}$** | $84.86_{\pm 0.51}$ | $63.87_{\pm 1.58}$ | $62.48_{\pm 3.95}$ | $71.13_{\pm 1.41}$ |
| 4 | $71.26_{\pm 0.19}$ | $66.32_{\pm 0.63}$ | $76.68_{\pm 0.37}$ | $82.85_{\pm 2.24}$ | $84.70_{\pm 0.55}$ | $62.02_{\pm 2.48}$ | $55.93_{\pm 3.59}$ | $69.42_{\pm 1.77}$ |
| 2 | $71.22_{\pm 0.15}$ | $65.09_{\pm 2.00}$ | $75.14_{\pm 1.65}$ | $83.11_{\pm 0.73}$ | $84.51_{\pm 0.99}$ | $56.02_{\pm 4.70}$ | $56.25_{\pm 3.65}$ | $67.32_{\pm 2.67}$ |

### J.1. Our Method

For RGVT, we use $\{\boldsymbol{I}\} \cup \{(\boldsymbol{D}^{-1}\boldsymbol{A})^k, (\boldsymbol{D}^{-\frac{1}{2}}\boldsymbol{A}\boldsymbol{D}^{-\frac{1}{2}})^k\}_{k=1}^{K}$ as the view-finder set, where $\boldsymbol{D}$ is the degree matrix and consider $K \in \{1, 2, 3\}$ as a hyperparameter. The search space further includes the number of MLP layers in the mapping $\phi$ of GVT $D \in \{1, 2, 3\}$, the recurrent depth $L \in \{2, 4, 6, 8\}$ during pretraining, and the learning rate $\eta \in \{0.01, 0.05\}$. We use Gaussian Error Linear Units (GELUs) as the activation function in the nonlinear GVT.

Hyperparameters are selected using only the pretraining dataset. After pretraining is completed, we reinitialize the predictor, retrain the predictor on the same dataset while keeping RGVT frozen, and evaluate validation accuracy. This procedure offers a more reliable basis for selecting a model that generalizes well than relying on the validation accuracy recorded during pretraining. The selected hyperparameter settings from the search are as follows:

- RGVT + Linear : $K = 2, D = 2, L = 8, \eta = 0.01$,
- RGVT + MLP : $K = 2, D = 2, L = 8, \eta = 0.005$.

### J.2. Baselines

For the GNN baselines(Kipf & Welling, 2017; Hamilton et al., 2017; Xu et al., 2018a;b; Veličković et al., 2018; Wu et al., 2019; Gasteiger et al., 2019; Chen et al., 2020; Brody et al., 2021; Zhu & Koniusz, 2021; Chien et al., 2021; Shi et al., 2021), we conducted a hyperparameter search over hidden dimensions {64, 128, 256}, depths {1,2,3,4,5}, and learning rates {0.01, 0.005, 0.001}, performed separately for each dataset. For GPRGNN and JKNet, we use the official implementation from the GPRGNN public repository (`https://github.com/jianhao2016/GPRGNN`), while all other models are adopted from PyG (Fey & Lenssen, 2019) implementation.

Both GNN baselines and RGVT are optimized with the Adam optimizer for up to 2500 epochs, with early stopping if no improvement in validation accuracy is observed for 200 consecutive epochs. This setup follows prior work (Luo et al., 2024), which demonstrated that extensive hyperparameter search leads to strong GNN performance. All hyperparameter searches, both for RGVT and for the GNN baselines are performed over 5 independent runs, and final test results are re-evaluated under the configuration selected based on average validation accuracy.

## K. Additional Ablation Studies

We conducted two additional ablation studies on (i) the recurrent depth of RGVT during pretraining and (ii) the choice of pretraining dataset. First, as shown in Table 11, RGVT generally exhibits improved performance as the recurrent depth increases. Although the main experiment's search space did not include the best-performing depth, this suggests that a more extensive hyperparameter search could further enhance RGVT's performance. We attribute these gains to two factors: deeper pretraining allows the model to experience more diverse feature distributions, and during adaptation, a finer-grained choice of depth enables more compatible transfer.

Second, we evaluated three alternative datasets for pretraining. RGVT consistently performed worse on these datasets compared to OGBN-Arxiv, suggesting that it benefits from the larger scale of OGBN-Arxiv. Unlike GraphAny, whose

*Table 12.* Performance (%) of RGVT + MLP under different pretraining datasets. Pretraining on OGBN-Arxiv yields consistent improvements across all feature types and achieves the highest overall performance compared to smaller pretraining datasets (AmzRatings, Cora, Wisconsin).

| Method | Ogbn-arxiv (1) | Signed Dense (5) | Unsigned Dense (4) | Sparse (4) | Binary Dense (3) | Binary Sparse (8) | One-hot (4) | Total Avg. (28) |
|---|---|---|---|---|---|---|---|---|
| Ogbn-Arxiv | $71.11_{\pm 0.28}$ | $66.37_{\pm 0.90}$ | $77.12_{\pm 0.45}$ | $83.98_{\pm 0.81}$ | $84.86_{\pm 0.51}$ | $63.87_{\pm 1.58}$ | $62.48_{\pm 3.95}$ | $71.13_{\pm 1.41}$ |
| AmzRatings | $69.09_{\pm 0.52}$ | $66.24_{\pm 0.69}$ | $76.37_{\pm 0.95}$ | $82.90_{\pm 1.18}$ | $83.55_{\pm 1.54}$ | $57.72_{\pm 4.12}$ | $58.23_{\pm 7.05}$ | $68.34_{\pm 2.78}$ |
| Cora | $68.31_{\pm 4.96}$ | $64.94_{\pm 3.01}$ | $75.41_{\pm 1.31}$ | $79.42_{\pm 7.97}$ | $82.93_{\pm 3.22}$ | $54.21_{\pm 5.47}$ | $61.07_{\pm 8.53}$ | $66.81_{\pm 4.99}$ |
| Wisconsin | $68.06_{\pm 3.09}$ | $64.64_{\pm 2.08}$ | $75.44_{\pm 1.60}$ | $79.78_{\pm 4.39}$ | $82.13_{\pm 2.88}$ | $57.33_{\pm 5.88}$ | $65.46_{\pm 7.08}$ | $68.25_{\pm 4.23}$ |

*Table 13.* Performance (%) comparison of RGVT with an MLP predictor and TabPFN as the classifier on 17 benchmark datasets, restricted to settings within TabPFN's limits. RGVT+TabPFN enables zero-training prediction on unseen datasets and outperforms the RGVT+MLP in average accuracy. **Bold** numbers indicate the higher value in each column.

| | Actor | AirBrazil | AirEU | AirUSA | AmzComp | AmzPhoto | AmzRatings | Cora | Cornell |
|---|---|---|---|---|---|---|---|---|---|
| RGVT+MLP | $\mathbf{34.00_{\pm 1.14}}$ | $66.15_{\pm 10.67}$ | $47.00_{\pm 2.70}$ | $56.79_{\pm 2.26}$ | $83.72_{\pm 0.46}$ | $\mathbf{92.15_{\pm 0.36}}$ | $48.61_{\pm 1.53}$ | $\mathbf{81.18_{\pm 1.15}}$ | $74.05_{\pm 1.48}$ |
| RGVT+TabPFN | $29.20_{\pm 0.97}$ | $\mathbf{71.54_{\pm 5.83}}$ | $\mathbf{54.00_{\pm 2.88}}$ | $\mathbf{63.86_{\pm 1.40}}$ | $\mathbf{85.13_{\pm 0.33}}$ | $91.90_{\pm 0.21}$ | $\mathbf{54.88_{\pm 1.45}}$ | $78.06_{\pm 0.90}$ | $74.05_{\pm 4.91}$ |

| | DBLP | Deezer | Minesweeper | Pubmed | Texas | Tolokers | WikiCS | Wisconsin | Total Avg. |
|---|---|---|---|---|---|---|---|---|---|
| RGVT+MLP | $\mathbf{81.08_{\pm 1.22}}$ | $\mathbf{57.58_{\pm 0.61}}$ | $79.98_{\pm 0.18}$ | $78.72_{\pm 0.70}$ | $\mathbf{78.92_{\pm 1.21}}$ | $79.23_{\pm 0.42}$ | $79.59_{\pm 0.19}$ | $71.76_{\pm 1.75}$ | $70.03_{\pm 1.65}$ |
| RGVT+TabPFN | $76.48_{\pm 0.84}$ | $55.85_{\pm 0.55}$ | $\mathbf{80.92_{\pm 0.38}}$ | $\mathbf{79.48_{\pm 1.75}}$ | $70.27_{\pm 3.82}$ | $\mathbf{81.51_{\pm 0.39}}$ | $\mathbf{80.26_{\pm 0.25}}$ | $\mathbf{74.51_{\pm 2.77}}$ | $\mathbf{70.70_{\pm 1.74}}$ |

performance remained stable across different datasets, RGVT showed sensitivity to pretraining data choice. This highlights the importance of pretraining dataset selection, and we leave systematic exploration of dataset choice and the construction of dedicated pretraining corpora for future work.

# L. Tabular Foundation Models as Predictor

Recent tabular foundation models (TFMs) (Hollmann et al., 2022; 2025) enable prediction through in-context learning during inference, removing the need for explicit predictor training. In this section, we investigate whether GVT representations can be effectively used as inputs to TFMs and how their performance compares with standard lightweight MLP predictors.

Due to current limitations of TFMs on the acceptable number of feature dimensions and classes, we conduct experiments on the 17 benchmark subsets that satisfy these constraints. As shown in Table 13, using TabPFN improves performance on 10 out of 17 benchmarks and achieves better average performance than MLP predictors.

While these results demonstrate the potential of combining GVT with TFMs, current TFMs still suffer from limited scalability with respect to feature dimensionality, making them impractical for many real-world graph datasets with thousands of features. We therefore consider lightweight MLP predictor training as the current default choice in GVT, while more scalable TFMs represent a promising future direction.

# M. Recurrent Depth Selection

RGVT requires selecting the recurrent depth when transferring to different graph datasets, since different graphs often favor different levels of aggregation depth. In this section, we analyze how sensitive GVT is to the choice of recurrent depth by evaluating each depth across all 28 node classification benchmarks.

As shown in Figure 6, the optimal depth varies across datasets, with some favoring shallow models and others deeper ones, making validation-based selection important. However, Figure 7 shows that fixed depths (e.g., 4 or 6) remain competitive with strong GNN baselines on average. This suggests that fixed depths can serve as a reliable alternative when validation is not feasible. Developing depth-adaptive architectures remains an important direction for further eliminate this dependency.

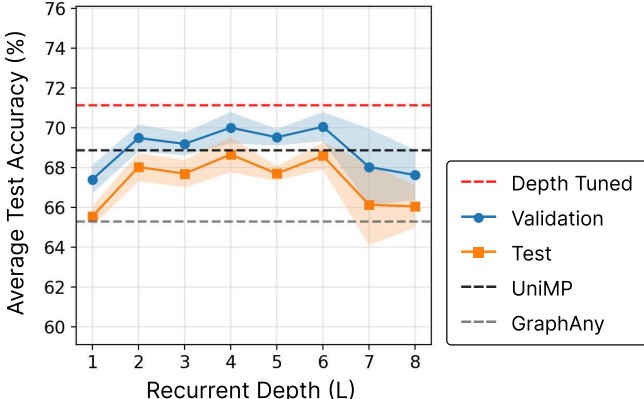

*Figure 6.* Accuracy (%) of RGVT + MLP across recurrent depths on 28 node classification benchmarks. The orange curve denotes test accuracy, while the blue curve represents validation accuracy. Performance varies with depth, with trends differing significantly across datasets. Selecting the depth based on validation accuracy is important for achieving consistently strong performance.

*Figure 7.* Average accuracy (%) of RGVT + MLP across recurrent depths. The model remains competitive with UniMP (black dotted line) at depths 4 and 6, and consistently outperforms GraphAny (gray dotted line) across all depths. With per-dataset depth selection via validation accuracy (red dotted line), RGVT surpasses UniMP.

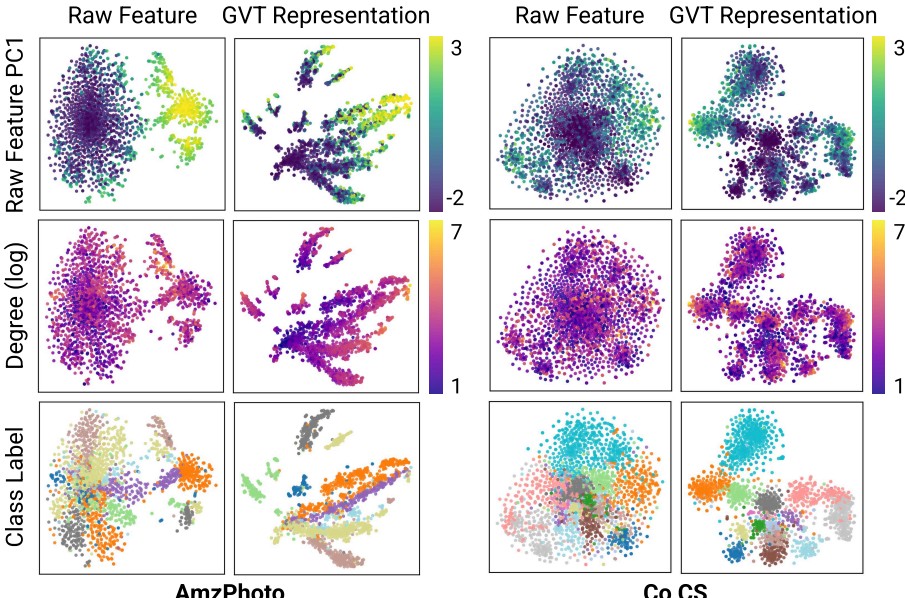

*Figure 8.* t-SNE visualizations of raw features and GVT representations on the transferred datasets: Amazon Photo (left) and Coauthor CS (right). Nodes are colored by the first principal component (PC1) of the raw features (top), node degree (middle), and class label (bottom). The GVT representations capture node degree trends while preserving trends in PC1, demonstrating their ability to encode structural information while maintaining feature semantics. This results in improved class separability and higher classification performance.

## N. Representation Visualization

In this section, we visualize the representations produced by GVT on transferred datasets using t-SNE (Figure 8). First, node-degree trends (middle row) emerge in the learned representations even though they are not apparent in the raw features, demonstrating that GVT embeds structural information into the representations. Second, the trends corresponding to the principal components of the raw features (top row) are largely preserved in the learned representations, suggesting that GVT maintains the underlying feature structure. This observation further supports our claim that, although GVT avoids explicit feature mixing, the original feature structure remains preserved, allowing downstream predictors to capture cross-feature interactions from the learned representations. Finally, the learned representations exhibit clearer class separability than the raw features, which likely contributes to improved downstream classification performance.

## O. Comparison with More Recent GNNs

We further compare RGVT against two groups of recent GNNs that provide complementary modeling perspectives beyond the baselines considered in the main experiments: (i) DirGNN (Rossi et al., 2024) and LSGNN (Chen et al., 2023), which are designed to improve performance on heterophilous graphs, and (ii) NodeFormer (Wu et al., 2022) and SGFormer (Wu et al., 2023), graph transformers that incorporate global attention mechanisms. We provide the full experimental results in Table 14, and summarize the average accuracy, Elo rating, and improvability in Figure 9.

DirGNN, LSGNN, and SGFormer improve over the original baseline set on average, with the largest gains observed on heterophilous datasets. Among them, DirGNN and SGFormer even outperform RGVT when using a linear predictor. However, neither surpasses RGVT with an MLP predictor on average, which remains the strongest configuration overall.

Although the performance gaps between RGVT and these recent baselines are smaller than those observed in the main experiments, we emphasize that the primary advantage of RGVT lies in efficient adaptation across unseen datasets rather than maximizing performance on a specific benchmark through per-dataset optimization. Unlike conventional GNNs, RGVT is transferred from a single pretrained model without dataset-specific retraining, while strong GNNs baselines typically require extensive hyperparameter tuning and retraining to achieve competitive performance.

Notably, evaluating these additional baselines required nearly three days on four H200 GPUs, whereas RGVT adapts across all 27 datasets in only 152 seconds (Appendix H). These results highlight that RGVT achieves competitive accuracy while providing substantially lower adaptation cost in practical deployment settings.

*Table 14.* Performance (%) comparison with six individually tuned GNNs, including the two best-performing baselines from our original set and four newly included recent methods, across 28 node classification benchmarks. The best and second-best results are highlighted in **bold** and underline, respectively. RGVT with an MLP predictor achieves the highest average accuracy.

| Dataset | GPRGNN | UniMP | NodeFormer | LSGNN | DirGNN | SGFormer | RGVT + Linear | RGVT + MLP |
|---|---|---|---|---|---|---|---|---|
| Actor | $34.51_{\pm0.31}$ | $35.97_{\pm1.39}$ | $34.37_{\pm0.89}$ | $34.32_{\pm1.25}$ | $33.78_{\pm1.88}$ | **$36.21_{\pm1.54}$** | $33.87_{\pm0.50}$ | $34.00_{\pm1.14}$ |
| AirBrazil | $31.54_{\pm4.21}$ | $44.62_{\pm9.65}$ | $43.85_{\pm10.39}$ | $48.46_{\pm3.44}$ | $45.38_{\pm8.77}$ | $46.92_{\pm3.22}$ | $52.31_{\pm6.99}$ | **$66.15_{\pm10.67}$** |
| AirEU | $34.37_{\pm1.65}$ | $39.25_{\pm3.14}$ | $42.88_{\pm2.28}$ | $38.88_{\pm2.31}$ | $42.38_{\pm3.26}$ | $46.13_{\pm4.11}$ | $46.25_{\pm1.93}$ | **$47.00_{\pm2.70}$** |
| AirUSA | $51.57_{\pm0.59}$ | $47.64_{\pm1.46}$ | $43.06_{\pm4.35}$ | $45.77_{\pm1.38}$ | $53.37_{\pm0.81}$ | $51.75_{\pm2.47}$ | **$57.08_{\pm3.38}$** | $56.79_{\pm2.26}$ |
| AmzComp | $85.55_{\pm0.31}$ | $84.06_{\pm0.70}$ | $74.32_{\pm0.62}$ | $82.62_{\pm1.00}$ | $86.16_{\pm0.47}$ | **$86.42_{\pm0.15}$** | $84.30_{\pm0.23}$ | $83.72_{\pm0.46}$ |
| AmzPhoto | $92.81_{\pm0.18}$ | $92.15_{\pm0.25}$ | $84.06_{\pm1.06}$ | $91.15_{\pm0.42}$ | $92.30_{\pm0.20}$ | **$92.82_{\pm0.10}$** | $92.02_{\pm0.30}$ | $92.15_{\pm0.36}$ |
| AmzRatings | $48.74_{\pm0.28}$ | $51.30_{\pm0.77}$ | $42.72_{\pm0.88}$ | $51.07_{\pm0.78}$ | $49.29_{\pm0.53}$ | **$52.69_{\pm0.36}$** | $42.98_{\pm0.13}$ | $48.61_{\pm1.53}$ |
| BlogCatalog | **$88.34_{\pm0.27}$** | $80.70_{\pm3.24}$ | $77.57_{\pm0.48}$ | $85.33_{\pm0.84}$ | $82.31_{\pm0.62}$ | $86.41_{\pm0.93}$ | $80.75_{\pm3.08}$ | $84.61_{\pm1.59}$ |
| Chameleon | $40.93_{\pm1.24}$ | $41.34_{\pm3.16}$ | $36.08_{\pm5.26}$ | $38.77_{\pm3.13}$ | $42.92_{\pm5.32}$ | $39.35_{\pm2.51}$ | $41.34_{\pm0.99}$ | **$44.74_{\pm2.17}$** |
| Citeseer | $70.30_{\pm0.62}$ | $68.56_{\pm0.19}$ | $71.30_{\pm0.80}$ | $68.42_{\pm2.24}$ | $66.50_{\pm1.07}$ | $65.92_{\pm0.20}$ | **$71.96_{\pm0.38}$** | $70.02_{\pm1.34}$ |
| CoCS | $91.83_{\pm0.21}$ | $89.84_{\pm0.39}$ | $92.25_{\pm0.78}$ | $91.45_{\pm0.31}$ | $91.42_{\pm0.13}$ | **$92.81_{\pm0.23}$** | $91.32_{\pm0.42}$ | $92.10_{\pm0.14}$ |
| CoPhysics | $93.20_{\pm0.10}$ | $91.53_{\pm0.76}$ | $93.81_{\pm0.23}$ | $93.29_{\pm0.37}$ | **$94.01_{\pm0.07}$** | $93.93_{\pm0.17}$ | $92.71_{\pm0.35}$ | $92.62_{\pm0.60}$ |
| Cora | **$82.08_{\pm0.66}$** | $79.18_{\pm1.13}$ | $76.70_{\pm1.06}$ | $75.62_{\pm1.03}$ | $78.04_{\pm1.51}$ | $79.14_{\pm1.09}$ | $81.32_{\pm0.64}$ | $81.18_{\pm1.15}$ |
| Cornell | $68.65_{\pm3.63}$ | $62.16_{\pm4.27}$ | $72.97_{\pm5.06}$ | **$78.38_{\pm3.31}$** | $72.43_{\pm6.73}$ | $77.30_{\pm3.08}$ | $73.51_{\pm4.44}$ | $74.05_{\pm1.48}$ |
| DBLP | $78.41_{\pm0.15}$ | $73.44_{\pm0.39}$ | $75.08_{\pm0.45}$ | $69.38_{\pm0.55}$ | $74.91_{\pm0.50}$ | $73.44_{\pm0.89}$ | **$81.29_{\pm1.01}$** | $81.08_{\pm1.22}$ |
| Deezer | $56.89_{\pm0.26}$ | $56.32_{\pm0.88}$ | $56.37_{\pm0.67}$ | $56.03_{\pm0.29}$ | $55.27_{\pm1.43}$ | $56.58_{\pm0.36}$ | **$57.79_{\pm0.29}$** | $57.58_{\pm0.61}$ |
| Minesweeper | $84.10_{\pm0.15}$ | **$88.25_{\pm0.17}$** | $80.00_{\pm0.00}$ | $84.04_{\pm0.46}$ | $86.30_{\pm0.25}$ | $83.46_{\pm0.50}$ | $79.77_{\pm0.27}$ | $79.98_{\pm0.18}$ |
| Pubmed | $77.44_{\pm0.44}$ | $76.46_{\pm0.65}$ | $75.96_{\pm2.38}$ | $75.44_{\pm0.84}$ | $75.88_{\pm0.77}$ | $77.28_{\pm0.69}$ | **$79.02_{\pm1.10}$** | $78.72_{\pm0.70}$ |
| Questions | $97.21_{\pm0.02}$ | $97.17_{\pm0.03}$ | $97.21_{\pm0.08}$ | $97.26_{\pm0.09}$ | $97.23_{\pm0.04}$ | $97.26_{\pm0.07}$ | $97.13_{\pm0.02}$ | $97.13_{\pm0.02}$ |
| Roman | $74.24_{\pm0.30}$ | $80.31_{\pm0.56}$ | $65.69_{\pm0.35}$ | $76.65_{\pm0.60}$ | **$89.17_{\pm0.31}$** | $76.35_{\pm0.40}$ | $67.71_{\pm0.87}$ | $70.18_{\pm0.28}$ |
| Squirrel | $35.80_{\pm0.96}$ | $40.18_{\pm1.97}$ | $35.87_{\pm1.52}$ | $34.90_{\pm1.59}$ | **$42.75_{\pm2.58}$** | $40.56_{\pm1.33}$ | $39.38_{\pm0.81}$ | $41.77_{\pm2.06}$ |
| Texas | $73.51_{\pm6.16}$ | $73.51_{\pm3.52}$ | $80.95_{\pm5.58}$ | **$82.04_{\pm3.79}$** | $76.59_{\pm5.19}$ | $80.38_{\pm6.51}$ | $75.14_{\pm2.96}$ | $78.92_{\pm1.21}$ |
| Tolokers | $78.41_{\pm0.06}$ | $81.46_{\pm0.38}$ | $78.33_{\pm0.24}$ | $80.61_{\pm0.83}$ | **$82.18_{\pm0.82}$** | $81.32_{\pm0.34}$ | $78.10_{\pm0.29}$ | $79.23_{\pm0.42}$ |
| Wiki | $74.60_{\pm0.37}$ | $72.93_{\pm1.12}$ | $77.02_{\pm0.31}$ | **$78.01_{\pm0.71}$** | $75.22_{\pm0.85}$ | $77.97_{\pm1.34}$ | $72.76_{\pm0.98}$ | $74.54_{\pm0.76}$ |
| Wisconsin | $64.71_{\pm2.40}$ | $65.88_{\pm9.76}$ | $85.49_{\pm4.92}$ | **$87.06_{\pm5.82}$** | $78.82_{\pm4.68}$ | $84.71_{\pm4.02}$ | $76.86_{\pm1.64}$ | $71.76_{\pm1.75}$ |
| WikiCS | $79.46_{\pm0.31}$ | $79.80_{\pm0.13}$ | $72.80_{\pm0.71}$ | $78.91_{\pm0.93}$ | $79.62_{\pm0.80}$ | **$79.96_{\pm0.70}$** | $79.59_{\pm0.16}$ | $79.59_{\pm0.19}$ |
| OGBN-Arxiv | $69.45_{\pm0.41}$ | **$71.78_{\pm0.12}$** | $53.53_{\pm0.23}$ | $68.12_{\pm0.10}$ | $69.64_{\pm0.23}$ | $70.82_{\pm0.13}$ | $70.14_{\pm0.28}$ | $71.11_{\pm0.28}$ |
| FullCora | $64.31_{\pm0.11}$ | $62.26_{\pm0.11}$ | $47.75_{\pm0.58}$ | $58.93_{\pm0.58}$ | $59.52_{\pm0.18}$ | $60.02_{\pm0.32}$ | **$64.35_{\pm0.77}$** | $62.35_{\pm2.22}$ |
| Avg. Acc. | $68.68_{\pm0.94}$ | $68.86_{\pm1.80}$ | $66.71_{\pm1.86}$ | $69.68_{\pm1.39}$ | $70.48_{\pm1.79}$ | $71.00_{\pm1.35}$ | $70.03_{\pm1.26}$ | **$71.13_{\pm1.41}$** |

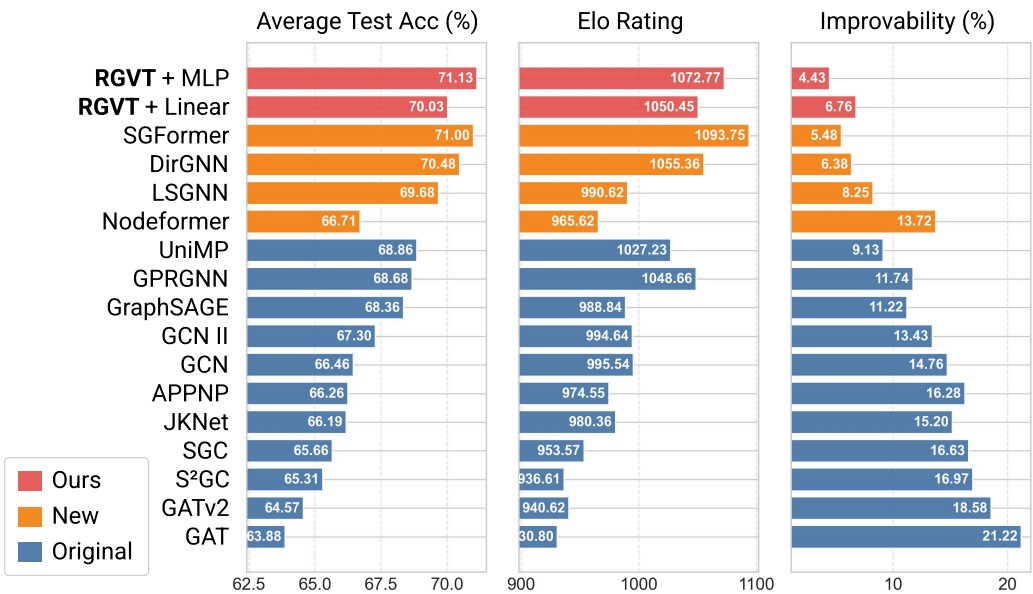

*Figure 9.* Bar plot summarizing experimental results across 28 node classification benchmarks. Results include 12 original supervised baselines (blue), 4 newly added supervised baselines (orange), and RGVT with linear and MLP predictors (red). RGVT with an MLP predictor ranks 1st in average test accuracy and improvability, and 2nd in Elo rating.

## P. Comparison with More Recent GFMs

Following GraphAny (Zhao et al., 2025), several fully inductive GFMs have recently been proposed to train a single model once and transfer it across arbitrary graph datasets without dataset-specific retraining. TS-Net (Finkelshtein et al., 2025) generalizes GraphAny by enforcing equivariance to both feature and node permutations while maintaining invariance to label permutations. TAG (Hayler et al., 2025) represents each node using neighborhood-smoothed features and positional embeddings, and performs prediction through subsampling and ensembles of tabular foundation models (TFMs). NodePFN (Choi et al., 2026) further extends TFMs to graph learning by incorporating graph message passing into the backbone architecture and introducing synthetic graph priors capable of generating varying levels of label homophily. We therefore additionally compare RGVT against these recent fully inductive GFMs. For fairness, we evaluate RGVT under their experimental pipelines and directly adopt the reported results of each method.

First, we emphasize that although these recent works also report strong improvements over tuned GNN baselines, the GNN baselines used in our experiments are substantially stronger, as shown in Table 15. In particular, our strongest baseline outperforms the strongest baseline reported in these works by 6.48% on average. This highlights both the extensive effort invested in our baseline optimization and the increased difficulty of outperforming our GNN baselines.

Second, as shown in Table 16, RGVT with an MLP predictor outperforms both TS-Net and TAG on average. While TAG remains relatively competitive, it requires substantially heavier adaptation procedures than RGVT. In particular, TAG relies on an ensemble of 14 predictors, including 10 TFMs whose complexity scales quadratically with feature dimensionality, making deployment substantially more expensive on high-dimensional datasets.

Third, as shown in Table 17, RGVT with an MLP predictor also outperforms NodePFN on average, although NodePFN remains competitive on several heterophilous datasets. However, NodePFN additionally requires extensive per-dataset preprocessing selection, including dimensionality reduction, feature smoothing, and ensemble configurations, resulting in approximately 400 candidate configurations per dataset.

Overall, these results demonstrate that RGVT achieves state-of-the-art fully inductive graph learning performance while remaining more efficient than recent GFMs.

## Q. Dataset Details

Our dataset selection and splitting strategy largely follows prior work GraphAny (Zhao et al., 2025). We access datasets through PyG (Fey & Lenssen, 2019), DGL (Wang et al., 2019), and the heterophilous graph collection from the Yandex-Research repository `https://github.com/yandex-research/heterophilous-graphs`. For the Chameleon and Squirrel datasets, we adopt the filtered versions provided in this repository, as the original datasets have been reported to contain issues (Platonov et al., 2023). Detailed statistics for each dataset, together with their assigned feature-type groups used in the main text, are summarized in Table 18.

## R. Full Experiment Results

Full experimental results are provided on pages 28–30.

*Table 15.* Average test accuracy (%) over 5 runs on 23 node classification benchmarks. We compare our GNN baselines with those reported in prior fully inductive works (GraphAny, TS-Net, TAG, and NodePFN) under identical datasets and split protocols. The best and second-best results are highlighted in **bold** and underline, respectively. Our GCN and GAT outperform prior counterparts in both average accuracy and rank, while GPRGNN and UniMP further surpass the strongest baselines by 6.34% and 6.48%, achieving second and first place, respectively.

| Dataset | GCN | GAT | GCN | GAT | GCN | GAT | GPRGNN | UniMP |
|---|---|---|---|---|---|---|---|---|
| Source | GraphAny, NodePFN | | TS-Net, TAG | | Ours | | | |
| Actor | $28.55_{\pm0.68}$ | $27.30_{\pm0.22}$ | $32.03_{\pm0.29}$ | $32.59_{\pm0.83}$ | $30.37_{\pm0.44}$ | $28.57_{\pm0.82}$ | $\underline{34.51}_{\pm0.31}$ | $\mathbf{35.97}_{\pm1.39}$ |
| AirBrazil | $42.31_{\pm7.98}$ | $\underline{57.69}_{\pm14.75}$ | $32.31_{\pm7.50}$ | $35.38_{\pm4.21}$ | $\mathbf{63.08}_{\pm3.44}$ | $44.62_{\pm14.54}$ | $31.54_{\pm4.21}$ | $44.62_{\pm9.65}$ |
| AirEU | $\mathbf{41.88}_{\pm3.60}$ | $32.50_{\pm8.45}$ | $39.12_{\pm6.44}$ | $39.00_{\pm4.30}$ | $\underline{41.50}_{\pm0.84}$ | $40.00_{\pm1.93}$ | $34.37_{\pm1.65}$ | $39.25_{\pm3.14}$ |
| AirUS | $46.49_{\pm1.81}$ | $48.47_{\pm4.17}$ | $43.93_{\pm1.16}$ | $43.03_{\pm2.08}$ | $\mathbf{53.30}_{\pm1.18}$ | $51.24_{\pm2.90}$ | $\underline{51.57}_{\pm0.59}$ | $47.64_{\pm1.46}$ |
| AmzComp | $\underline{85.83}_{\pm0.86}$ | $\mathbf{87.01}_{\pm0.50}$ | $73.88_{\pm0.88}$ | $70.94_{\pm3.40}$ | $84.69_{\pm0.18}$ | $84.43_{\pm1.28}$ | $85.55_{\pm0.31}$ | $84.06_{\pm0.70}$ |
| AmzPhoto | $91.88_{\pm0.79}$ | $91.86_{\pm1.07}$ | $88.95_{\pm1.08}$ | $80.78_{\pm3.59}$ | $91.71_{\pm0.18}$ | $89.40_{\pm0.71}$ | $\mathbf{92.81}_{\pm0.18}$ | $\underline{92.15}_{\pm0.25}$ |
| AmzRatings | $47.35_{\pm0.26}$ | $47.18_{\pm0.42}$ | $40.74_{\pm0.13}$ | $40.63_{\pm0.66}$ | $49.17_{\pm0.43}$ | $48.99_{\pm0.56}$ | $48.74_{\pm0.28}$ | $\mathbf{51.30}_{\pm0.77}$ |
| BlogCatalog | $71.51_{\pm2.62}$ | $61.98_{\pm5.99}$ | $84.48_{\pm0.74}$ | $78.20_{\pm7.23}$ | $\underline{86.00}_{\pm0.52}$ | $55.16_{\pm4.77}$ | $\mathbf{88.34}_{\pm0.27}$ | $80.70_{\pm3.24}$ |
| Citeseer | $63.40_{\pm0.63}$ | $69.10_{\pm1.59}$ | $65.06_{\pm1.30}$ | $63.92_{\pm0.84}$ | $69.22_{\pm0.62}$ | $69.46_{\pm1.36}$ | $\mathbf{70.30}_{\pm0.62}$ | $68.56_{\pm0.19}$ |
| CoCS | $\mathbf{91.83}_{\pm0.71}$ | $88.47_{\pm0.79}$ | $80.87_{\pm0.69}$ | $82.28_{\pm0.86}$ | $90.60_{\pm0.09}$ | $90.27_{\pm0.57}$ | $\mathbf{91.83}_{\pm0.21}$ | $89.84_{\pm0.39}$ |
| CoPhysics | $\mathbf{93.93}_{\pm0.37}$ | $93.01_{\pm0.89}$ | $79.05_{\pm1.13}$ | $85.92_{\pm1.10}$ | $92.85_{\pm0.24}$ | $91.22_{\pm2.06}$ | $\underline{93.20}_{\pm0.10}$ | $91.53_{\pm0.76}$ |
| Cornell | $35.14_{\pm6.51}$ | $35.14_{\pm3.52}$ | $63.78_{\pm1.48}$ | $\mathbf{69.73}_{\pm2.26}$ | $45.41_{\pm3.52}$ | $40.00_{\pm2.26}$ | $\underline{68.65}_{\pm3.63}$ | $62.16_{\pm4.27}$ |
| DBLP | $73.02_{\pm2.22}$ | $73.87_{\pm1.35}$ | $65.01_{\pm2.40}$ | $67.34_{\pm2.75}$ | $\mathbf{79.52}_{\pm1.03}$ | $\underline{79.07}_{\pm0.78}$ | $78.41_{\pm0.15}$ | $73.44_{\pm0.39}$ |
| Deezer | $53.69_{\pm2.29}$ | $55.99_{\pm3.78}$ | $54.91_{\pm1.81}$ | $55.22_{\pm2.33}$ | $\mathbf{57.09}_{\pm0.62}$ | $55.57_{\pm0.34}$ | $\underline{56.89}_{\pm0.26}$ | $56.32_{\pm0.88}$ |
| FullCora | $61.06_{\pm0.24}$ | $58.95_{\pm0.55}$ | $55.85_{\pm1.04}$ | $59.95_{\pm0.88}$ | $\underline{62.69}_{\pm0.13}$ | $60.71_{\pm0.16}$ | $\mathbf{64.31}_{\pm0.11}$ | $62.26_{\pm0.11}$ |
| Minesweeper | $81.12_{\pm0.37}$ | $80.08_{\pm0.04}$ | $84.06_{\pm0.18}$ | $84.15_{\pm0.24}$ | $80.16_{\pm0.26}$ | $81.70_{\pm0.53}$ | $\underline{84.10}_{\pm0.15}$ | $\mathbf{88.25}_{\pm0.17}$ |
| OGBN-Arxiv | $71.74_{\pm0.29}$ | $\mathbf{73.65}_{\pm0.11}$ | $50.94_{\pm0.38}$ | $57.93_{\pm3.44}$ | $71.19_{\pm0.30}$ | $71.89_{\pm0.24}$ | $69.45_{\pm0.41}$ | $71.78_{\pm0.12}$ |
| Pubmed | $76.60_{\pm0.32}$ | $\underline{77.30}_{\pm0.60}$ | $74.56_{\pm0.13}$ | $75.12_{\pm0.89}$ | $76.90_{\pm0.42}$ | $76.26_{\pm0.70}$ | $\mathbf{77.44}_{\pm0.44}$ | $76.46_{\pm0.65}$ |
| Questions | $97.15_{\pm0.04}$ | $97.11_{\pm0.02}$ | $97.16_{\pm0.06}$ | $97.13_{\pm0.05}$ | $97.06_{\pm0.02}$ | $97.06_{\pm0.02}$ | $\mathbf{97.21}_{\pm0.02}$ | $\underline{97.17}_{\pm0.03}$ |
| Roman | $45.08_{\pm0.43}$ | $43.93_{\pm0.45}$ | $69.37_{\pm0.66}$ | $69.80_{\pm4.18}$ | $45.33_{\pm0.25}$ | $43.42_{\pm0.18}$ | $\underline{74.24}_{\pm0.30}$ | $\mathbf{80.31}_{\pm0.56}$ |
| Tolokers | $79.93_{\pm0.10}$ | $78.50_{\pm0.55}$ | $78.59_{\pm0.66}$ | $78.22_{\pm0.37}$ | $80.60_{\pm0.12}$ | $\mathbf{81.67}_{\pm0.18}$ | $78.41_{\pm0.06}$ | $\underline{81.46}_{\pm0.38}$ |
| Wiki | $70.09_{\pm1.51}$ | $59.63_{\pm4.16}$ | $74.23_{\pm0.89}$ | $68.91_{\pm9.50}$ | $69.24_{\pm0.76}$ | $58.64_{\pm1.93}$ | $\mathbf{74.60}_{\pm0.37}$ | $72.93_{\pm1.12}$ |
| WikiCS | $79.12_{\pm0.45}$ | $79.27_{\pm0.20}$ | $71.97_{\pm1.70}$ | $74.99_{\pm0.59}$ | $79.07_{\pm0.24}$ | $79.44_{\pm0.40}$ | $\underline{79.46}_{\pm0.31}$ | $\mathbf{79.80}_{\pm0.13}$ |
| **Average accuracy** | 66.47 | 66.00 | 65.25 | 65.70 | 69.42 | 66.03 | 70.69 | **70.78** |
| **Average rank** | **4th** | **6th** | **8th** | **7th** | **3rd** | **5th** | **2nd** | **1st** |

*Table 16.* Performance (%) comparison across 28 node classification benchmarks against recent fully inductive methods: GraphAny, TS-Net, and TAG. Results on the Chameleon and Squirrel datasets are reproduced using the filtered versions, while all other results are adopted from TAG. For fairness, RGVT is also evaluated using the same TAG experimental pipeline. The best and second-best results are highlighted in **bold** and underline, respectively. RGVT achieves the best performance in both average accuracy and rank.

| **Dataset** | GraphAny (Cora) | GraphAny (Arxiv) | GraphAny (Products) | GraphAny (Wisconsin) | TS-Mean | TS-GAT | TAG | **RGVT +MLP** |
|---|---|---|---|---|---|---|---|---|
| Actor | $27.91_{\pm 0.16}$ | $28.60_{\pm 0.21}$ | $28.99_{\pm 0.61}$ | $29.51_{\pm 0.55}$ | $28.09_{\pm 0.93}$ | $28.80_{\pm 2.12}$ | $31.18_{\pm 0.18}$ | $\mathbf{32.68}_{\pm 1.26}$ |
| AirBrazil | $33.07_{\pm 16.68}$ | $34.61_{\pm 16.09}$ | $34.61_{\pm 16.54}$ | $36.15_{\pm 16.68}$ | $39.23_{\pm 5.70}$ | $36.92_{\pm 10.03}$ | $\underline{73.08}_{\pm 4.03}$ | $\mathbf{75.38}_{\pm 6.44}$ |
| AirEU | $40.50_{\pm 7.01}$ | $41.50_{\pm 6.50}$ | $41.75_{\pm 6.84}$ | $41.13_{\pm 6.02}$ | $35.88_{\pm 6.91}$ | $38.38_{\pm 6.32}$ | $55.38_{\pm 2.30}$ | $\mathbf{57.12}_{\pm 3.55}$ |
| AirUS | $43.46_{\pm 1.45}$ | $43.64_{\pm 1.83}$ | $43.57_{\pm 2.07}$ | $43.86_{\pm 1.44}$ | $42.34_{\pm 2.12}$ | $41.69_{\pm 2.59}$ | $\mathbf{60.50}_{\pm 0.80}$ | $57.55_{\pm 1.28}$ |
| AmzComp | $82.99_{\pm 1.22}$ | $83.04_{\pm 1.24}$ | $82.90_{\pm 1.25}$ | $82.00_{\pm 1.14}$ | $81.37_{\pm 1.25}$ | $80.23_{\pm 2.44}$ | $\mathbf{84.33}_{\pm 0.33}$ | $84.14_{\pm 1.76}$ |
| AmzPhoto | $90.14_{\pm 0.93}$ | $90.60_{\pm 0.82}$ | $\underline{90.64}_{\pm 0.82}$ | $90.18_{\pm 0.91}$ | $90.18_{\pm 1.30}$ | $89.66_{\pm 1.23}$ | $89.63_{\pm 0.48}$ | $\mathbf{92.29}_{\pm 0.44}$ |
| AmzRatings | $42.84_{\pm 0.04}$ | $42.74_{\pm 0.12}$ | $42.70_{\pm 0.10}$ | $42.57_{\pm 0.34}$ | $42.27_{\pm 1.40}$ | $41.92_{\pm 0.47}$ | $44.34_{\pm 0.32}$ | $\mathbf{45.58}_{\pm 1.16}$ |
| BlogCatalog | $72.52_{\pm 3.22}$ | $73.63_{\pm 2.95}$ | $74.73_{\pm 3.19}$ | $77.69_{\pm 1.90}$ | $76.30_{\pm 2.92}$ | $78.44_{\pm 5.18}$ | $\underline{79.77}_{\pm 1.06}$ | $\mathbf{81.39}_{\pm 2.75}$ |
| Chameleon | $32.17_{\pm 4.14}$ | $32.72_{\pm 4.78}$ | $32.75_{\pm 5.15}$ | $32.24_{\pm 4.18}$ | $43.83_{\pm 3.99}$ | $36.40_{\pm 8.26}$ | $\mathbf{44.98}_{\pm 1.37}$ | $44.74_{\pm 2.17}$ |
| Citeseer | $\underline{68.90}_{\pm 0.07}$ | $68.34_{\pm 0.23}$ | $67.94_{\pm 0.29}$ | $67.50_{\pm 0.44}$ | $68.66_{\pm 0.19}$ | $68.70_{\pm 0.48}$ | $68.08_{\pm 0.61}$ | $\mathbf{70.12}_{\pm 1.47}$ |
| CoCS | $90.47_{\pm 0.63}$ | $90.45_{\pm 0.59}$ | $90.46_{\pm 0.54}$ | $90.85_{\pm 0.63}$ | $90.92_{\pm 0.47}$ | $\underline{91.13}_{\pm 0.43}$ | $90.49_{\pm 0.50}$ | $\mathbf{92.28}_{\pm 0.37}$ |
| CoPhysics | $\underline{92.70}_{\pm 0.54}$ | $92.69_{\pm 0.52}$ | $92.66_{\pm 0.52}$ | $92.54_{\pm 0.43}$ | $92.61_{\pm 0.61}$ | $92.23_{\pm 0.61}$ | $92.31_{\pm 0.27}$ | $\mathbf{93.18}_{\pm 0.51}$ |
| Cora | $\underline{80.18}_{\pm 0.13}$ | $79.38_{\pm 0.16}$ | $79.36_{\pm 0.23}$ | $77.82_{\pm 1.15}$ | $58.08_{\pm 0.48}$ | $22.42_{\pm 12.90}$ | $\mathbf{83.50}_{\pm 0.40}$ | $81.68_{\pm 1.13}$ |
| Cornell | $64.86_{\pm 1.91}$ | $65.94_{\pm 1.48}$ | $64.86_{\pm 0.00}$ | $66.49_{\pm 1.48}$ | $68.65_{\pm 2.42}$ | $71.35_{\pm 4.52}$ | $\mathbf{75.14}_{\pm 2.16}$ | $\underline{74.05}_{\pm 2.42}$ |
| DBLP | $71.73_{\pm 0.94}$ | $70.90_{\pm 0.88}$ | $70.62_{\pm 0.97}$ | $70.13_{\pm 0.77}$ | $66.42_{\pm 3.65}$ | $64.30_{\pm 3.75}$ | $71.92_{\pm 1.46}$ | $\mathbf{78.12}_{\pm 1.19}$ |
| Deezer | $51.98_{\pm 2.79}$ | $52.11_{\pm 2.79}$ | $52.09_{\pm 2.78}$ | $52.13_{\pm 3.02}$ | $52.31_{\pm 2.52}$ | $52.88_{\pm 2.86}$ | $\underline{52.99}_{\pm 1.65}$ | $\mathbf{56.18}_{\pm 1.38}$ |
| FullCora | $56.73_{\pm 0.41}$ | $\underline{57.25}_{\pm 0.43}$ | $57.13_{\pm 0.37}$ | $56.29_{\pm 0.17}$ | $53.58_{\pm 0.73}$ | $52.82_{\pm 0.90}$ | $54.56_{\pm 0.19}$ | $\mathbf{63.26}_{\pm 0.75}$ |
| Minesweeper | $80.46_{\pm 0.15}$ | $80.30_{\pm 0.13}$ | $80.27_{\pm 0.16}$ | $80.13_{\pm 0.09}$ | $80.68_{\pm 0.38}$ | $\underline{80.72}_{\pm 0.78}$ | $\mathbf{85.83}_{\pm 0.13}$ | $80.00_{\pm 0.00}$ |
| OGBN-Arxiv | $58.62_{\pm 0.05}$ | $58.68_{\pm 0.17}$ | $58.58_{\pm 0.11}$ | $57.79_{\pm 0.56}$ | $56.33_{\pm 2.58}$ | $53.13_{\pm 2.04}$ | $66.70_{\pm 0.21}$ | $\mathbf{70.33}_{\pm 0.35}$ |
| Pubmed | $76.60_{\pm 0.31}$ | $76.36_{\pm 0.17}$ | $76.54_{\pm 0.34}$ | $77.46_{\pm 0.30}$ | $74.98_{\pm 0.56}$ | $75.46_{\pm 0.86}$ | $\mathbf{78.96}_{\pm 0.43}$ | $78.20_{\pm 0.82}$ |
| Questions | $97.06_{\pm 0.03}$ | $97.09_{\pm 0.02}$ | $97.10_{\pm 0.01}$ | $97.11_{\pm 0.00}$ | $97.02_{\pm 0.01}$ | $97.02_{\pm 0.01}$ | $\mathbf{97.14}_{\pm 0.01}$ | $\underline{97.13}_{\pm 0.06}$ |
| Roman | $64.25_{\pm 0.64}$ | $64.25_{\pm 1.09}$ | $64.66_{\pm 0.84}$ | $64.06_{\pm 0.78}$ | $66.36_{\pm 1.02}$ | $67.76_{\pm 0.60}$ | $\mathbf{74.12}_{\pm 0.28}$ | $70.53_{\pm 0.70}$ |
| Squirrel | $24.87_{\pm 1.68}$ | $25.85_{\pm 1.11}$ | $26.47_{\pm 1.28}$ | $25.58_{\pm 1.14}$ | $39.80_{\pm 0.53}$ | $35.84_{\pm 3.33}$ | $\mathbf{46.29}_{\pm 1.01}$ | $41.77_{\pm 2.06}$ |
| Texas | $71.89_{\pm 1.48}$ | $72.97_{\pm 2.71}$ | $73.52_{\pm 2.96}$ | $73.51_{\pm 1.21}$ | $73.51_{\pm 4.01}$ | $74.05_{\pm 4.10}$ | $\mathbf{81.08}_{\pm 2.09}$ | $78.92_{\pm 2.26}$ |
| Tolokers | $78.20_{\pm 0.02}$ | $78.18_{\pm 0.04}$ | $78.18_{\pm 0.03}$ | $78.24_{\pm 0.03}$ | $78.12_{\pm 0.09}$ | $78.01_{\pm 0.23}$ | $\mathbf{82.82}_{\pm 0.15}$ | $78.32_{\pm 0.35}$ |
| Wiki | $60.56_{\pm 3.62}$ | $62.96_{\pm 3.68}$ | $63.08_{\pm 3.61}$ | $61.10_{\pm 4.36}$ | $69.89_{\pm 1.31}$ | $\mathbf{72.53}_{\pm 1.59}$ | $70.09_{\pm 0.92}$ | $72.26_{\pm 4.66}$ |
| WikiCS | $74.39_{\pm 0.71}$ | $74.95_{\pm 0.61}$ | $75.01_{\pm 0.54}$ | $73.77_{\pm 0.83}$ | $74.16_{\pm 2.07}$ | $73.24_{\pm 2.49}$ | $\mathbf{79.07}_{\pm 0.54}$ | $78.92_{\pm 0.58}$ |
| Wisconsin | $61.18_{\pm 5.08}$ | $65.10_{\pm 3.22}$ | $65.89_{\pm 2.23}$ | $71.77_{\pm 5.98}$ | $61.18_{\pm 11.38}$ | $65.88_{\pm 9.86}$ | $\mathbf{83.14}_{\pm 1.33}$ | $\underline{78.04}_{\pm 4.47}$ |
| **Average accuracy** | 63.97 | 64.46 | 64.54 | 64.63 | 64.38 | 62.93 | 71.34 | **71.58** |
| **Average rank** | **7th** | **5th** | **4th** | **3rd** | **6th** | **8th** | **2nd** | **1st** |

*Table 17.* Performance (%) comparison against NodePFN on 23 node classification benchmarks. Results for all baselines are adopted from the original paper, while RGVT is re-evaluated under the same pipeline for fairness. The best and second-best results are highlighted in **bold** and underline, respectively. RGVT achieves the best performance in both average accuracy and ranking.

| Dataset | MLP | GCN | GAT | GraphAny (Products) | GraphAny (Arxiv) | GraphAny (Wisconsin) | GraphAny (Cora) | NodePFN | RGVT + MLP |
|---|---|---|---|---|---|---|---|---|---|
| **Homophily Graphs** | | | | | | | | | |
| AirBrazil | $23.08_{\pm 5.83}$ | $42.31_{\pm 7.98}$ | $\underline{57.69}_{\pm 14.75}$ | $34.61_{\pm 16.54}$ | $34.61_{\pm 16.09}$ | $36.15_{\pm 16.68}$ | $33.07_{\pm 16.68}$ | $\mathbf{75.38}_{\pm 1.88}$ | $\mathbf{75.38}_{\pm 6.44}$ |
| AirEU | $21.25_{\pm 2.31}$ | $41.88_{\pm 3.60}$ | $\underline{32.50}_{\pm 8.45}$ | $41.75_{\pm 6.84}$ | $41.50_{\pm 6.50}$ | $41.13_{\pm 6.02}$ | $40.50_{\pm 7.01}$ | $57.00_{\pm 1.21}$ | $\mathbf{57.12}_{\pm 3.55}$ |
| AirUS | $22.88_{\pm 1.46}$ | $46.49_{\pm 1.81}$ | $48.47_{\pm 4.17}$ | $43.57_{\pm 2.07}$ | $43.64_{\pm 1.83}$ | $43.86_{\pm 1.44}$ | $43.46_{\pm 1.40}$ | $\mathbf{61.66}_{\pm 0.31}$ | $\underline{57.55}_{\pm 1.28}$ |
| Cora | $48.42_{\pm 0.63}$ | $81.40_{\pm 0.70}$ | $\underline{81.70}_{\pm 1.43}$ | $79.36_{\pm 0.23}$ | $79.38_{\pm 0.16}$ | $77.82_{\pm 1.15}$ | $80.18_{\pm 0.13}$ | $\mathbf{82.06}_{\pm 0.29}$ | $81.68_{\pm 1.13}$ |
| Citeseer | $44.40_{\pm 0.44}$ | $63.40_{\pm 0.63}$ | $69.10_{\pm 1.59}$ | $67.94_{\pm 0.29}$ | $68.34_{\pm 0.23}$ | $67.50_{\pm 0.44}$ | $68.90_{\pm 0.07}$ | $67.30_{\pm 0.83}$ | $\mathbf{70.12}_{\pm 1.47}$ |
| Pubmed | $69.50_{\pm 1.79}$ | $76.60_{\pm 0.32}$ | $77.30_{\pm 0.60}$ | $76.54_{\pm 0.34}$ | $76.36_{\pm 0.17}$ | $77.46_{\pm 0.30}$ | $76.60_{\pm 0.31}$ | $\underline{78.00}_{\pm 0.24}$ | $\mathbf{78.20}_{\pm 0.82}$ |
| WikiCS | $72.72_{\pm 0.43}$ | $\underline{79.12}_{\pm 0.45}$ | $\mathbf{79.27}_{\pm 0.20}$ | $75.01_{\pm 0.54}$ | $74.95_{\pm 0.61}$ | $73.77_{\pm 0.83}$ | $74.39_{\pm 0.71}$ | $75.98_{\pm 0.80}$ | $78.92_{\pm 0.58}$ |
| Amazon-Photo | $68.20_{\pm 0.88}$ | $91.88_{\pm 0.79}$ | $91.86_{\pm 1.07}$ | $90.64_{\pm 0.82}$ | $90.60_{\pm 0.82}$ | $90.18_{\pm 0.91}$ | $90.14_{\pm 0.93}$ | $90.53_{\pm 0.13}$ | $\mathbf{92.29}_{\pm 0.44}$ |
| Amazon-Comp | $58.28_{\pm 2.98}$ | $\underline{85.83}_{\pm 0.86}$ | $\mathbf{87.01}_{\pm 0.50}$ | $82.90_{\pm 1.25}$ | $83.04_{\pm 1.24}$ | $82.00_{\pm 1.14}$ | $82.99_{\pm 1.22}$ | $81.42_{\pm 0.48}$ | $84.14_{\pm 1.76}$ |
| DBLP | $56.27_{\pm 0.62}$ | $73.02_{\pm 2.22}$ | $73.87_{\pm 1.35}$ | $70.62_{\pm 0.97}$ | $70.90_{\pm 0.88}$ | $70.13_{\pm 0.77}$ | $71.73_{\pm 0.94}$ | $\underline{74.71}_{\pm 0.39}$ | $\mathbf{78.12}_{\pm 1.19}$ |
| Coauthor CS | $85.88_{\pm 0.93}$ | $91.83_{\pm 0.71}$ | $88.47_{\pm 0.79}$ | $90.46_{\pm 0.54}$ | $90.45_{\pm 0.59}$ | $90.85_{\pm 0.63}$ | $90.47_{\pm 0.63}$ | $\underline{91.55}_{\pm 0.32}$ | $\mathbf{92.28}_{\pm 0.37}$ |
| Coauthor Physics | $87.43_{\pm 1.98}$ | $\mathbf{93.93}_{\pm 0.37}$ | $93.01_{\pm 0.89}$ | $92.66_{\pm 0.52}$ | $92.69_{\pm 0.52}$ | $92.54_{\pm 0.43}$ | $92.70_{\pm 0.54}$ | $\underline{93.43}_{\pm 0.13}$ | $93.18_{\pm 0.51}$ |
| Deezer | $54.24_{\pm 2.15}$ | $53.69_{\pm 2.29}$ | $\underline{55.99}_{\pm 3.78}$ | $52.09_{\pm 2.78}$ | $52.11_{\pm 2.79}$ | $52.13_{\pm 3.02}$ | $51.98_{\pm 2.79}$ | $53.45_{\pm 0.65}$ | $\mathbf{56.18}_{\pm 1.38}$ |
| **Average Accuracy** | 54.81 | 70.88 | 72.02 | 69.09 | 69.12 | 68.89 | 69.01 | $\underline{75.57}$ | **76.55** |
| **Average Ranking** | 8.31 | 3.38 | $\underline{3.31}$ | 5.92 | 5.69 | 6.15 | 6.00 | 3.38 | **1.69** |
| **Heterophily Graphs** | | | | | | | | | |
| Cornell | $67.57_{\pm 5.06}$ | $35.14_{\pm 6.51}$ | $35.14_{\pm 3.52}$ | $64.86_{\pm 0.00}$ | $65.94_{\pm 1.48}$ | $66.49_{\pm 1.48}$ | $64.86_{\pm 1.91}$ | $\underline{71.89}_{\pm 2.76}$ | $\mathbf{74.05}_{\pm 2.42}$ |
| Texas | $48.65_{\pm 4.01}$ | $51.35_{\pm 2.71}$ | $54.05_{\pm 2.41}$ | $73.52_{\pm 2.96}$ | $72.97_{\pm 2.71}$ | $73.51_{\pm 1.21}$ | $71.89_{\pm 1.48}$ | $\underline{76.22}_{\pm 7.53}$ | $\mathbf{78.92}_{\pm 2.26}$ |
| Wisconsin | $66.67_{\pm 3.51}$ | $37.25_{\pm 1.64}$ | $52.94_{\pm 3.10}$ | $65.89_{\pm 2.23}$ | $65.10_{\pm 3.22}$ | $71.77_{\pm 5.98}$ | $61.18_{\pm 5.08}$ | $\mathbf{79.22}_{\pm 6.97}$ | $\underline{78.04}_{\pm 4.47}$ |
| Chameleon | $38.87_{\pm 2.21}$ | $41.31_{\pm 3.05}$ | $39.83_{\pm 2.10}$ | $39.45_{\pm 4.20}$ | $37.40_{\pm 3.11}$ | $36.67_{\pm 5.32}$ | $37.99_{\pm 4.54}$ | $\mathbf{50.13}_{\pm 3.30}$ | $\underline{44.74}_{\pm 2.17}$ |
| Actor | $\mathbf{33.95}_{\pm 0.80}$ | $28.55_{\pm 0.68}$ | $27.30_{\pm 0.22}$ | $28.99_{\pm 0.61}$ | $28.60_{\pm 0.21}$ | $29.51_{\pm 0.55}$ | $27.91_{\pm 0.16}$ | $\underline{32.99}_{\pm 1.09}$ | $32.68_{\pm 1.26}$ |
| Minesweeper | $80.00_{\pm 0.00}$ | $\mathbf{81.12}_{\pm 0.37}$ | $80.08_{\pm 0.04}$ | $80.27_{\pm 0.16}$ | $80.30_{\pm 0.13}$ | $80.13_{\pm 0.09}$ | $80.46_{\pm 0.15}$ | $\underline{80.66}_{\pm 0.25}$ | $80.00_{\pm 0.00}$ |
| Tolokers | $78.16_{\pm 0.02}$ | $\mathbf{79.93}_{\pm 0.10}$ | $78.50_{\pm 0.55}$ | $78.18_{\pm 0.03}$ | $78.18_{\pm 0.04}$ | $78.24_{\pm 0.03}$ | $78.20_{\pm 0.02}$ | $\underline{78.61}_{\pm 0.06}$ | $78.32_{\pm 0.35}$ |
| Amazon-Ratings | $\mathbf{47.90}_{\pm 0.45}$ | $47.35_{\pm 0.26}$ | $47.18_{\pm 0.42}$ | $42.70_{\pm 0.10}$ | $42.74_{\pm 0.12}$ | $42.57_{\pm 0.34}$ | $42.84_{\pm 0.04}$ | $\underline{44.68}_{\pm 0.48}$ | $45.58_{\pm 1.16}$ |
| Questions | $\mathbf{97.33}_{\pm 0.06}$ | $\underline{97.15}_{\pm 0.04}$ | $97.11_{\pm 0.02}$ | $97.10_{\pm 0.01}$ | $97.09_{\pm 0.02}$ | $97.11_{\pm 0.00}$ | $97.06_{\pm 0.03}$ | $97.02_{\pm 0.01}$ | $97.13_{\pm 0.06}$ |
| Squirrel | $35.55_{\pm 0.98}$ | $38.67_{\pm 1.84}$ | $38.78_{\pm 2.39}$ | $38.92_{\pm 2.98}$ | $37.73_{\pm 2.31}$ | $36.76_{\pm 3.55}$ | $37.25_{\pm 2.65}$ | $\mathbf{43.40}_{\pm 1.03}$ | $\underline{41.77}_{\pm 2.06}$ |
| **Average Accuracy** | 59.46 | 53.78 | 55.09 | 60.99 | 60.60 | 61.28 | 59.96 | **65.48** | $\underline{65.12}$ |
| **Average Ranking** | 5.00 | 4.50 | 5.60 | 5.20 | 6.00 | 5.60 | 6.30 | **2.60** | $\underline{3.00}$ |
| **Avg. Accuracy** | 56.83 | 63.44 | 64.66 | 65.57 | 65.42 | 65.58 | 65.08 | $\underline{71.19}$ | **71.58** |
| **Avg. Ranking** | 6.87 | 3.87 | 4.30 | 5.61 | 5.83 | 5.91 | 6.13 | $\underline{3.04}$ | **2.26** |

*Table 18.* Statistics of the 28 node-classification benchmarks used in this work, spanning diverse domains, feature specifications and structural properties.

| Dataset | #Nodes | #Edges | #Features | #Classes | Feature Type | Train/Val/Test Ratios (%) | #Train Nodes | #Val Nodes | #Test Nodes | Homophily Ratio | Avg. Degree |
|---|---|---|---|---|---|---|---|---|---|---|---|
| AirBrazil | 131 | 2137 | 131 | 4 | One-hot | 61.07/19.08/19.85 | 80 | 25 | 26 | 0.4307 | 16.31 |
| Texas | 183 | 741 | 1703 | 5 | Binary Sparse | 47.54/31.69/20.22 | 87 | 58 | 37 | 0.0609 | 4.05 |
| Cornell | 183 | 737 | 1703 | 5 | Binary Sparse | 47.54/32.24/20.22 | 87 | 59 | 37 | 0.1227 | 4.03 |
| Wisconsin | 251 | 1151 | 1703 | 5 | Binary Sparse | 47.81/31.87/20.32 | 120 | 80 | 51 | 0.1778 | 4.59 |
| AirEU | 399 | 12385 | 399 | 4 | One-hot | 20.05/39.85/40.10 | 80 | 159 | 160 | 0.4046 | 31.04 |
| Chameleon | 890 | 18598 | 2325 | 5 | Binary Sparse | 45.96/32.25/21.80 | 409 | 287 | 194 | 0.2361 | 20.9 |
| AirUSA | 1190 | 28388 | 1190 | 4 | One-hot | 6.72/46.64/46.64 | 80 | 555 | 555 | 0.6978 | 23.86 |
| Squirrel | 2223 | 96219 | 2089 | 5 | Binary Sparse | 47.37/32.30/20.33 | 1053 | 718 | 452 | 0.2072 | 43.28 |
| Wiki | 2405 | 25597 | 4973 | 17 | Unsigned Dense | 14.14/42.91/42.95 | 340 | 1032 | 1033 | 0.6097 | 10.64 |
| Cora | 2708 | 13264 | 1433 | 7 | Sparse | 5.17/18.46/36.93 | 140 | 500 | 1000 | 0.8100 | 4.9 |
| Citeseer | 3327 | 12431 | 3703 | 6 | Sparse | 3.61/15.03/30.06 | 120 | 500 | 1000 | 0.7355 | 3.74 |
| BlogCatalog | 5196 | 348682 | 8189 | 6 | Binary Sparse | 2.31/48.85/48.85 | 120 | 2538 | 2538 | 0.4011 | 67.11 |
| Actor | 7600 | 60918 | 932 | 5 | Binary Sparse | 48.00/32.00/20.00 | 3648 | 2432 | 1520 | 0.2167 | 8.02 |
| AmzPhoto | 7650 | 245812 | 745 | 8 | Binary Dense | 2.09/48.95/48.95 | 160 | 3745 | 3745 | 0.8272 | 32.13 |
| Minesweeper | 10000 | 88804 | 7 | 2 | One-hot | 50.00/25.00/25.00 | 5000 | 2500 | 2500 | 0.6828 | 8.88 |
| WikiCS | 11701 | 442907 | 300 | 10 | Signed Dense | 4.96/15.12/49.97 | 580 | 1769 | 5847 | 0.6543 | 37.85 |
| Tolokers | 11758 | 1049758 | 10 | 2 | Unsigned Dense | 50.00/25.00/25.00 | 5879 | 2939 | 2940 | 0.5945 | 89.28 |
| AmzComputer | 13752 | 505474 | 767 | 10 | Binary Dense | 1.45/49.27/49.27 | 200 | 6776 | 6776 | 0.7772 | 36.76 |
| DBLP | 17716 | 123450 | 1639 | 4 | Binary Sparse | 0.45/49.77/49.77 | 80 | 8818 | 8818 | 0.8279 | 6.97 |
| CoCS | 18333 | 182121 | 6805 | 15 | Sparse | 1.64/49.18/49.18 | 300 | 9016 | 9017 | 0.8081 | 9.93 |
| Pubmed | 19717 | 108365 | 500 | 3 | Binary Dense | 0.30/2.54/5.07 | 60 | 500 | 1000 | 0.8024 | 5.5 |
| FullCora | 19793 | 146635 | 8710 | 70 | Signed Dense | 7.07/46.46/46.47 | 1400 | 9196 | 9197 | 0.5670 | 7.41 |
| Roman Empire | 22662 | 88516 | 300 | 18 | Signed Dense | 50.00/25.00/25.00 | 11331 | 5665 | 5666 | 0.0469 | 3.91 |
| Amazon Ratings | 24492 | 210592 | 300 | 5 | Signed Dense | 50.00/25.00/25.00 | 12246 | 6123 | 6123 | 0.3804 | 8.6 |
| Deezer | 28281 | 213785 | 128 | 2 | Unsigned Dense | 0.14/49.93/49.93 | 40 | 14120 | 14121 | 0.5251 | 7.56 |
| CoPhysics | 34493 | 530417 | 8415 | 5 | Sparse | 0.29/49.85/49.86 | 100 | 17196 | 17197 | 0.9314 | 15.38 |
| Questions | 48921 | 356001 | 301 | 2 | Unsigned Dense | 50.00/25.00/25.00 | 24460 | 12230 | 12231 | 0.8396 | 7.28 |
| OGBN-Arxiv | 169343 | 2484941 | 128 | 40 | Signed Dense | 53.70/17.60/28.70 | 90941 | 29799 | 48603 | 0.6542 | 14.67 |

*Table 19.* Performance(%) comparison against its predictors, and GraphAny across 28 datasets. **1st**, 2nd best performance is highlighted. Δ rows report relative improvements over each comparator. RGVT consistently outperforms both predictors and GraphAny variants, demonstrating robust generalization across diverse datasets.

| Method | Actor | AirBrazil | AirEU | AirUS | AmzComp | AmzPhoto | AmzRatings | BlogCatalog |
|---|---|---|---|---|---|---|---|---|
| **#Train Nodes** | 3648 | 80 | 80 | 80 | 200 | 160 | 12246 | 120 |
| Linear | **35.38**$_{\pm 0.33}$ | 25.38$_{\pm 6.44}$ | 23.87$_{\pm 2.36}$ | 26.16$_{\pm 1.27}$ | 70.42$_{\pm 0.45}$ | 77.44$_{\pm 0.72}$ | 37.99$_{\pm 0.10}$ | 70.80$_{\pm 0.49}$ |
| MLP | 35.32$_{\pm 0.38}$ | 25.38$_{\pm 4.39}$ | 26.62$_{\pm 4.11}$ | 25.37$_{\pm 1.22}$ | 69.73$_{\pm 0.29}$ | 78.64$_{\pm 0.63}$ | 45.37$_{\pm 0.48}$ | 75.60$_{\pm 0.59}$ |
| GraphAny (Wiscon) | 30.68$_{\pm 2.06}$ | 42.31$_{\pm 0.00}$ | 43.00$_{\pm 1.28}$ | 45.16$_{\pm 0.21}$ | 81.31$_{\pm 0.35}$ | 91.50$_{\pm 0.14}$ | 42.25$_{\pm 0.61}$ | 77.49$_{\pm 1.29}$ |
| GraphAny (Cora) | 29.28$_{\pm 1.62}$ | 45.38$_{\pm 1.72}$ | 41.87$_{\pm 1.59}$ | 45.52$_{\pm 0.27}$ | 82.42$_{\pm 0.11}$ | 91.10$_{\pm 0.11}$ | 42.59$_{\pm 0.38}$ | 73.48$_{\pm 0.08}$ |
| GraphAny (Arxiv) | 29.70$_{\pm 1.11}$ | 42.31$_{\pm 0.00}$ | 43.12$_{\pm 0.76}$ | 46.31$_{\pm 0.49}$ | 82.17$_{\pm 0.10}$ | 91.83$_{\pm 0.05}$ | 42.51$_{\pm 0.19}$ | 75.18$_{\pm 1.19}$ |
| GraphAny (Best) | 30.68$_{\pm 2.06}$ | 45.38$_{\pm 1.72}$ | 43.12$_{\pm 0.76}$ | 46.31$_{\pm 0.49}$ | 82.42$_{\pm 0.11}$ | 91.83$_{\pm 0.05}$ | 42.59$_{\pm 0.38}$ | 77.49$_{\pm 1.29}$ |
| **RGVT**+ Linear (Arxiv) | 33.87$_{\pm 0.50}$ | 52.31$_{\pm 6.99}$ | 46.25$_{\pm 1.93}$ | **57.08**$_{\pm 3.38}$ | **84.30**$_{\pm 0.23}$ | 92.02$_{\pm 0.30}$ | 42.98$_{\pm 0.13}$ | 80.75$_{\pm 3.08}$ |
| Δ Linear | ↓ **4.27**% | ↑ **106.11**% | ↑ **93.76**% | ↑ **118.20**% | ↑ **19.71**% | ↑ **18.83**% | ↑ **13.14**% | ↑ **14.05**% |
| Δ GraphAny (Best) | ↑ **10.40**% | ↑ **15.27**% | ↑ **7.26**% | ↑ **23.26**% | ↑ **2.28**% | ↑ **0.21**% | ↑ **0.92**% | ↑ **4.21**% |
| **RGVT**+ MLP (Arxiv) | 34.00$_{\pm 1.14}$ | **66.15**$_{\pm 10.67}$ | **47.00**$_{\pm 2.70}$ | 56.79$_{\pm 2.26}$ | 83.72$_{\pm 0.46}$ | **92.15**$_{\pm 0.36}$ | **48.61**$_{\pm 1.53}$ | **84.61**$_{\pm 1.59}$ |
| Δ MLP | ↓ **3.74**% | ↑ **160.64**% | ↑ **76.56**% | ↑ **123.85**% | ↑ **20.06**% | ↑ **17.18**% | ↑ **7.14**% | ↑ **11.92**% |
| Δ GraphAny (Best) | ↑ **10.82**% | ↑ **45.77**% | ↑ **9.00**% | ↑ **22.63**% | ↑ **1.58**% | ↑ **0.35**% | ↑ **14.13**% | ↑ **9.19**% |

| Method | Chameleon | Citeseer | CoCS | CoPhysics | Cora | Cornell | DBLP | Deezer |
|---|---|---|---|---|---|---|---|---|
| **#Train Nodes** | 409 | 120 | 300 | 100 | 140 | 87 | 80 | 40 |
| Linear | 36.39$_{\pm 1.95}$ | 49.22$_{\pm 0.28}$ | 85.60$_{\pm 0.13}$ | 82.21$_{\pm 0.10}$ | 48.60$_{\pm 1.76}$ | **74.05**$_{\pm 1.48}$ | 52.03$_{\pm 0.31}$ | 56.39$_{\pm 1.30}$ |
| MLP | 36.08$_{\pm 4.30}$ | 49.74$_{\pm 0.94}$ | 87.23$_{\pm 0.53}$ | 86.54$_{\pm 0.89}$ | 52.56$_{\pm 0.72}$ | 71.89$_{\pm 3.08}$ | 53.23$_{\pm 0.32}$ | 55.40$_{\pm 1.15}$ |
| GraphAny (Wiscon) | 31.14$_{\pm 6.89}$ | 68.20$_{\pm 0.14}$ | 89.42$_{\pm 0.10}$ | 91.92$_{\pm 0.12}$ | 76.90$_{\pm 0.45}$ | 65.95$_{\pm 2.42}$ | 70.59$_{\pm 0.82}$ | 54.08$_{\pm 0.08}$ |
| GraphAny (Cora) | 30.87$_{\pm 6.28}$ | 68.12$_{\pm 0.29}$ | 88.77$_{\pm 0.03}$ | 92.25$_{\pm 0.02}$ | 76.84$_{\pm 0.05}$ | 63.24$_{\pm 2.42}$ | 72.18$_{\pm 0.05}$ | 53.98$_{\pm 0.04}$ |
| GraphAny (Arxiv) | 30.41$_{\pm 6.12}$ | 67.82$_{\pm 0.54}$ | 89.00$_{\pm 0.17}$ | 92.20$_{\pm 0.02}$ | 77.70$_{\pm 0.20}$ | 64.86$_{\pm 0.00}$ | 71.47$_{\pm 0.18}$ | 54.20$_{\pm 0.13}$ |
| GraphAny (Best) | 31.14$_{\pm 6.89}$ | 68.20$_{\pm 0.14}$ | 89.42$_{\pm 0.10}$ | 92.25$_{\pm 0.02}$ | 77.70$_{\pm 0.20}$ | 65.95$_{\pm 2.42}$ | 72.18$_{\pm 0.05}$ | 54.20$_{\pm 0.13}$ |
| **RGVT**+ Linear (Arxiv) | 41.34$_{\pm 0.99}$ | **71.96**$_{\pm 0.38}$ | 91.32$_{\pm 0.42}$ | **92.71**$_{\pm 0.35}$ | **81.32**$_{\pm 0.64}$ | 73.51$_{\pm 4.44}$ | **81.29**$_{\pm 1.01}$ | **57.79**$_{\pm 0.29}$ |
| Δ Linear | ↑ **13.60**% | ↑ **46.20**% | ↑ **6.68**% | ↑ **12.77**% | ↑ **67.33**% | ↓ **0.73**% | ↑ **56.24**% | ↑ **2.48**% |
| Δ GraphAny (Best) | ↑ **32.76**% | ↑ **5.51**% | ↑ **2.12**% | ↑ **0.50**% | ↑ **4.66**% | ↑ **11.46**% | ↑ **12.62**% | ↑ **6.62**% |
| **RGVT**+ MLP (Arxiv) | **44.74**$_{\pm 2.17}$ | 70.02$_{\pm 1.34}$ | **92.10**$_{\pm 0.14}$ | 92.62$_{\pm 0.60}$ | 81.18$_{\pm 1.15}$ | **74.05**$_{\pm 1.48}$ | 81.08$_{\pm 1.22}$ | 57.58$_{\pm 0.61}$ |
| Δ MLP | ↑ **24.00**% | ↑ **40.77**% | ↑ **5.58**% | ↑ **7.03**% | ↑ **54.45**% | ↑ **3.00**% | ↑ **52.32**% | ↑ **3.94**% |
| Δ GraphAny (Best) | ↑ **43.67**% | ↑ **2.67**% | ↑ **3.00**% | ↑ **0.40**% | ↑ **4.48**% | ↑ **12.28**% | ↑ **12.33**% | ↑ **6.24**% |

| Method | Minesweeper | Pubmed | Questions | Roman | Squirrel | Texas | Tolokers | Wiki |
|---|---|---|---|---|---|---|---|---|
| **#Train Nodes** | 5000 | 60 | 24460 | 11331 | 1053 | 87 | 5879 | 340 |
| Linear | 80.00$_{\pm 0.00}$ | 68.68$_{\pm 0.90}$ | 97.05$_{\pm 0.00}$ | 63.19$_{\pm 0.04}$ | 30.62$_{\pm 0.51}$ | 78.38$_{\pm 0.00}$ | 78.12$_{\pm 0.11}$ | 71.13$_{\pm 1.14}$ |
| MLP | 80.00$_{\pm 0.00}$ | 70.28$_{\pm 1.00}$ | 97.12$_{\pm 0.09}$ | 64.43$_{\pm 0.28}$ | 34.73$_{\pm 2.40}$ | **79.46**$_{\pm 1.48}$ | 78.28$_{\pm 0.05}$ | 72.62$_{\pm 1.43}$ |
| GraphAny (Wiscon) | 80.26$_{\pm 0.12}$ | 77.50$_{\pm 0.23}$ | 97.11$_{\pm 0.01}$ | 63.88$_{\pm 0.20}$ | 24.97$_{\pm 0.82}$ | 74.89$_{\pm 2.63}$ | 78.19$_{\pm 0.02}$ | 57.75$_{\pm 0.46}$ |
| GraphAny (Cora) | **80.42**$_{\pm 0.13}$ | 76.52$_{\pm 0.04}$ | 97.08$_{\pm 0.01}$ | 63.59$_{\pm 0.62}$ | 24.60$_{\pm 0.86}$ | 71.56$_{\pm 1.93}$ | 78.17$_{\pm 0.06}$ | 57.81$_{\pm 0.10}$ |
| GraphAny (Arxiv) | 80.34$_{\pm 0.15}$ | 76.68$_{\pm 0.08}$ | 97.10$_{\pm 0.01}$ | 63.74$_{\pm 0.62}$ | 26.01$_{\pm 0.44}$ | 71.52$_{\pm 5.78}$ | 78.14$_{\pm 0.03}$ | 61.06$_{\pm 0.42}$ |
| GraphAny (Best) | **80.42**$_{\pm 0.13}$ | 77.50$_{\pm 0.23}$ | 97.11$_{\pm 0.01}$ | 63.88$_{\pm 0.20}$ | 26.01$_{\pm 0.44}$ | 74.89$_{\pm 2.63}$ | 78.19$_{\pm 0.02}$ | 61.06$_{\pm 0.42}$ |
| **RGVT**+ Linear (Arxiv) | 79.77$_{\pm 0.27}$ | **79.02**$_{\pm 1.10}$ | **97.13**$_{\pm 0.02}$ | 67.71$_{\pm 0.87}$ | 39.38$_{\pm 0.81}$ | 75.14$_{\pm 2.96}$ | 78.10$_{\pm 0.29}$ | 72.76$_{\pm 0.98}$ |
| Δ Linear | ↓ **0.29**% | ↑ **15.06**% | ↑ **0.08**% | ↑ **7.15**% | ↑ **28.61**% | ↓ **4.13**% | ↓ **0.03**% | ↑ **2.29**% |
| Δ GraphAny (Best) | ↓ **0.81**% | ↑ **1.96**% | ↑ **0.02**% | ↑ **6.00**% | ↑ **51.40**% | ↑ **0.33**% | ↓ **0.12**% | ↑ **19.16**% |
| **RGVT**+ MLP (Arxiv) | 79.98$_{\pm 0.18}$ | 78.72$_{\pm 0.70}$ | **97.13**$_{\pm 0.02}$ | **70.18**$_{\pm 0.28}$ | **41.77**$_{\pm 2.06}$ | 78.92$_{\pm 1.21}$ | **79.23**$_{\pm 0.42}$ | **74.54**$_{\pm 0.76}$ |
| Δ MLP | ↓ **0.02**% | ↑ **12.01**% | ↑ **0.01**% | ↑ **8.92**% | ↑ **20.27**% | ↓ **0.68**% | ↑ **1.21**% | ↑ **2.64**% |
| Δ GraphAny (Best) | ↓ **0.55**% | ↑ **1.57**% | ↑ **0.02**% | ↑ **9.86**% | ↑ **60.59**% | ↑ **5.38**% | ↑ **1.33**% | ↑ **22.08**% |

| Method | Wisconsin | WikiCS | OGBN-Arxiv | FullCora | **Total Avg.** |
|---|---|---|---|---|---|
| **#Train Nodes** | 120 | 580 | 90941 | 1400 | - |
| Linear | **79.22**$_{\pm 1.07}$ | 72.50$_{\pm 0.04}$ | 52.44$_{\pm 0.04}$ | 40.32$_{\pm 0.20}$ | 59.41$_{\pm 0.84}$ |
| MLP | 74.90$_{\pm 2.15}$ | 72.08$_{\pm 0.19}$ | 53.80$_{\pm 0.14}$ | 39.75$_{\pm 0.44}$ | 60.43$_{\pm 1.20}$ |
| GraphAny (Wiscon) | 66.28$_{\pm 4.02}$ | 74.61$_{\pm 0.80}$ | 57.77$_{\pm 0.45}$ | 57.12$_{\pm 0.21}$ | 64.72$_{\pm 0.96}$ |
| GraphAny (Cora) | 61.96$_{\pm 6.30}$ | 74.91$_{\pm 0.92}$ | 58.58$_{\pm 0.10}$ | 57.21$_{\pm 0.09}$ | 64.30$_{\pm 0.94}$ |
| GraphAny (Arxiv) | 64.31$_{\pm 5.08}$ | 75.53$_{\pm 0.78}$ | 58.63$_{\pm 0.14}$ | 58.08$_{\pm 0.08}$ | 64.71$_{\pm 0.89}$ |
| GraphAny (Best) | 66.28$_{\pm 4.02}$ | 75.53$_{\pm 0.78}$ | 58.63$_{\pm 0.14}$ | 58.08$_{\pm 0.08}$ | 65.30$_{\pm 0.93}$ |
| **RGVT**+ Linear (Arxiv) | 76.86$_{\pm 1.64}$ | **79.59**$_{\pm 0.16}$ | 70.14$_{\pm 0.28}$ | **64.35**$_{\pm 0.77}$ | 70.03$_{\pm 1.26}$ |
| Δ Linear | ↓ **2.98**% | ↑ **9.78**% | ↑ **33.75**% | ↑ **59.60**% | ↑ **17.88**% |
| Δ GraphAny (Best) | ↑ **15.96**% | ↑ **5.38**% | ↑ **19.63**% | ↑ **10.80**% | ↑ **7.24**% |
| **RGVT**+ MLP (Arxiv) | 71.76$_{\pm 1.75}$ | **79.59**$_{\pm 0.19}$ | **71.11**$_{\pm 0.28}$ | 62.35$_{\pm 2.22}$ | **71.13**$_{\pm 1.41}$ |
| Δ MLP | ↓ **4.19**% | ↑ **10.42**% | ↑ **32.17**% | ↑ **56.86**% | ↑ **17.71**% |
| Δ GraphAny (Best) | ↑ **8.27**% | ↑ **5.38**% | ↑ **21.29**% | ↑ **7.35**% | ↑ **8.93**% |

*Table 20.* Performance (%) comparison with 12 individually trained GNN models across 28 datasets. 1st, 2nd, 3rd best results are highlighted. RGVT achieves the best performance on average, ranking 1st with the MLP predictor and 2nd with the linear predictor.

| Method | Actor | AirBrazil | AirEU | AirUSA | AmzComp | AmzPhoto | AmzRatings | BlogCatalog |
|---|---|---|---|---|---|---|---|---|
| **#Train Nodes** | 3648 | 80 | 80 | 80 | 200 | 160 | 12246 | 120 |
| GIN | $27.75_{\pm1.52}$ | $49.23_{\pm10.67}$ | $40.50_{\pm3.11}$ | $51.32_{\pm1.47}$ | $36.26_{\pm1.10}$ | $28.02_{\pm6.08}$ | $46.01_{\pm2.73}$ | $17.00_{\pm0.51}$ |
| GAT | $28.57_{\pm0.82}$ | $44.62_{\pm14.54}$ | $40.00_{\pm1.93}$ | $51.24_{\pm2.90}$ | $84.43_{\pm1.28}$ | $89.40_{\pm0.71}$ | $48.99_{\pm0.56}$ | $55.16_{\pm4.77}$ |
| GATv2 | $29.80_{\pm0.46}$ | $54.62_{\pm8.34}$ | $40.12_{\pm2.23}$ | $47.14_{\pm2.30}$ | $84.51_{\pm0.71}$ | $89.54_{\pm0.39}$ | $49.57_{\pm0.67}$ | $57.94_{\pm3.22}$ |
| S$^2$GC | $28.93_{\pm0.21}$ | $54.62_{\pm3.22}$ | $41.12_{\pm1.20}$ | $50.92_{\pm0.49}$ | $84.70_{\pm0.10}$ | $92.07_{\pm0.27}$ | $42.05_{\pm0.20}$ | $80.09_{\pm0.06}$ |
| SGC | $29.70_{\pm0.08}$ | $71.54_{\pm10.39}$ | $42.62_{\pm0.81}$ | $54.49_{\pm1.45}$ | $84.39_{\pm0.08}$ | $92.26_{\pm0.16}$ | $43.07_{\pm0.08}$ | $71.06_{\pm0.04}$ |
| JKNet | $30.53_{\pm0.36}$ | $56.92_{\pm15.72}$ | $41.88_{\pm0.99}$ | $53.26_{\pm2.71}$ | $84.96_{\pm0.30}$ | $91.41_{\pm0.60}$ | $48.93_{\pm0.40}$ | $72.11_{\pm0.31}$ |
| APPNP | $31.72_{\pm0.54}$ | $36.92_{\pm3.44}$ | $31.25_{\pm0.99}$ | $47.50_{\pm0.52}$ | $84.39_{\pm0.18}$ | $91.87_{\pm0.21}$ | $50.63_{\pm0.44}$ | $86.00_{\pm0.52}$ |
| GCN | $30.37_{\pm0.44}$ | $63.08_{\pm3.44}$ | $41.50_{\pm0.84}$ | $53.30_{\pm1.18}$ | $84.69_{\pm0.18}$ | $91.71_{\pm0.18}$ | $49.17_{\pm0.43}$ | $71.75_{\pm0.24}$ |
| GCNII | $34.22_{\pm0.35}$ | $46.15_{\pm5.44}$ | $41.00_{\pm9.45}$ | $51.24_{\pm1.69}$ | $84.81_{\pm0.23}$ | $91.39_{\pm0.30}$ | $50.37_{\pm0.17}$ | $87.43_{\pm0.69}$ |
| SAGE | $36.37_{\pm0.93}$ | $34.62_{\pm7.20}$ | $37.50_{\pm4.35}$ | $48.18_{\pm1.72}$ | $83.65_{\pm0.19}$ | $90.74_{\pm0.42}$ | $50.86_{\pm0.41}$ | $80.27_{\pm3.55}$ |
| GPRGNN | $34.51_{\pm0.31}$ | $31.54_{\pm4.21}$ | $34.37_{\pm1.65}$ | $51.57_{\pm0.59}$ | $85.55_{\pm0.31}$ | $92.81_{\pm0.18}$ | $48.74_{\pm0.28}$ | $88.34_{\pm0.27}$ |
| UniMP | $35.97_{\pm1.39}$ | $44.62_{\pm9.65}$ | $39.25_{\pm3.14}$ | $47.64_{\pm1.46}$ | $84.06_{\pm0.70}$ | $92.15_{\pm0.25}$ | $51.30_{\pm0.77}$ | $80.70_{\pm3.24}$ |
| **RGVT + Linear** | $33.87_{\pm0.50}$ | $52.31_{\pm6.99}$ | $46.25_{\pm1.93}$ | $57.08_{\pm3.38}$ | $84.30_{\pm0.23}$ | $92.02_{\pm0.30}$ | $42.98_{\pm0.13}$ | $80.75_{\pm3.08}$ |
| **RGVT + MLP** | $34.00_{\pm1.14}$ | $66.15_{\pm10.67}$ | $47.00_{\pm2.70}$ | $56.79_{\pm2.26}$ | $83.72_{\pm0.46}$ | $92.15_{\pm0.36}$ | $48.61_{\pm1.53}$ | $84.61_{\pm1.59}$ |

| Method | Chameleon | Citeseer | CoCS | CoPhysics | Cora | Cornell | DBLP | Deezer |
|---|---|---|---|---|---|---|---|---|
| **#Train Nodes** | 409 | 120 | 300 | 100 | 140 | 87 | 80 | 40 |
| GIN | $35.15_{\pm3.14}$ | $47.48_{\pm10.03}$ | $84.20_{\pm0.79}$ | $85.98_{\pm4.54}$ | $70.06_{\pm3.28}$ | $37.84_{\pm2.70}$ | $71.68_{\pm1.56}$ | $55.33_{\pm0.51}$ |
| GAT | $39.18_{\pm2.70}$ | $69.46_{\pm1.36}$ | $90.27_{\pm0.57}$ | $91.22_{\pm2.06}$ | $79.06_{\pm1.04}$ | $40.00_{\pm2.26}$ | $79.07_{\pm0.78}$ | $55.57_{\pm0.34}$ |
| GATv2 | $38.76_{\pm0.92}$ | $68.60_{\pm0.62}$ | $88.71_{\pm0.98}$ | $89.43_{\pm3.15}$ | $79.98_{\pm1.02}$ | $37.84_{\pm3.82}$ | $78.28_{\pm1.92}$ | $55.69_{\pm0.40}$ |
| S$^2$GC | $41.75_{\pm0.82}$ | $67.32_{\pm0.37}$ | $90.75_{\pm0.03}$ | $91.69_{\pm0.04}$ | $78.40_{\pm0.71}$ | $43.78_{\pm2.26}$ | $70.27_{\pm0.07}$ | $54.48_{\pm0.36}$ |
| SGC | $43.09_{\pm0.28}$ | $67.98_{\pm0.58}$ | $90.28_{\pm0.03}$ | $92.70_{\pm0.03}$ | $78.46_{\pm0.83}$ | $43.78_{\pm1.21}$ | $74.40_{\pm0.05}$ | $54.56_{\pm0.52}$ |
| JKNet | $41.75_{\pm1.09}$ | $68.32_{\pm1.68}$ | $90.28_{\pm0.22}$ | $92.65_{\pm0.15}$ | $81.96_{\pm0.77}$ | $40.54_{\pm0.00}$ | $75.96_{\pm1.32}$ | $56.30_{\pm0.43}$ |
| APPNP | $40.10_{\pm0.43}$ | $68.68_{\pm0.43}$ | $91.13_{\pm0.09}$ | $93.45_{\pm0.07}$ | $81.38_{\pm0.95}$ | $47.03_{\pm6.22}$ | $77.75_{\pm0.28}$ | $55.97_{\pm0.53}$ |
| GCN | $39.90_{\pm1.19}$ | $69.22_{\pm0.62}$ | $90.60_{\pm0.09}$ | $92.85_{\pm0.24}$ | $81.26_{\pm0.77}$ | $45.41_{\pm3.52}$ | $79.52_{\pm1.03}$ | $57.09_{\pm0.62}$ |
| GCNII | $41.55_{\pm2.54}$ | $63.76_{\pm0.92}$ | $91.07_{\pm0.10}$ | $92.76_{\pm0.25}$ | $76.42_{\pm2.62}$ | $37.84_{\pm2.70}$ | $78.44_{\pm1.62}$ | $55.82_{\pm0.43}$ |
| SAGE | $40.31_{\pm1.43}$ | $66.96_{\pm2.24}$ | $90.55_{\pm0.09}$ | $92.10_{\pm0.54}$ | $78.96_{\pm0.72}$ | $67.03_{\pm3.52}$ | $73.47_{\pm0.37}$ | $56.20_{\pm1.07}$ |
| GPRGNN | $40.93_{\pm1.24}$ | $70.30_{\pm0.62}$ | $91.83_{\pm0.21}$ | $93.20_{\pm0.10}$ | $82.08_{\pm0.66}$ | $68.65_{\pm3.63}$ | $78.41_{\pm0.15}$ | $56.89_{\pm0.26}$ |
| UniMP | $41.34_{\pm3.16}$ | $68.56_{\pm0.19}$ | $89.84_{\pm0.39}$ | $91.53_{\pm0.76}$ | $79.18_{\pm1.13}$ | $62.16_{\pm4.27}$ | $73.44_{\pm0.39}$ | $56.32_{\pm0.88}$ |
| **RGVT + Linear** | $41.34_{\pm0.99}$ | $71.96_{\pm0.38}$ | $91.32_{\pm0.42}$ | $92.71_{\pm0.35}$ | $81.32_{\pm0.64}$ | $73.51_{\pm4.44}$ | $81.29_{\pm1.01}$ | $57.79_{\pm0.29}$ |
| **RGVT + MLP** | $44.74_{\pm2.17}$ | $70.02_{\pm1.34}$ | $92.10_{\pm0.14}$ | $92.62_{\pm0.60}$ | $81.18_{\pm1.15}$ | $74.05_{\pm1.48}$ | $81.08_{\pm1.22}$ | $57.58_{\pm0.61}$ |

*Table 21.* Performance (%) comparison with 12 individually trained GNN models across 28 datasets. 1st, 2nd, 3rd best results are highlighted. RGVT achieves the best performance on average, ranking 1st with the MLP predictor and 2nd with the linear predictor.

| Method | Minesweeper | Pubmed | Questions | Roman | Squirrel | Texas | Tolokers | Wiki |
|---|---|---|---|---|---|---|---|---|
| **#Train Nodes** | 5000 | 60 | 24460 | 11331 | 1053 | 87 | 5879 | 340 |
| GIN | $80.18_{\pm 0.26}$ | $70.22_{\pm 6.97}$ | $97.02_{\pm 0.01}$ | $51.41_{\pm 1.83}$ | $34.73_{\pm 0.00}$ | $68.11_{\pm 5.20}$ | $78.16_{\pm 0.02}$ | $41.28_{\pm 15.08}$ |
| GAT | $81.70_{\pm 0.53}$ | $76.26_{\pm 0.70}$ | $97.06_{\pm 0.02}$ | $43.42_{\pm 0.18}$ | $36.46_{\pm 0.50}$ | $63.78_{\pm 6.22}$ | $81.67_{\pm 0.18}$ | $58.64_{\pm 1.93}$ |
| GATv2 | $82.60_{\pm 0.78}$ | $75.56_{\pm 1.28}$ | $97.04_{\pm 0.02}$ | $59.97_{\pm 0.82}$ | $38.41_{\pm 2.13}$ | $62.16_{\pm 6.34}$ | $80.73_{\pm 1.55}$ | $54.52_{\pm 2.90}$ |
| $S^2$GC | $81.24_{\pm 0.50}$ | $76.68_{\pm 0.04}$ | $97.06_{\pm 0.00}$ | $48.73_{\pm 0.11}$ | $37.52_{\pm 0.25}$ | $62.16_{\pm 3.31}$ | $78.65_{\pm 0.06}$ | $70.94_{\pm 1.00}$ |
| SGC | $81.52_{\pm 0.30}$ | $77.54_{\pm 0.05}$ | $97.09_{\pm 0.00}$ | $43.71_{\pm 0.06}$ | $38.14_{\pm 0.20}$ | $55.68_{\pm 1.48}$ | $78.58_{\pm 0.04}$ | $70.47_{\pm 0.45}$ |
| JKNet | $81.79_{\pm 0.31}$ | $75.70_{\pm 0.57}$ | $97.06_{\pm 0.03}$ | $53.76_{\pm 4.05}$ | $34.34_{\pm 1.83}$ | $63.24_{\pm 6.22}$ | $80.97_{\pm 0.28}$ | $70.15_{\pm 0.94}$ |
| APPNP | $81.47_{\pm 0.11}$ | $76.08_{\pm 0.24}$ | $97.08_{\pm 0.02}$ | $60.40_{\pm 0.31}$ | $35.44_{\pm 0.77}$ | $67.57_{\pm 1.91}$ | $78.14_{\pm 0.04}$ | $73.90_{\pm 0.85}$ |
| GCN | $80.16_{\pm 0.26}$ | $76.90_{\pm 0.42}$ | $97.06_{\pm 0.02}$ | $45.33_{\pm 0.25}$ | $38.54_{\pm 1.64}$ | $65.41_{\pm 2.26}$ | $80.60_{\pm 0.12}$ | $69.24_{\pm 0.76}$ |
| GCNII | $86.04_{\pm 0.41}$ | $74.76_{\pm 2.91}$ | $97.18_{\pm 0.06}$ | $75.80_{\pm 0.41}$ | $35.97_{\pm 0.58}$ | $62.16_{\pm 2.70}$ | $81.79_{\pm 0.66}$ | $75.35_{\pm 0.66}$ |
| SAGE | $87.06_{\pm 0.30}$ | $76.68_{\pm 0.91}$ | $97.14_{\pm 0.04}$ | $77.69_{\pm 0.48}$ | $38.19_{\pm 1.12}$ | $82.16_{\pm 6.51}$ | $80.66_{\pm 0.29}$ | $72.53_{\pm 1.22}$ |
| GPRGNN | $84.10_{\pm 0.15}$ | $77.44_{\pm 0.44}$ | $97.21_{\pm 0.02}$ | $74.24_{\pm 0.30}$ | $35.80_{\pm 0.96}$ | $73.51_{\pm 6.16}$ | $78.41_{\pm 0.06}$ | $74.60_{\pm 0.37}$ |
| UniMP | $88.25_{\pm 0.17}$ | $76.46_{\pm 0.65}$ | $97.17_{\pm 0.03}$ | $80.31_{\pm 0.56}$ | $40.18_{\pm 1.97}$ | $73.51_{\pm 3.52}$ | $81.46_{\pm 0.38}$ | $72.93_{\pm 1.12}$ |
| **RGVT** + Linear | $79.77_{\pm 0.27}$ | $79.02_{\pm 1.10}$ | $97.13_{\pm 0.02}$ | $67.71_{\pm 0.87}$ | $39.38_{\pm 0.81}$ | $75.14_{\pm 2.96}$ | $78.10_{\pm 0.29}$ | $72.76_{\pm 0.98}$ |
| **RGVT** + MLP | $79.98_{\pm 0.18}$ | $78.72_{\pm 0.70}$ | $97.13_{\pm 0.02}$ | $70.18_{\pm 0.28}$ | $41.77_{\pm 2.06}$ | $78.92_{\pm 1.21}$ | $79.23_{\pm 0.42}$ | $74.54_{\pm 0.76}$ |

| Method | Wisconsin | WikiCS | OGBN-Arxiv | FullCora | **Total Avg.** |
|---|---|---|---|---|---|
| **#Train Nodes** | 120 | 580 | 90941 | 1400 | - |
| GIN | $54.12_{\pm 5.30}$ | $56.85_{\pm 13.75}$ | $65.77_{\pm 1.08}$ | $57.80_{\pm 0.32}$ | $54.98_{\pm 3.70}$ |
| GAT | $51.37_{\pm 2.56}$ | $79.44_{\pm 0.40}$ | $71.89_{\pm 0.24}$ | $60.71_{\pm 0.16}$ | $63.88_{\pm 1.87}$ |
| GATv2 | $53.73_{\pm 3.56}$ | $79.78_{\pm 0.53}$ | $72.13_{\pm 0.11}$ | $60.80_{\pm 0.12}$ | $64.57_{\pm 1.83}$ |
| $S^2$GC | $53.73_{\pm 2.24}$ | $79.90_{\pm 0.05}$ | $69.17_{\pm 0.02}$ | $59.84_{\pm 0.03}$ | $65.31_{\pm 0.64}$ |
| SGC | $53.33_{\pm 3.51}$ | $79.40_{\pm 0.06}$ | $68.69_{\pm 0.03}$ | $59.86_{\pm 0.00}$ | $65.66_{\pm 0.82}$ |
| JKNet | $55.29_{\pm 1.64}$ | $79.46_{\pm 0.26}$ | $71.73_{\pm 0.24}$ | $61.99_{\pm 1.00}$ | $66.19_{\pm 1.59}$ |
| APPNP | $56.08_{\pm 1.07}$ | $78.80_{\pm 0.06}$ | $70.85_{\pm 0.15}$ | $63.65_{\pm 0.09}$ | $66.26_{\pm 0.77}$ |
| GCN | $53.33_{\pm 1.64}$ | $79.07_{\pm 0.24}$ | $71.19_{\pm 0.30}$ | $62.69_{\pm 0.13}$ | $66.46_{\pm 0.82}$ |
| GCNII | $61.57_{\pm 2.63}$ | $79.03_{\pm 0.17}$ | $72.05_{\pm 0.08}$ | $58.31_{\pm 0.44}$ | $67.30_{\pm 1.47}$ |
| SAGE | $62.75_{\pm 3.10}$ | $78.77_{\pm 0.48}$ | $71.22_{\pm 0.24}$ | $61.35_{\pm 0.09}$ | $68.36_{\pm 1.55}$ |
| GPRGNN | $64.71_{\pm 2.40}$ | $79.46_{\pm 0.31}$ | $69.45_{\pm 0.41}$ | $64.31_{\pm 0.11}$ | $68.68_{\pm 0.94}$ |
| UniMP | $65.88_{\pm 9.76}$ | $79.80_{\pm 0.13}$ | $71.78_{\pm 0.12}$ | $62.26_{\pm 0.11}$ | $68.86_{\pm 1.80}$ |
| **RGVT** + Linear | $76.86_{\pm 1.64}$ | $79.59_{\pm 0.16}$ | $70.14_{\pm 0.28}$ | $64.35_{\pm 0.77}$ | $70.03_{\pm 1.26}$ |
| **RGVT** + MLP | $71.76_{\pm 1.75}$ | $79.59_{\pm 0.19}$ | $71.11_{\pm 0.28}$ | $62.35_{\pm 2.22}$ | $71.13_{\pm 1.41}$ |

