# OpenReview forum: "View Space: Learning Representation across Arbitrary Graphs"
_ICML.cc/2026/Conference — ICML 2026 regular_

### Official Review · Reviewer_PaxS · 2026-02-26

**Soundness:** 3
**Presentation:** 4
**Significance:** 3
**Originality:** 4
**Overall Recommendation:** 5
**Confidence:** 4

**Summary:**

The paper introduces a new framework for learning representations across arbitrary graphs, tackling the challenge of feature heterogeneity. The authors propose  __View Space__, a new representational axis that allows both node permutation and feature permutation. To operate within this framework, the authors introduce GVT, which enables fully inductive learning of node representations without retraining on new graphs. Experiments show that RGVT outperforming previous models and further ablation shows effectiveness of each module

**Compliance With Llm Reviewing Policy:**

Affirmed.

**Final Justification:**

The paper studies an important problem in graph learning and proposes a novel and technically sound approach. The method is clearly presented and shows promising results.

My main concerns were about limited comparisons and evaluation. The rebuttal has adequately addressed these points by providing additional experimental results and clarifications, which improved my confidence in the work.

Overall, I find the paper to be solid and meaningful, and I support acceptance.

**Key Questions For Authors:**

1.Can GVT be applied to more generalized graph scenarios, such as graphs with rich edge features or graph classification tasks? Exploring this could demonstrate the method's broader applicability across different types of graphs and tasks.

2.Could you provide additional comparisons with the latest Graph Foundation Models and GNN baselines? Including more recent models in the comparison would help better highlight the strengths and effectiveness of GVT in relation to current state-of-the-art methods.

**Limitations:**

No. While the work does not seem to pose direct negative societal risks, the authors could improve the paper by explicitly discussing limitations such as generalizability to more complex or real-world graph scenarios, sensitivity to view finder, and potential challenges in applying the method to diverse graph types or tasks outside of the current scope.

**Strengths And Weaknesses:**

###Strength:

1.__Highly impactful problem in graph learning__ Learning representations across heterogeneous graphs is an ongoing challenge, and the author formalize the fully inductive node representation learning (FI-NRL) problem, where a model maps arbitrary graphs to node representations

2.__Novel Approach__ The view space concept provides a principled way of learning across heterogeneous graph datasets, offering a novel method of dealing with feature heterogeneity The view space concept is an original contribution that distinguishes this paper from prior work.

3.__Methods are technically solid__ Graph View Transformation (GVT) is built on solid theoretical foundations in graph representation learning, and the proposed method shows clear improvements in generalization and efficiency.

4.__The paper is well-written and clearly structured__. The overall narrative is coherent, and the methodology is presented in a step-by-step fashion, which is easy to follow. Figures help deepen the understanding of the view space and GVT, making the complex ideas more accessible.

###Weakness:

1.__Limited exploration of arbitrary graph types__ The paper primarily focuses on node features and does not explore how the method might be adapted for graphs with rich edge features. Additionally, its applicability to graph classification tasks remains underexplored and warrants further investigation.

2.__Lack of additional baseline comparisons__ The authors mainly compare to GraphAny and some classic algorithms (most of which are from 2021). A comparison with more recent GNN baselines and graph foundation models would better highlight the effectiveness of GVT.

3.__Sensitivity to view finder selection__ The choice of view finders significantly impacts model performance, requiring careful selection to ensure optimal results.

---

> ### Author Rebuttal · Authors · 2026-03-31
>
> Thank you for taking the time to review our work and for your constructive feedback. We sincerely appreciate your suggestions, and we hope the additional experiments and clarifications below address your concerns.
>
> ---
>
> **(W2, Q2) Additional comparisons with recent GNN baselines.**
>
> We appreciate this suggestion and agree that comparisons with more recent GNNs are important.
> Before presenting the new results, we would like to highlight our effort in constructing strong baselines. While several recent fully inductive works report larger gains over supervised GNNs, their baselines are often limited and less extensively tuned. Our best baseline, UniMP, achieves significantly stronger performance than those previously reported, with at least a +6.48\% average improvement ([Table A](https://shorturl.at/guiOy)). We believe this reflects the rigor of our evaluation and underscores the challenge of surpassing our baselines.
>
> Following your suggestion, we include four additional baselines especially with complementary perspectives: (i) DirGNN (Rossi et al., 2024) and LSGNN (Chen et al., 2023), designed to perform well on heterophilous graphs, and (ii) NodeFormer (Wu et al., 2022) and SGFormer (Wu et al., 2023), graph transformers that leverage global attention. As shown in [Figure A](https://shorturl.at/AFc6V), DirGNN, LSGNN, and SGFormer outperform our original baselines on average, with improvements mainly on heterophilous datasets. DirGNN and SGFormer even outperform RGVT with a linear predictor, but do not surpass RGVT with MLP predictor.
>
> Although the performance gains of RGVT over these baselines are modest, we emphasize that the primary advantage of fully inductive methods lies in their minimal adaptation overhead for unseen datasets. GNNs require extensive hyperparameter tuning for each dataset to achieve strong performance, and these baselines above took almost 3 days on 4 H200 GPUs. In contrast, RGVT can adapt across 27 datasets in 152 seconds (Appendix H), demonstrating a substantial efficiency advantage in practice.
>
> **(W2, Q2) Additional comparisons with latest Graph Foundation Models (GFMs).**
>
> We also appreciate the suggestion to compare with latest GFMs. Accordingly, we extend our evaluation to include TS-Net (Finkelshtein et al., 2025), TAG (Hayler et al., 2025), and NodePFN (Choi et al., 2026), which are also designed to generalize across arbitrary datasets without retraining. For fairness, we evaluate RGVT under their experimental pipelines and adopt their reported results.
>
> We note that these methods involve higher computational overhead. TAG uses an ensemble of 14 predictors, including 10 tabular foundation models with quadratic complexity, while NodePFN requires per-dataset selection over a preprocessing space (e.g., dimensionality reduction, smoothing steps, and ensemble size), totaling ~400 configurations. Despite this, RGVT consistently achieves better average rank and accuracy ([Tables B and C](https://shorturl.at/gpIyY)). These results indicate that RGVT achieves state-of-the-art performance while maintaining strong efficiency.
>
> **(W1, Q1) Limited exploration of graph types.**
>
> We agree that evaluating a broader range of graph types is important. In this work, our primary goal is to investigate whether learning solely on the view axis can capture transferable knowledge across arbitrary datasets while remaining competitive with conventional methods. To establish this core claim, we focus on node classification as the first step. We fully agree that extending the framework to more diverse settings, such as graphs with rich edge features, heterogeneous graphs, and other tasks (e.g., edge- and graph-level prediction), is an important direction for future work.
>
> **(W3) Sensitivity to view finder selection.**
>
> We appreciate this point and agree that performance depends on the choice of view finders. Our primary contribution, however, is not the choice of view finders, but the introduction of a new learning paradigm based on the view axis as an alternative to feature transformation. We view the selection of view finders as a flexible design space, similar to the choice of graph filters in GNNs. Our results show that even simple local aggregations can achieve competitive performance. We agree that developing more expressive and robust view finders is an important direction for future work.
>
> **Discussion of limitations.**
>
> Thank you for this helpful suggestion. We will include a dedicated discussion section in the final version to more clearly outline the limitations of our work. This will include aspects such as the current focus on node-level tasks, limited exploration of diverse graph types, and sensitivity to view finder selection.
>
> ---
>
> Thank you again for your time and effort in reviewing our work. We will incorporate the additional experiments and discussions above into the final version if given the opportunity. We hope our responses are helpful and that you find our work interesting.

---

> > ### Author Rebuttal · Reviewer_PaxS · 2026-04-02
> >
> > Most of my concerns have been addressed in the rebuttal. I appreciate the effort and the detailed responses.
> >
> > However, I am still unable to clearly locate the quantitative results of the newly added experiments (e.g., comparisons with recent GNNs and GFMs). I would appreciate it if the authors could point to the specific numbers or tables/figures for these results, as this is important for fully assessing the improvements.

---

> > > ### Author Response · Authors · 2026-04-02
> > >
> > > Thank you for your response. We’re glad to hear that most of your concerns have been addressed.
> > >
> > > Regarding the detailed results of the newly added experiments, we apologize for the lack of clarity. In our previous response, we included hyperlinks (via markdown) for each referenced *Figure* and *Table*. However, we realize that these hyperlinks may not have been sufficiently visible or easy to locate.
> > >
> > > Thank you for pointing this out. To address this, we explicitly provide direct links to the relevant results below:
> > >
> > > - **Table A**: Comparison between our GNN baselines and those in prior works
> > >
> > >      https://anonymous.4open.science/r/view-space-rebuttal-352B/Table_A.pdf
> > >
> > > - **Table E and Figure A**: Performance comparison against recent GNN baselines, along with a summary plot
> > >
> > >      https://anonymous.4open.science/r/view-space-rebuttal-352B/Table_E_Figure_A.pdf
> > >
> > > - **Tables B and C**: Performance comparison against recent GFMs
> > >
> > >      https://anonymous.4open.science/r/view-space-rebuttal-352B/Table_B_C.pdf
> > >
> > > Additionally, we sincerely appreciate your thoughtful feedback, which led to valuable discussions and additional experiments that we believe have further improved the clarity and overall quality of our work. We will incorporate these improvements into the final version if given the opportunity.
> > >
> > > We hope you found the review process engaging, and we sincerely thank you for your time and consideration.

---

### Official Review · Reviewer_bQrd · 2026-03-06

**Soundness:** 3
**Presentation:** 3
**Significance:** 3
**Originality:** 3
**Overall Recommendation:** 4
**Confidence:** 3

**Summary:**

This paper addresses the problem of fully inductive node representation learning across graphs with heterogeneous feature spaces, a key challenge toward graph foundation models. The core observation is that while feature spaces vary widely across graph datasets, all graphs share a universal property: connectivity. The paper formalizes this insight by introducing the view space, a novel representational axis induced by graph structure that is orthogonal to the conventional feature space and node space.

Building on this, the paper proposes Graph View Transformation (GVT), a parametric mapping that satisfies dual permutation equivariance with respect to both node and feature permutations. The paper further shows that linear GVT generalizes static GNN aggregations, while nonlinear GVT enables node-feature dynamic aggregation that goes beyond existing GNN expressivity. A practical architecture, Recurrent GVT (RGVT), is proposed, which applies a single GVT layer recurrently with shared parameters to decouple depth from parameterization. Pretrained on OGBN-Arxiv and evaluated on 27 downstream benchmarks, RGVT outperforms GraphAny by +8.93% on average and surpasses 12 individually tuned GNNs by at least +3.30%.

**Compliance With Llm Reviewing Policy:**

Affirmed.

**Key Questions For Authors:**

see weakness

**Limitations:**

yes

**Strengths And Weaknesses:**

Strengths:
1. The formalization of the view space as a structure-induced representational axis is conceptually elegant and well-grounded. The dual permutation equivariance requirements (R1 and R2) are clearly motivated. The connection between linear GVT and existing GNN aggregations (Lemma 5.1) provides a clean unification that situates the contribution within the broader landscape.
2. RGVT is evaluated on 28 benchmarks spanning diverse feature types, and consistently outperforms GraphAny and 12 individually tuned GNNs on average.

Weaknesses:
1. The view space amounts to stacking multiple propagated versions of the node-feature matrix using standard adjacency operators, which is conceptually similar to existing multi-scale or multi-hop feature propagation methods.
2. As shown in Table 12, RGVT's performance degrades noticeably when pretrained on smaller datasets such as Cora or Wisconsin compared to OGBN-Arxiv.
3. Although RGVT eliminates per-dataset retraining of the encoder, it still requires selecting the recurrent depth L for each downstream dataset via validation accuracy. And the paper does not discuss how sensitive performance is to suboptimal depth choices.
4. All experiments focus exclusively on node classification, and the paper explicitly defers edge-level and graph-level tasks to future work. This limits the scope of the claimed contribution as a foundation for graph representation learning more broadly.

---

> ### Author Rebuttal · Authors · 2026-03-31
>
> Thank you for taking the time to review our work. We appreciate your insightful concerns and hope our clarifications and additional experiments are helpful.
>
> ---
>
> **(W1) The view space is conceptually similar to multi-hop/scale methods.**
>
> We agree that both multi-hop/scale methods and GVT utilize multiple aggregations and may feel similar. However, they differ fundamentally in *how these views are used*.
>
> In multi-hop methods, multiple views are used to enrich information from different neighborhood ranges. These views are typically combined via concatenation or summation and followed by feature transformations. As a result, learning remains tied to a dataset-specific feature space, as in conventional GNNs. In contrast, GVT uses multiple views to construct a view axis as an alternative to the feature axis, and performs learning exclusively along this axis. This fundamentally different mechanism enables GVT to transfer across datasets, which is our core contribution and a property which is not achievable with existing multi-hop methods.
>
> While conceptually simple, this idea has not, to our knowledge, been explored in prior work. We believe this is because feature transformation has long been treated as the default paradigm in deep graph learning, making alternatives less apparent. Our work departs from this convention and demonstrates the empirical potential of view-axis learning.
>
> **(W2) Performance degrades with smaller pretraining datasets.**
>
> While RGVT shows degraded performance when pretrained on smaller datasets, we view this as expected rather than a weakness. Since RGVT transfers mappings of aggregation patterns, its effectiveness depends on the diversity and quality of structures seen during pretraining. Thus, degradation with smaller datasets is natural. Although GraphAny appears robust to pretraining choice, even our weakest variant (trained on Cora) outperforms its best variant.
>
> **(W3-1) RGVT still requires selecting depth L.**
>
> As the reviewer noted, RGVT requires selecting depth via validation, but the overhead is small in practice. The downstream predictor, a one-hidden-layer MLP, trains in usually seconds and this repeats only L (=8) times for validation. Empirically, depth selection and MLP training across 27 datasets take about total 152 seconds (Table 10), demonstrating low overhead.
>
> In terms of adaptation overhead, while GraphAny may appear faster, it relies on pseudoinverse computation in LinearGNNs, which can scale quadratically in the worst case and become slower than MLP training for high-dimensional features ([Figure C](https://shorturl.at/nwlma)). Therefore, we believe our MLP predictor approach is more scalable in practice.
>
> Moreover, recent fully inductive methods such as TAG (Hayler et al., 2025) and NodePFN (Choi et al., 2026) incur higher overhead. TAG uses ensembles of 14 predictors, including tabular foundation models with quadratic complexity in both nodes and features. NodePFN mitigates this via dimensionality reduction but requires per-dataset preprocessing choices, leading to a large search space (~400). Despite requiring far fewer adaptation resources, GVT achieves the best average accuracy and rank ([Table B, C](https://shorturl.at/gpIyY)), demonstrating state-of-the-art performance among fully inductive methods.
>
> **(W3-2) Sensitivity to suboptimal depth choices.**
>
> We appreciate this suggestion and have added analysis. As shown in [Figure D](https://shorturl.at/stwXM), optimal depth varies across datasets, with some favoring shallow models and others deeper ones, making validation-based selection important. However, [Figure E](https://shorturl.at/stwXM) shows that fixed depths (e.g., 4 or 6) remain competitive with strong GNN baselines on average. This suggests fixed depths are a reliable alternative when validation is not feasible. Developing depth-adaptive architectures is an important direction to further reduce this dependency.
>
> **(W4) Limited to node classification tasks.**
>
> Our primary goal is to examine whether learning solely on the view axis enables meaningful knowledge transfer across arbitrary datasets while remaining competitive with conventional methods. This is a non-trivial claim that must first be established, as most existing methods rely on feature transformation. For this reason, we focus on node classification as the first step, rather than addressing all task levels simultaneously. As noted, we acknowledge this limitation and believe that extending the framework to edge-level and graph-level tasks is an important next step. This will likely require self-supervised objectives and generative frameworks, and we view GVT as a key building block for such extensions.
>
> ---
> Thank you again for your time and thoughtful feedback. We will incorporate these discussions and additional experiments into the final version if given the opportunity. We hope our responses are helpful and that you find our work interesting.

---

> > ### Author Rebuttal · Reviewer_bQrd · 2026-04-06
> >
> > Thank you for the detailed rebuttal. My concerns have been adequately addressed overall.
> >
> > For W2, W3, and W4, the responses are convincing.
> >
> >
> >
> > For W1, while the distinction between view-axis learning and feature-axis learning provides a reasonable conceptual separation from multi-hop methods, I acknowledge that the novelty argument could be made more explicit in the revised manuscript. I encourage the authors to include this clarification in the final version.
> >
> > Overall, I maintain my score of 4.

---

> > > ### Author Response · Authors · 2026-04-06
> > >
> > > Thank you for your response. We are glad to hear that your concerns have been largely addressed.
> > >
> > > We sincerely appreciate your helpful feedback and suggestions, which led to additional discussions and experiments that improved the clarity and overall quality of our work. We will incorporate these improvements into the final version if given the opportunity.
> > >
> > > In particular, regarding W1, we will explicitly include an explanation of where learning occurs in our method compared to existing works, focusing on the distinction between feature-axis and view-axis learning. We will also expand the related work section to cover prior multi-hop/scale methods and clarify how our approach differs from them.
> > >
> > > We hope you found the review process engaging, and we sincerely thank you again for your time and consideration.

---

### Official Review · Reviewer_eywn · 2026-03-08

**Soundness:** 3
**Presentation:** 3
**Significance:** 2
**Originality:** 2
**Overall Recommendation:** 4
**Confidence:** 3

**Summary:**

To learn a model that works for arbitrary graph has been an critical and difficult in the graph learning community. The author identifies two properties for such inductive learning, equivariance for both node permutation and feature permutaion. To handle feature equivariance, the authors propose to use 'view' to expand the representation of single features into higher-dimensions, and use a view-channel function to map the high dimension information back to single features, enabling model to satisfy the two properties. The author then propose a recurrent variant of its proposed model to conduct pretraining on graph. By applying the pretrained model to various downstream dataset, the authors shows the inductive learning ability of the proposed model.

**Compliance With Llm Reviewing Policy:**

Affirmed.

**Final Justification:**

Originally I think the paper was on the boarderline, and the author then provided valid response address my concern on comparison fairness and key contribution, and I am prone to acceptance.

**Key Questions For Authors:**

- In terms of R2, a more interesting and significant question to consider is invaraince, which could potentially lift the requirement of training. I understand this might not be in the scope of this paper, but do you have results on this? Can be a strength.

**Limitations:**

I am not sure if the author include limitation discussion. What I would encourage is to include the fact that this is not fully inductive, and the features do not communicate.

**Strengths And Weaknesses:**

## Strength

The paper is generally well-written. Most claims are backed by meaningful theorem. The graph 'view' provides a solid way to handle the feature equivariance issue. The proposed GVT is also interesting, and capture higher-order graph information. These finding are well-backed by promising empirical results.

## Weakness

My main concern revolves around the inductive nature of the method.

- The key claim in the paper is to solve R2, feature equivariance, but looking at the approach, what the author does is essentially isolating each feature, there is no surprise that this will solve the feature equivariance problem. Notably, in the actual actual experiment, the author just computed some linear feature aggregation, then this is essentially local node feature pooling, which is not new to the literature at all. Some more advanced view finder should be studied to show that this framework is not overcomplicaing the problem to a framework but is demonstrating a new framework worth exploring.

- Because of the design, the model naturally ignores all inter-feature connection. Some form of this study should be conducted for examine the impact of this.

- The inductiveness this paper is referring to is in terms of the embedded feature, since a new classifier/ predictor still need to be trained for every new dataset. This has few implication: 1) the comparison to GraphAny can be unfair, graphany does not need training/optimization, this give advantages. 2) frankly, since training is still involved, this method should compare to more advanced supervised method like ones on ogb benchmark. The expectation is that the pretraining leads to performance improvement.

- Related to above, so far, looks like the improvement on supervised methods are minimal.

### Minor

- Equations has no label, making it difficult to refer to.

- Notation in R2 seems off. What comes out of $\Psi$ is NxH, but Q is F*F, you seem to assume same hidden and feature dimension, which is necessary for R2 to make sense.

---

> ### Author Rebuttal · Authors · 2026-03-31
>
> Thank you for your thoughtful feedback. We appreciate your detailed concerns and address them in a unified manner below due to space limitations.
>
> ---
>
> **(W1-1) No surprise that isolating each feature solves feature equivariance.**
>
> Applying a shared function to each feature may seem like an obvious way to achieve feature equivariance; however, this principle has largely been overlooked despite strong interest in learning across graph datasets. We believe this is because feature transformation has long been treated as the default paradigm, making models without it nontrivial to consider. Our key novelty is to depart from this convention, formalize the principle, and empirically show that it captures meaningful knowledge across arbitrary datasets while remaining competitive with conventional methods.
>
> **(W1-2) View finders are local feature pooling.**
>
> We agree that individual view finders are local node feature pooling.
> Our novelty, however, lies not in the specific choice of local aggregation, but in *where the learning takes place.* The key distinction is that:
>
> - conventional GNNs learn feature-mixing functions, while
> - GVT learns view-mixing functions.
>
> By shifting learning from the feature axis to the view axis, GVT makes transfer across arbitrary graphs possible, which is our main contribution. We consider view finders as a flexible design space, much like pooling operators in GNNs, and our results show strong performance even with simple choices. Exploring more advanced view finders is a valuable direction, but orthogonal to our core contribution.
>
> **(W2, Q) Feature invariance might be better than equivariance.**
>
> We agree that invariance is an appealing direction. However, invariant mappings typically rely on aggregation (e.g., pooling across features), which discards cross-feature interactions and therefore requires such interactions to be captured *prior* to aggregation. This is challenging in a fully inductive setting, where transferred datasets may exhibit substantially different feature relationships.
>
> In contrast, equivariant mappings can preserve such feature structure (i.e., without pooling), thereby deferring the modeling of cross-feature interactions to downstream predictors. This allows GVT to focus on learning transferable structural patterns while avoiding dataset-specific feature dependencies. Consistent performance across 27 transferred datasets supports this design choice.
>
> While invariance may remove the need for predictor training as noted by the reviewer, this advantage can be mitigated by pairing GVT representations with recent tabular foundation models (TFMs), which even further improves performance in our new experiment ([Table D](https://shorturl.at/PAj55)).
>
> **(W3) Comparison with GraphAny is unfair.**
>
> Thank you for raising this concern. We do not consider GraphAny to be training-free: its backbone predictor (LinearGNN) relies on pseudoinverse (p-inv), which is equivalent to fitting a linear model in closed form and must be recomputed for each dataset. Moreover, p-inv computation scales quadratically with the number of samples in worst case and can be more expensive than training an MLP ([Figure C](https://shorturl.at/nwlma)). We therefore believe the comparison is fair and that our method is more scalable.
>
> **(W3, W4) Current results are weak; stronger baselines are required.**
>
> Regarding supervised baselines, GNNs typically require extensive hyperparameter tuning to adapt to new datasets, often taking hours to days. In contrast, RGVT requires only training a small MLP, typically within a minute (Appendix H). Thus, it operates under a significantly lower adaptation budget and must rely heavily on transferred knowledge, making large gains over well-tuned supervised models inherently challenging. We also note that our baselines are stronger than those in recent fully inductive works ([Table A](https://shorturl.at/guiOy)).
>
> For a more direct comparison, we include three recent fully inductive methods: TS-Net (Finkelshtein et al., 2025), TAG (Hayler et al., 2025), and NodePFN (Choi et al., 2026). TS-Net, like GraphAny, relies on a p-inv predictor; NodePFN requires a dataset-specific preprocessing choices over a search space of 400; and TAG is an ensemble of 14 predictors (including 10 TFMs and 4 p-inv predictors). Despite substantially lower adaptation cost, GVT achieves the best average accuracy and rank ([Table B, C](https://shorturl.at/gpIyY)), demonstrating state-of-the-art performance among fully inductive methods.
>
> **Notations and limitations.**
> We will fix equation numbering, correct notation in R2, and add a discussion of limitations, including the need for per-dataset predictors and the lack of explicit feature mixing.
>
> ---
>
> Thank you again for your time and effort. We will incorporate the additional experiments and discussions above into the final version if given the opportunity. We hope our clarifications are helpful and that you find our work interesting.

---

> > ### Author Rebuttal · Reviewer_eywn · 2026-04-03
> >
> > My main concerns are resolved, I am prone to acceptance ( 3->4). While author's explanation on W1 and W2 does not add much to the original paper, I think the argument using TFM is a good plus.

---

> > > ### Author Response · Authors · 2026-04-03
> > >
> > > Thank you for your response. We are glad to hear that your main concerns have been addressed and appreciate your consideration toward acceptance.
> > >
> > > Your feedback led to meaningful discussions and additional experiments, which we believe have further improved the clarity and overall quality of our work. We will incorporate these improvements into the final version if given the opportunity.
> > >
> > > In particular, regarding feature equivariance, we will add a dedicated discussion section to more clearly present our motivation for adopting feature equivariance over invariance, along with the rationale behind this choice and its associated limitations. We will also expand the discussion on how tabular foundation models (TFMs) can mitigate the need for extensive predictor training, supported by the corresponding experimental results.
> > >
> > > We hope you found the review process engaging, and we sincerely thank you again for your time and consideration.

---

### Official Review · Reviewer_x7kx · 2026-03-13

**Soundness:** 3
**Presentation:** 3
**Significance:** 3
**Originality:** 3
**Overall Recommendation:** 4
**Confidence:** 2

**Summary:**

This paper addresses the problem of fully inductive node representation learning. The goal is for a single pretrained model to generate node representations for arbitrary unseen graphs that differ in size, topology, feature dimensionality, and semantics. The authors define a third representational axis  (i.e., the view space), which was obtained by stacking multiple adjacency-preprocessed versions of the node-feature matrix along a new dimension. For evaluation, RGVT is pretrained on OGBN-Arxiv and evaluated on 27 downstream benchmarks using only a lightweight, outperforming GraphAny and surpassing 12 individually tuned GNNs.

**Compliance With Llm Reviewing Policy:**

Affirmed.

**Key Questions For Authors:**

1. Since the output representation has the same dimensionality as the input features, in what sense should it be interpreted as a representation? Given that the predictor must be retrained for each dataset, could you clarify what it means for knowledge to be transferred?

2. Have you tried pretraining on multiple datasets simultaneously? What prevents this, and could it improve robustness?

3. Standard GNNs learn weights that mix features. GVT explicitly cannot do this due to R2. How does this affect performance on datasets where cross-feature interactions are important?

**Limitations:**

Yes

**Strengths And Weaknesses:**

- Strengths

S1. The view space concept is intuitive and well-presented.

S2. The paper provides formal proofs for the dual equivariance of GVT and connects the framework to the existing GNN literature.

S3. The evaluation covers many benchmarks spanning diverse feature types (signed/unsigned dense, sparse, binary, one-hot).

S4. Pretraining takes approximately 10 minutes and inference across 28 datasets requires only 52ms, making it practical.

S5. The Wilcoxon signed-rank test provides appropriate statistical validation of the improvements.

- Weaknesses

W1. The fact that the output representation preserves the original feature dimensionality is a significant limitation. This means the "representation" does not reside in a shared latent space across datasets; it is dataset-specific in both dimensionality and semantics. The predictor must still be trained separately for each dataset, and representations from different graphs cannot be directly compared. This considerably weakens the claim of a "unified representation" and limits downstream flexibility.

W2. The claim of new representational power beyond existing GNNs requires stronger justification.

W3. RGVT is pretrained on a single dataset (OGBN-Arxiv). Table 12 shows sensitivity to pretraining dataset choice, with smaller datasets yielding worse results. A multi-dataset pretraining strategy has not been explored.

W4. The comparison with GNNs is asymmetric in a way that may favor RGVT. While the paper correctly notes that GNNs are individually tuned per dataset (which favors them), RGVT also selects recurrent depth L per dataset and trains a per-dataset predictor.

W5. Although the complexity scales linearly with F, the cost becomes substantial for graphs with very high-dimensional features.

---

> ### Author Rebuttal · Authors · 2026-03-31
>
> Thank you for your detailed review and thoughtful feedback. We appreciate your insightful concerns and address them in a unified manner below due to space limitations.
>
> ---
>
> **(W1, Q1) Meaning of representations; what knowledge is transferred?**
>
> We believe that a *representation* is defined not by changes in dimensionality, but by whether it encodes meaningful (or *semantic*) information useful for downstream tasks. GVT effectively injects the structural information of a graph into the feature space, allowing downstream predictors to consistently outperform models that rely on raw features alone (Table 1).
>
> From this perspective, what GVT transfers between datasets is *which structural information should be embedded into the features.*
> This is modeled as a learnable mapping from view vectors to aggregation rules.
> Our new $t$-SNE visualizations ([Figure B](https://shorturl.at/nwlma)) further support this interpretation: on unseen datasets, GVT stably encodes structural information while preserving feature semantics, improving class separability. Consistent gains across 27 transferred datasets indicate that this mechanism generalizes well.
>
>
> **(W1) Representations from different graphs are not directly comparable.**
>
>
> We would like to clarify that our notion of a shared space refers to the *view space*, whose dimensionality remains fixed across datasets regardless of the number of nodes or features. The representation learner defined on this view space is shared, and each node or graph is represented as a set of view vectors in the shared space. Although the resulting node embeddings are not directly comparable (e.g., cosine similarity) because their sizes may differ, GVT still enables comparison at the level of distributions or sets of view vectors. We believe is a promising direction of cross-graph comparability.
>
> **(W3, Q2) Sensitivity to the pretraining dataset and multi-dataset extension.**
>
> It is not surprising that RGVT performs worse when pretrained on smaller datasets, as its effectiveness depends on the diversity and quality of structural patterns observed during pretraining. Pretraining on multiple datasets could potentially improve its generalization, but extending the current framework in that direction is not straightforward.
>
> In particular, supervised pretraining requires jointly optimizing dataset-specific predictors, which introduces additional complexity. Moreover, because different graphs tend to favor different model depths, joint optimization under a fixed architecture is likely to be suboptimal. Addressing this issue may require a self-supervised objective with a shared, depth-adaptive autoencoder, which we leave as an important direction for future work.
>
>
> **(W4) Asymmetry in comparison with GNNs.**
>
> We agree that our comparison between GVT and GNNs is asymmetric, but this reflects a fundamental difference in paradigm and, in practice, favors GNNs. GNNs require extensive hyperparameter tuning to adapt to new datasets, often taking hours to days. In contrast, RGVT requires training an MLP only for a few trials, typically within a minute (Appendix H).
>
> For a more direct comparison, we include recent fully inductive GFMs in [Tables B and C](https://shorturl.at/gpIyY) which are also designed to adapt to new datasets without training. Notably, TAG (Hayler et al., 2025) uses an ensemble of 14 predictors, and NodePFN (Choi et al., 2026) requires dataset-specific preprocessing choices from a search space of 400.  Despite substantially lower adaptation cost, RGVT outperforms these baselines, demonstrating state-of-the-art performance among fully inductive methods.
>
> **(W5, Q3) Complexity and cross-feature interactions.**
>
> Not modeling the cross-feature interactions is our important design choice to satisfy feature equivariance (R2).
> To our knowledge, R2 can be satisfied either by sharing a function across features (as in GVT) or by using feature-wise self-attention (e.g., transformers), which can model cross-feature interactions but incurs quadratic complexity in the number of features. We adopt the former because quadratic complexity is impractical for high-dimensional dataset.
>
> Importantly, this choice does not imply that cross-feature information is lost. Since GVT does not *compress* (or reduce) features, the cross-feature information can still be captured by downstream predictors. We believe this design is particularly suitable for fully inductive settings, where unseen datasets may share structural patterns with the pretraining data but differ in feature relationships.
>
> ---
>
> Thank you again for your time and effort. We will incorporate discussions and additional experiments into the final version if given the opportunity. We hope our responses are helpful and that you find our work interesting.

---

> > ### Author Rebuttal · Reviewer_x7kx · 2026-04-03
> >
> > My main concerns have been adequately addressed, and W4 and W5 seem to be inherent limitations. However, the work has good potential and value for the graph research community, so I would like to maintain my positive score.

---

> > > ### Author Response · Authors · 2026-04-03
> > >
> > > Thank you for your response. We are glad to hear that your main concerns have been adequately addressed.
> > >
> > > We sincerely appreciate your helpful feedback and suggestions, which led to additional discussions and experiments that improved the clarity and overall quality of our work. We will incorporate these improvements into the final version if given the opportunity.
> > >
> > > Regarding the remaining limitations, we will continue working toward a more generalizable and impactful framework for graph foundation models. We believe this work provides key insights and that graph view transformation can serve as an important building block for future research.
> > >
> > > We hope you found the review process engaging, and we sincerely thank you again for your time and consideration.

---

### Decision · Program_Chairs · 2026-04-30

**Decision:**

Accept (regular)

**Comment:**

All four reviewers recommended accepting the work. According to the reviews, the paper has the following advantages:
1. The paper introduces a new "view space" that allows a single model to learn from graphs with different types of node features.
2. The proposed method outperforms 12 specialized GNNs across 27 different datasets.
3. The model is very efficient, requiring only 10 minutes to pretrain and just seconds to adapt to new datasets.
4. The work also provides some mathematical proofs for why the model works.

Therefore, I believe that the paper should be accepted for publication.